# The non-opponent nature of colour afterimages
Christoph Witzel ✉

Complementary colour afterimages have driven our understanding of human colour perception since the foundations of modern colour science. Despite their fundamental importance, decades of research have failed to establish the precise nature of colour afterimages and the neural mechanisms of adaptation at their origin. To date, it is unclear whether afterimage formation is caused by adaptation in the cone photoreceptors, of colour-opponent neurons in the subcortical pathway, or requires the assumption of yet unknown cortical mechanisms. To establish the neural mechanisms underlying afterimage formation, this study exploited the fact that different candidate mechanisms make fundamentally different predictions about the hue and saturation of afterimages. Using tailormade experimental paradigms, the exact colours perceived in afterimages were measured for a large range of inducers to test those predictions. Three experiments tested predictions of afterimage hue and saturation with varying inducer hues, and changes of afterimage hues depending on inducer saturation (Exp. 1.a: 8 colours, tested across 31 participants; Exp. 1.b: 24 colours, tested across 52 participants; Exp.2.a: 72 colours, replicated with 10 participants; Exp. 2b: 72 colours, replicated with 2 participants; Exp. 3: 48-216 colours, replicated with 5 participants). Results across all three experiments very consistently demonstrated that afterimage colours are not colour-opponent, as widely assumed, but closely follow a quantitative model of adaptation in the cone photoreceptors. These findings unequivocally establish the origin of afterimages along the hierarchy of neural processing, hence resolving all prevailing misconceptions and contradictions. By linking the perceptual nature and the neural origin of afterimages, the present paradigm also provides a technique for probing those neural mechanisms behaviourally and in first-person experience.

A colour afterimage arises, when an observer fixates a coloured area over a sustained period[1,2]. Through this exposure, the underlying neural mechanisms of colour processing adapt and become insensitive to the colour of that area, the *inducer*. These mechanisms remain most sensitive to the colour that is least similar to the inducer and produce a complementary (or negative) afterimage when replacing the inducer by a colourless grey area. For example, if you fixate on a yellow circle, when taken away you will see a blue-purple circle even though the circle's area is now physically the same as the colourless surround (see Supplementary Movie 1 and section *A.1 Illustration of Afterimages* of the Supplementary Material). The underlying mechanisms of temporal adaptation to colour enable us to reliably perceive colour even though lighting constantly changes in our everyday environment. Illusory afterimages have thus been used to disseminate the most fundamental tenets of human colour perception in reviews, textbooks, and other broader communications, including the concepts of chromatic adaptation and colour constancy, colour opponency

and complementarity, and the dependence of colours on the context and the beholder[3–8].

Although candidate mechanisms of adaptation have been known for more than a century[9], state-of-the-art explanations of this important phenomenon remain contradictory and confused. Human colour perception is the result of several stages of processing, starting with the excitation of the cone photoreceptors in the eyes, propagating through cone-opponent mechanisms in the retinal ganglion cells, the LGN of the thalamus, and the single- and double-opponent cells in the primary visual cortex, until the colour signal reaches higher cortical areas that produce the subjective experience of colour[10]. Evidence for adaptation has been found at many points along this visual hierarchy. Adaptation of photoreceptors at the first stage of colour processing produces divisive adaptation, similar to von Kries' original proposal[9], and can be approximated by contrast coding following Weber's law[11–16]. A slower type of "second-site adaptation" occurs in the retinal ganglion cells[17], and

School of Psychology, University of Southampton, Southampton, UK. ✉e-mail: c.witzel@soton.ac.uk

is assumed to be subtractive[11,18–20]. Other observations suggested "higher-order" adaptation of cortical mechanisms[3,21–24].

Researchers have disagreed about the neural mechanisms of adaptation that cause complementary afterimages[6,20,25,26]. Classical studies suggested that such afterimages are caused by photoreceptor desensitisation ("cone bleaching") at the first stage of processing[27–31]. Later studies firmly contradicted this idea[20,32,33], attributed afterimages to adaptation in the cone-opponent channels of the second stage[20,32,34], or emphasised the effects of cortical adaptation[33,35–43]. Even more importantly, the link between neural mechanisms and the perceived colours of afterimages has been ignored or misconstrued. Irrespective of the hypothesised neural origin, most state-of-the-art research assumes that the perceived colours of afterimages are cone-opponent[20,34,36,38,44], but there is evidence against that assumption[25,45]. As a result of this confusion, textbooks, reviews, and other broader communications either do not attempt any explanation of colour afterimages[2,4] or disseminate misleading narratives, such as the idea that afterimages reflect Hering-opponency between red/green and blue/yellow[6,8].

This study leveraged the fact that adaptation of mechanisms at different stages along the hierarchy of colour processing make markedly different quantitative predictions about the colours of complementary afterimages (Fig. 1). Predicted hue and chroma of afterimages differ across models. Hue describes how reddish, yellowish, greenish, and bluish a colour is. Chroma refers to the colourfulness of a hue. At equal brightness (isoluminance), it is equivalent to saturation and corresponds to the difference of a colour from grey. Using tailormade experimental paradigms, the exact colours perceived in afterimages were measured for a large range of inducers to test those predictions. Three comprehensive experiments unambiguously showed that the colours of afterimages tightly follow the non-opponent predictions that are specific to cone adaptation.

## Methods

*Experiment 1* was approved by the ethics committee at the University of Gießen (LEK 2017-0030). *Experiments 2 and 3* were approved by the Faculty Ethics Committee at the University of Southampton, ERGO 65442. All participants provided informed consent prior to participating. None of the experiments were preregistered. All information about sex is provided by participants.

### Experiment 1: Fixed-Location Afterimages

**Participants**. In *Experiment 1a*, 31 observers participated (25 women and 6 men, age: $M = 25.9 \pm SD = 4.2$ years). In *Experiment 1b*, 52 observers took part (36 women and 16 men, age: $25.1 \pm 4.3$ years).

Participants were compensated by course credits or €8 per hour. Colour vision deficiencies were excluded using the HRR plates[46].

**Apparatus**. Stimuli were presented on an *Eizo Colour Edge Monitor* (36.5 × 27 cm) with an *AMD Radeon Firepro* graphics card with a colour resolution of 10 bits per channel. The *CIE1931* chromaticity coordinates and luminance ($xyY$) of the monitor primaries were $R = (0.6847, 0.3111, 26.4)$, $G = (0.2138, 0.7263, 69.9)$, and $B = (0.1521, 0.0453, 4.8)$. Gamma was 2.2 for all channels and has been corrected.

**Stimuli**. Colours were represented in CIELUV space. The white-point was $xyY = [0.3304, 0.3526, 101.1]$, background lightness was $L^* = 70$. At isoluminance, opponent hues in CIELUV are the same as cone-opponent hues in Derrington-Krauskopf-Lennie (DKL) space[47] (Fig. 1a) and as opponent hues determined along lines in (gamma-corrected) HSV and RGB space[6,25], tristimulus values (XYZ) and chromaticity coordinates (xyY); see sections *E.2-3* of the Supplementary Material for mathematical details[45]. CIELUV space was preferred to DKL space because it better controls for perceived chroma[48,49]. The eight inducer colours in *Experiment 1a* were chosen to correspond to the typical lightness and hue of red, orange, yellow, green, turquoise, blue, purple, and magenta at the maximum chroma possible within monitor gamut (coloured lines in Fig. 1a). The nine hues of the comparison colours (cf. *Procedure*) were obtained by adding four hues in 10-degree (deg) steps to either direction (low or high azimuth) of the opponent hue. Lightness and chroma of the comparisons were determined through piloting. In *Experiment 1b*, twenty-four inducers were sampled along a hue circle in CIELUV at chroma 71 and equal steps of 15 deg starting at 0 deg (black circles in Fig. 1a). Chroma was chosen to be the highest chroma achievable within monitor gamut for all hue directions. Lightness of inducers, comparison colours, and the achromatic disc in the centre of the test display were the same as the background ($L^* = 70$). This brightness corresponds to 1.7% cone bleaching according to the Rushton and Henry half-bleach constant[50]. Hues of comparison colours were determined as in *Experiment 1a* and chroma was kept constant at 30 for all comparison colours. This level of chroma was determined through piloting and accounts for the lower saturation of the afterimages compared to the inducers. Table S2 in the Supplementary Material provides detailed colour specifications.

**Procedure**. In each trial, the inducer display (Fig. 2a) was shown for 20 s (*Experiment 1a*) and 30 s (*Experiment 1b*). A cover task ensured that the observer fixated a dot in the centre (details in section *B.2 Details on*

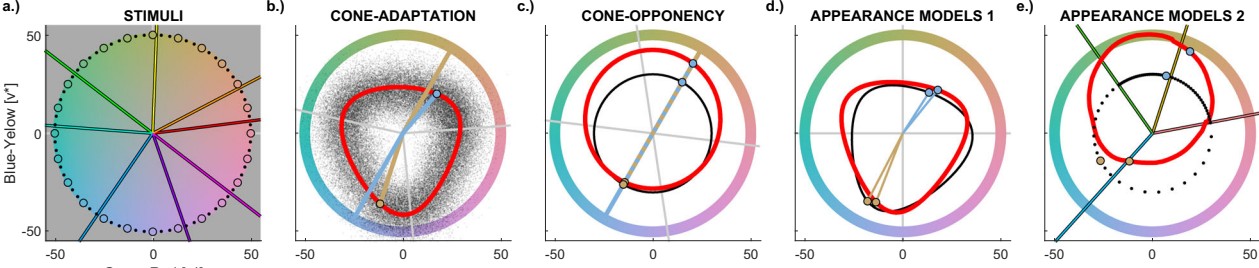

**Fig. 1 | Afterimage stimuli and models.** In all panels, colours are represented in CIELUV colour space; see Fig. S11 of the Supplementary Material for analogous diagrams in cone-opponent DKL space. **a** illustrates the stimulus samples in *Experiment 1a* (coloured lines), *Experiment 1b* (black circles), and *Experiment 2a* (black circles and dots). In *Experiment 2b* stimuli were sampled along a hue circle as in *Experiment 2a*, but in DKL instead of CIELUV (Fig. S11.a). **b** shows the results of cone-adaptation without (red line) and with noise (black dots); the noise was used for modelling hue histograms. **c** illustrates adaptation of the cone-opponent channels, where the afterimage is shifted to the hue opposite to the inducer hue (thick red line). For comparison, predictions based on adaptation in CIELUV space (thin black line) are also shown. CIELUV predictions of hue are the same as for cone-opponency

DKL space but predicted chroma differs from cone-opponent predictions. **d** illustrates predictions of CIECAM02 (thick red line) and CIELAB (thin black line), which are common models of colour appearance. **e** illustrates Hering-opponency (black) and opponency in the Munsell colour system (red). The coloured lines in the background correspond to measured red, yellow, green, and blue. Two cone-opponent examples of inducers (60 and 240°) are shown in (**b**–**e**). The crossing point between lines and hue circle indicates the inducer, large dots indicate the colour of the corresponding afterimage predicted by the respective model. Lines and dots are shown in the inducer colour. For first-person inspection, the yellowish inducer (60°) corresponds to the inducer in Supplementary Movie 1.

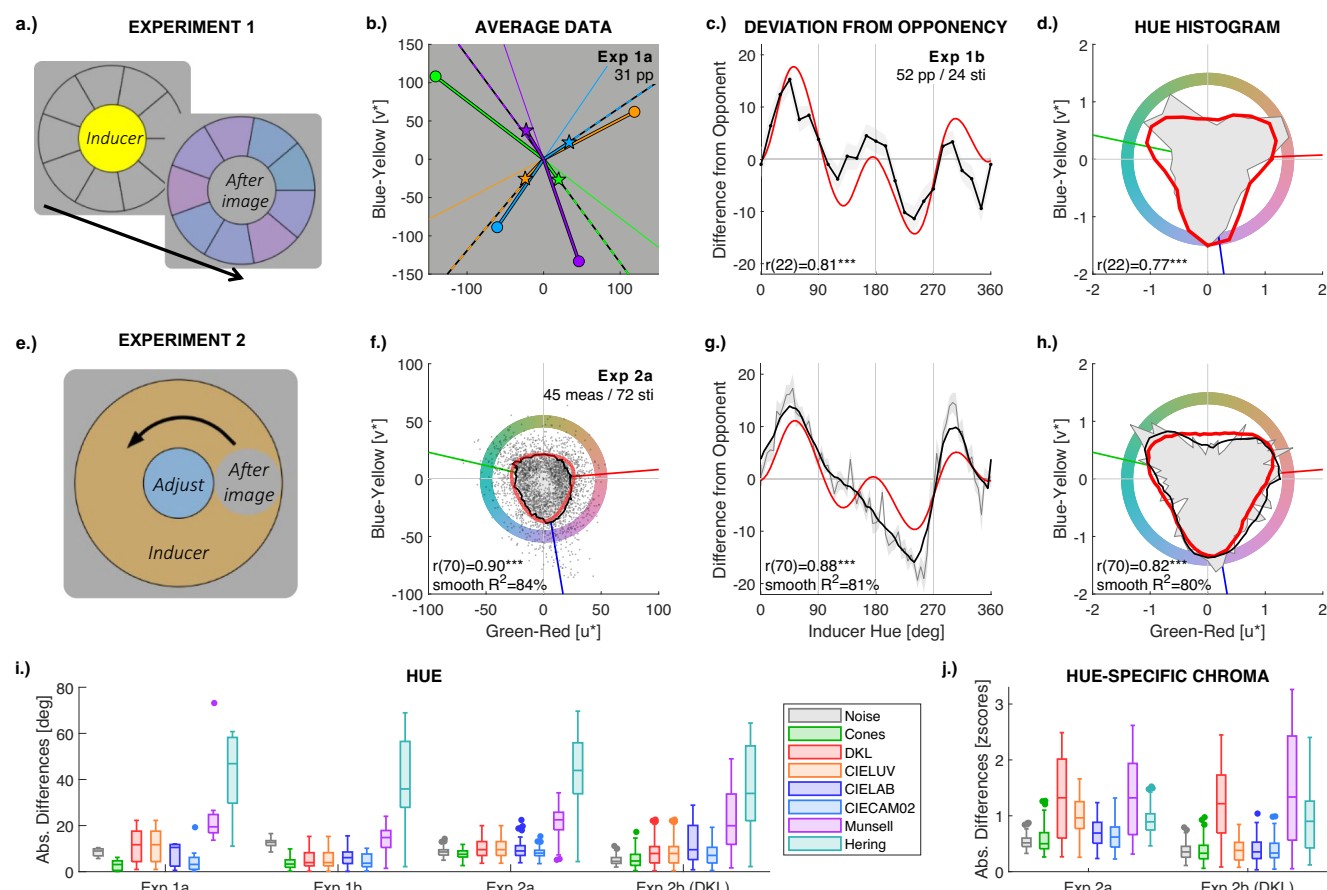

**Fig. 2 | Experiment 1-2.** The first row (**a–d**) illustrates stimulus displays and task (**a**), and results (**c**, **d**) for *Experiment 1*. Observers adapt to the inducer display for 30 s (yellow in **a**). Then the inducer display is replaced by a circle of background grey together with nine comparison colours (purple in **a**) in a circular arrangement around the grey circle. The grey circle in the centre is perceived as the induced afterimage, and observers select the comparison colour that best matches the colour they see in the centre. **b** gives examples of afterimage matches averaged across $N = 31$ participants in *Experiment 1a* with maximally saturated colours. Discs correspond to inducers, stars to average matches of perceived afterimages. The thin line in the background indicates the cone-opponent direction, the dashed line the prediction of the cone-adaptation model (see Figs. S2a and S3 for results with the other four colours). **c**, **d** illustrate results from $N = 52$ participants in *Experiment 1b*. The second row (**e–h**) corresponds to *Experiment 2a* with the chaser-like paradigm. In (**e**), the chromatic ring is the inducer. Participants fixate the centre, and the moving grey circle on the ring reveals the afterimage. Observers adjusted hue and chroma of the centre circle to match the moving one. Average afterimage matches from $N = 10$ participants and overall $N = 45$ measurements per colour (black curve in **f**) closely correspond with predictions by cone adaptation (red outline), yielding a correlation (bottom left) between measured and simulated afterimage intensity (i.e., chroma) across the $N = 72$ inducer colours. The curves in the third column (**c**, **g**) show the deviation of the cone-adaptation model (red curve) and of the measured afterimages (black curve) from the colours opponent to the inducers. Correlations between simulated and measured deviations from opponency are shown at the bottom of the

diagrams. The correlation reflects the high similarity in profile of the two curves. The hue histograms in the last column (**d**, **h**) counts hue responses (azimuth in **f**) and displays the resulting frequencies as a function of azimuth in a polar plot. The histogram of the measurements is shown by the grey area in the background, and the histogram of the simulated afterimages by the red outline. Panel d provides the hue histogram for the $N = 52$ measurements for 24 colours in *Experiment 1b* and (**h**) those for the $N = 45$ measurements for 72 inducer colours in *Experiment 2a*. In (**h**), a smoothed version of the hue histogram is shown by the black line. Hue histograms feature three clusters that closely correspond with model predictions (red line), hence confirming the results from (**c**, **f**, **g**). Additional results from *Experiment 2b* with stimuli in DKL space are provided by Fig. S12 of Supplementary Material. The last row illustrates average deviations of different model predictions from measured afterimage hue (**i**) and chroma (**j**). The top and bottom edges of each box are the upper and lower quartiles, the line inside each box is the median; whiskers are the minimum and maximum values that are not outliers, and dots outside the whiskers are outliers identified as values outside the interquartile range (box) by 1.5 times the size of the interquartile range. The grey symbols (*Noise*) provide estimates of noise calculated as the average difference of each individual's measurement from the group mean (interindividual variation) in *Experiment 1* and of each single measurement from the mean of each participant (intraindividual variation) in *Experiment 2*. Chroma has been z-scored to focus on relative differences across hues. Supplementary Movies 1–4 visualise the differences between opponent and afterimage hues (see also section *A* and Fig. S1 of Supplementary Material).

*Afterimage measurements* of the Supplementary Material). After this adaptation period, observers determined which of the nine comparisons was the closest match to the afterimage colour while seeing the afterimage in the centre (Fig. 2a). Afterimages appeared to 'melt into' the segment with the colour that looked closest to the afterimage, making the task intuitive. Participants used the mouse to indicate the segment with the matching comparison. If they could not see any of the comparison colours in the centre, they could also click on the centre disk to indicate this and skip the trial (resulting in a missing value). A 10-s intertrial display

followed by a self-parsed break was used to cancel remaining afterimages and to prevent afterimage interference across trials (see section *B.2 Details on Afterimage measurements* of Supplementary Material). The order of trials was randomised. In *Experiment 1a*, trials were repeated across four blocks. Few trials were skipped (about 2% in both parts).

**Colour naming and control task.** For details about the measurement of colour categories and prototypes and about the control task, see sections *B.3-4* of the Supplementary Material.

## Experiments 2-3: Chaser-Like Afterimages

Tables S3 and S4 in the Supplementary Material provide details about participants and apparatus in *Experiments 2 and 3*. Supplementary Movies 1–4 and section *A.2 Illustration of the Method in Experiments 2-3* in the Supplementary Material provide a simplified illustration of the tasks in *Experiments 2 and 3*.

**Participants**. Ten volunteer participants (6 women, and 4 men, age: $M = 23.4 \pm SD = 8.3$ years), including the author (*CW*, male, 43 years old) took part (cf. Table S4) in *Experiment 2a*. Three more observers had started one round of measurements (8 hues) but did not come back to complete a sufficient dataset for all 72 hues. *CW* and a naïve, 37-year-old female observer (*f1*) participated in *Experiment 2b* (cf. Table S4). In *Experiment 3*, measurements across chroma were done by the author (*CW*), and four naïve female, 19-year-old participants (Table S5).

**Apparatus and stimuli**. Three different experimental set-ups were used in *Experiments 2–3* and calibrated as explained for *Experiment 1* (Table S4–S5). All measurements were conducted in ambient darkness. Figure 2e illustrates the stimulus display (see also Supplementary Movies 1–4 and Fig. S1). In *Experiment 2a*, 72 inducer colours were sampled in 5-deg hue steps along an isoluminant hue circle at $L^* = 70$ in CIELUV-space. This sampling resolution is almost exhaustive, considering that hue discrimination thresholds are larger than 5 degrees[48]. The brightness corresponded to about 3–3.5% cone bleaching according to the Rushton and Henry half-bleach constant[50]. Inducer chroma was set to be maximal within the respective monitor gamut, resulting in a chroma of 38 (one participant), 42 (two participants), or 50 (everyone else); for details see Table S4. In *Experiment 2b*, the sampling was done along an isoluminant hue circle in DKL space with a radius of 0.5 (*CW*) and 0.7 (*f1*), as illustrated by Fig. S11 of the Supplementary Material. The chromatic axes were scaled relative to the monitor gamut to avoid gamut artefacts (see section *E.2 Cone-Opponent Model (DKL)* of the Supplementary Material for mathematical details). For measurements across chroma in *Experiment 3* (Fig. 3), additional chroma levels were measured at 20, 30, and maximum within gamut (see Table S5 for details).

**Procedure**. In each trial, participants were asked to fixate the centre of the display until the moving circle reached maximum colourfulness. Then, they used the cursor keys to adjust the hue (left/right) and saturation (up/down) of the centre circle. The initial colour of this circle was set to a random hue at inducer chroma. During a coarse adjustment, participants could continuously change colours; but before confirming the adjusted colour, participants needed to do a fine adjustment (by holding space). In the fine adjustment, the colour changes by single steps (in radius or in 1 deg azimuth) with each separate key press.

Since the adjustment takes time, observers would see afterimages from the adjusted colour attenuating the perceived chroma of the adjusted colour. Two measures were taken to avoid this: During coarse adjustment, black and white circles (not areas) were extending from the centre of the adjusted disk to its rim. These circles would disappear during fine adjustments to avoid interference with the finalised match. In addition, holding the control key would turn the centre disk temporarily into grey until the key was released. Observers were asked to wipe out unwanted afterimages from the adjusted centre disk using that key before confirming the adjustment. So, they would hold control, wait and move their eyes until the centre disk appeared achromatic. They would then stare at the grey circle until the moving circle reached maximum chroma. Only then would they release the key and compare the colour they had previously adjusted with the colour of the moving circle. Typically, the adjusted colour was too saturated due to the overlaying of its own afterimage during the adjustment. So, participants needed to lower chroma after readaptation. They reiterated this procedure until the chroma did not need adjustment after releasing the control key. Only then did participants confirmed their adjustments by pressing enter.

As in *Experiment 1*, an intertrial period was used to prevent afterimage carry-over across trials.

Inducer colours were split into nine series of eight colours. Within a series, the eight colours were separated by 45 degrees in azimuth. Different series were defined by different starting points (0, 5, 10…40 degree) so that the series together covered all 72 inducers. Each block of measurements featured one trial for each of eight inducer colours in a series, presented in random order. Measurements of each block were repeated up to five times (cf. Figs. S8–10). Blocks of measurements were spread over several days. Prior to measurements, participants were trained with practice blocks with the presence and feedback of the experimenter (*CW*) to make sure they understood task handling and the aim of measuring the illusory afterimage colour as precisely as possible.

## Models

With all models, the colour signal of the grey probe circle (same as background colour) was calculated under local adaptation to the inducer. Then, the colour of the resulting afterimage was determined as the locally adapted colour signal under global adaptation to the grey background. The known fact that afterimages are not as saturated as inducers (Fig. 2f) implies that adaptation to inducers is not complete. As an estimate of the strength of adaptation, adapting chroma for modelling afterimages was set to the chroma of the comparison colours in Experiment 1, and to the grand average chroma (27.0 in CIELUV and 0.47 in DKL) of adjustments in *Experiment 2*. Even if complete adaptation to the chroma of inducers had been assumed, results would be largely the same. However, not surprisingly, correlation coefficients involving cone adaptation would be slightly lower because non-linearities in the model are then higher than in the measurements. There are no free parameters in any of the models.

### First-stage adaptation

Cone adaptation at the first stage of colour processing was modelled by cone contrasts[11–16]. Cone contrasts (*CC*) are Weber fractions, calculated as the difference between cone excitations of the stimulus (*LMS*) and cone excitations of the adapting colour (*LMS0*) relative to the cone excitations of the adapting colour: $CC = (LMS\text{-}LMS0)/LMS0$. Cone contrast is calculated independently for the short- (S), medium- (M), and long-wavelength (L) sensitive cones, resulting in S-cone, M-cone, and L-cone contrasts. In psychophysical experiments, adaptation is typically controlled by the colour of the background, which is then *LMS0*. To compute the induced colour of the afterimage, the roles are swapped because afterimages correspond to the perception of the achromatic background after local adaptation to the inducer. Local adaptation to the inducer is modelled by inserting the cone excitations of the inducer, *iLMS*, instead of the background into *LMS0*. As the afterimage is elicited on a grey probe, *LMS* now corresponds to *bgLMS*, i.e., the achromatic, isoluminant grey of the background:

(1) Cone adaptation to inducer:

$$CC_{LMS} = \frac{bgLMS - iLMS}{iLMS}$$

Where *bgLMS* and *iLMS* refer to Stockman-Sharpe cone excitations of the grey background and the adapting inducer. Cone excitations were scaled to match the luminous efficiency function[51]. Cone contrast changes with increasing cone adaptation *iLMS* according to a multiplicative inverse function (because *iLMS* increases in the denominator). This adaptation produces shifts towards the peak cone sensitivities (Fig. 1b, Fig. S14).

### Second-stage adaptation

Cone-opponency has been modelled with DKL colour space[47]. Details on the computation of the cone-opponent axes are provided in section *E.2 Cone-Opponent Model (DKL)* of the Supplementary Material. Cone-opponent adaptation is assumed to be subtractive[11,19]. So, it was calculated by

**Fig. 3 | Change of Afterimage Hue with Chroma (Experiment 3). a** Visualisation of the effect of inducer chroma on afterimage hue. Numbers in the coloured rectangles indicate hue. Rows correspond to example inducer stimuli of constant hue (60 deg) and different chroma levels (20–70), the cone-opponent hue (240 deg), the prediction by cone-adaptation with varying hue, and the average hue of the prediction that is used for the analyses. The variation in chroma predicted by cone adaptation has been ignored here to facilitate the visual comparison of hue changes. **b** $N = 216$ inducer stimuli (coloured dots) varying across three levels of chroma (20, 50, maximum). The grey circles in the background indicate chroma (radius) varying from 10 to 80 in steps of 10. The doted tetragon shows the monitor gamut at that luminance level. **c** Afterimages modelled through cone contrasts. The black lines illustrate how afterimages vary when inducer chroma corresponds to the grey circles. Coloured dots indicate simulated afterimages for measurements with the $N = 216$ inducer stimuli of (**b**). **d** Comparison between measurements and cone-adaptation predictions. Grey dots are the $N = 216$ measurements; thick transparent black lines highlight the change across chroma for hues at 45-deg intervals. Grey lines indicate the reference average used to calculate hue-specific chroma differences. Red lines show the corresponding cone-adaptation predictions. **e, f** Scatterplots illustrating the correlation between simulated and measured changes of afterimage hue across chroma. The axes indicate the afterimage hue difference from the cone-opponent hue (**e**) and from the reference average (**f**) in azimuth degree. Dot colours indicate inducer colours. This figure shows results for observer *f7*. Results for four other observers can be found in Table 4 and Fig. S13, each replicating these results.

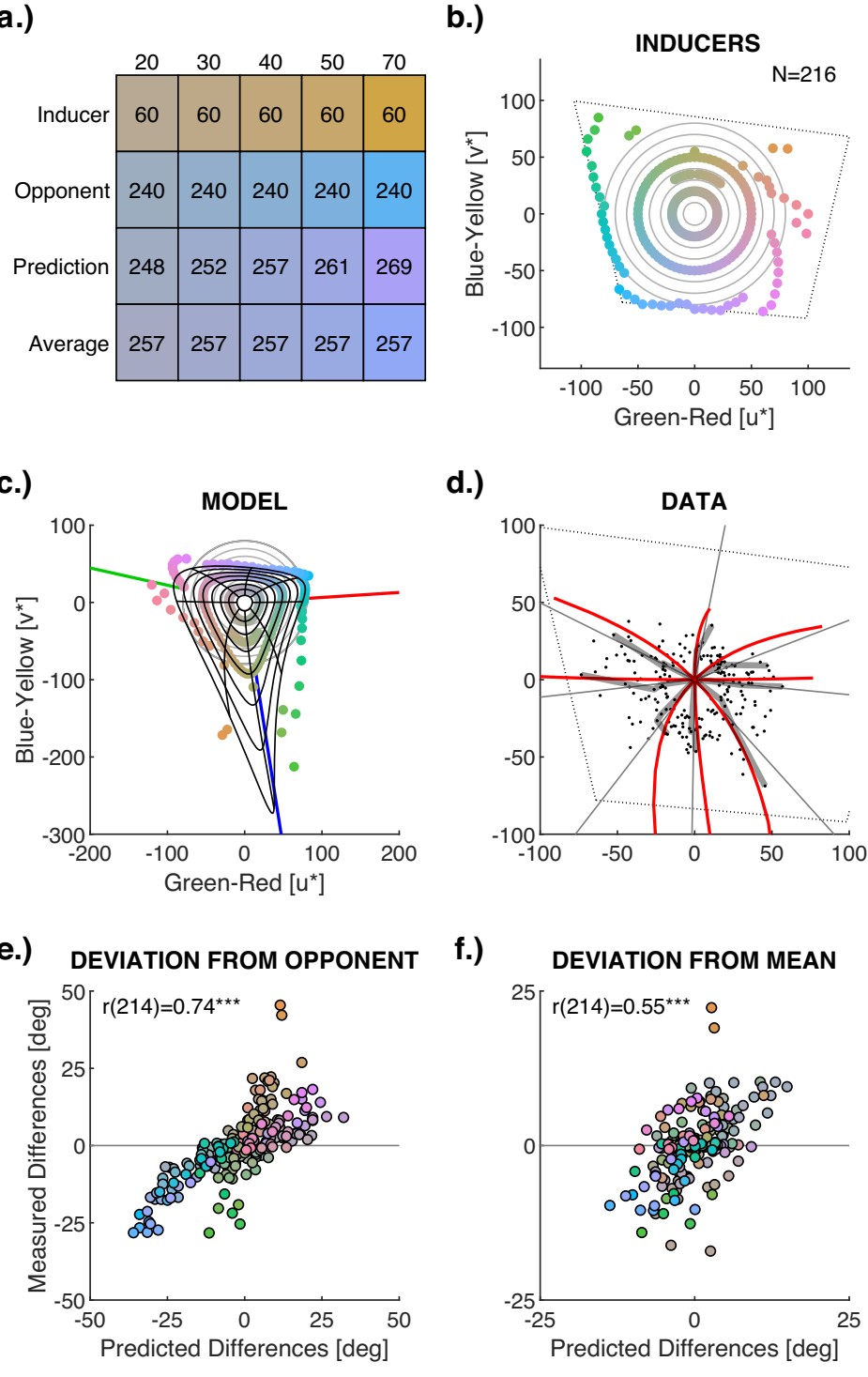

subtracting the cone-opponent signal of the adapting colour from the cone-opponent signal of the achromatic probe[20].

(2) Cone-opponent adaptation to inducer:

$$DKL_{L-M} = bgDKL_{L-M} - k * iDKL_{L-M}$$

$$DKL_{S} = bgDKL_{S} - k * iDKL_{S}$$

Where $DKL_{L-M}$ and $DKL_{S}$ are the adapted cone-opponent signals of L–M and S mechanisms, respectively, and $bgDKL$ and $iDKL$ are the cone-opponent signal of the background and inducer, respectively. The constant k indicates the strength of adaptation. As the background is achromatic, $bgDKL$ is 0 in all channels, and the equations can be simplified to:

$$DKL_{L-M} = -k * iDKL_{L-M}$$

$$DKL_{S} = -k * iDKL_{S}$$

Therefore, cone-opponent adaptation produces proportional shifts along the DKL axes, predicting cone-opponent afterimage hues to be 180

degrees rotated away from the inducer and afterimage chroma that is k times the inducer chroma. Here, k was set to match the estimated strength of the afterimages, but the value of k was irrelevant to the results. At isoluminance, the cone-opponent mechanisms can be expressed as a function of cone-contrasts, resulting in:

$$DKL_{L-M} = -(iCC_M - iCC_L)$$

$$DKL_S = -iCC_S$$

Where *iCC* is the cone contrast of the inducers calculated with *equation (1)*, and the index indicates the respective cone (see section *E.2 Cone-Opponent Model (DKL)* of the Supplementary Material). Although a theoretically complete model would differentiate between the two separate mechanisms that constitute each opponent channel ("half-wave rectification"), such a separation did not change the predictions in this study.

### Adaptation of colour appearance

Predictions by colour appearance models (CIELUV, CIELAB, CIE-CAM02) were computed as the colour appearance of the grey background when the adapting white-point was the inducer, using the respective adaptation transform. At isoluminance, CIELUV is a projective transformation of DKL space and involves a subtractive adaptation, which explains the identity of predicted hues. Sections *E3-E5* in the Supplementary Material provide mathematical details and explanations. For predictions by Munsell and Hering colours, opponent colours were determined in the respective coordinate systems, which is equivalent to the effects of subtractive adaptation. Opponent colours in the Munsell system were interpolated using the CIELAB coordinates of the Munsell renotation table[52] as provided by the *Munsell Lab of the Rochester Institute of Technology*. Details about the interpolations are provided in section *E.6 Munsell-Opponent Model* of the Supplementary Material. Hering-opponent colours were calculated by linearly interpolating the hue direction between the respective two empirically measured prototypes for red, yellow, green, and blue in CIELUV space. There was no prediction for Hering chroma. For details, see section *E.7 Hering Opponent Model* of the Supplementary Material.

### Data distributions

Measurements of hue and chroma were approximately normally distributed across repeated measurements and participants, as checked by histograms (Figs. S3–S5) and normal probability plots (Figs. S6–S7).

### Reporting summary

Further information on research design is available in the Nature Portfolio Reporting Summary linked to this article.

### Results

Figure 1 illustrates the colour stimuli (Fig. 1a), and the corresponding predictions of afterimage hue and chroma by candidate models of afterimages (Fig. 1b–e). Predictions for cone adaptation were computed with the mathematical model of Weber's law[11–16]. Using the inducer as the adapting colour produces the triangular shape of the red curve in Fig. 1b. The triangular shape results from the predicted colour being dominated by the least-adapted cone due to the multiplicative inverse effects of cone adaptation (cf. *Method* and Fig. S14). This triangular shape means that cone adaptation contradicts the wide-spread assumption that afterimages are cone-opponent. In contrast, cone-opponent afterimages result from subtractive, second-site adaptation, as illustrated by the red circle in Fig. 1c. Alternative models have been proposed to better predict several aspects of colour appearance than those physiological models. They might better match the colour appearance of afterimages, too. CIELUV, CIELAB, and CIECAM02 implement complex, nonlinear models to approximate various empirical data on colour appearance[7]. At

isoluminance (as here), CIELUV predicts the same afterimage hues as cone-opponency, but different afterimage chromas (Fig. 1c; cf. *Method*)[48,49]. CIELAB and CIECAM02 (Fig. 1d) incorporate adaptation transforms inspired by von-Kries adaptation and produce a triangular shape similar to, but differently oriented from, that of cone-adaptation. The Munsell system and Hering colours (Fig. 1e) provide yet another approach to identify complementary colours, which have been considered as points of reference for colour appearance[7,8].

*Experiments 1 and 2* tested the predictions with varying inducer hue, using two fundamentally different tasks. *Experiment 3* targeted the effect of varying inducer chroma on afterimage hue. All data and code is available through Zenodo[53]. Code for the cone-adaptation and cone-opponency models is also included and explained in sections *E.1-2* of the Supplementary Material.

### Experiment 1

The first task combined a typical afterimage display with simultaneous matching (Fig. 2a). Observers matched the hue of comparison stimuli with the hue of an afterimage that appears at the location of an inducing colour, after the inducing colour had disappeared. Chroma of comparison stimuli was kept constant (cf. *Method*).

In a first part (*Experiment 1a*), measurements with highly saturated colours showed that afterimage hues were neither cone-opponent nor Hering-opponent to the hues of the inducers (Fig. 2b). Afterimage hues for six out of eight inducers differed significantly from cone-opponent hues (Table 1; see also Figs. S2a, S3 and Table S2). In naming tasks, afterimage hues of red, yellow, green and blue were called blue, purple, purple/pink and orange/brown, in stark contrast to Hering's red-green and blue-yellow opponency (cf. Table 1; for details, see Tables S2–S3, Fig. S3). Most importantly, measured afterimage hues were close to the predictions by cone adaptation (Fig. 2i and Fig. S2), and their deviations from cone-opponent hues were almost perfectly correlated with the deviations predicted by cone adaptation (Pearson $r(6) = 0.98$, $CI_{95\%} = [0.88, 1.00]$, $p < 0.001$). These shifts in hue selection did not occur in a control task, in which observers matched real colours in an otherwise same test display, thus excluding spurious technical artifacts or response biases (Fig. S2b).

A second version of this experiment with a larger stimulus sample (*Experiment 1b*) tested the predictions of afterimage hues by the different models (Fig. 1b–d). Twenty-four inducers were sampled at equal intervals from an isoluminant, equally saturated circle in CIELUV space (Fig. 1a). If afterimages were attributable to adaption of cone-opponent mechanisms, then afterimage hues would be equally distributed around the hue circle (Fig. 1c). Instead, observers' matches of afterimage hues were systematically

### Table 1 | Differences from Opponency in Experiment 1a

| Inducer | *t* | df | CI$_{95\%}$ | *p* | *d* | Naming |
|---|---|---|---|---|---|---|
| Red | 5.35 | 30 | [7.0, 15.6] | <0.001 | 0.96 | *Blue* |
| Orange | 10.00 | 30 | [15.1, 22.9] | <0.001 | 1.80 | *Blue* |
| Yellow | 1.98 | 30 | [−0.1, 4.9] | 0.057 | 0.35 | *Purple* |
| Green | −7.15 | 30 | [−20.7, −11.5] | <0.001 | −1.28 | *Pink* |
| Turquoise | 0.63 | 30 | [−2.2, 4.1] | 0.54 | 0.11 | *Pink* |
| Blue | −8.52 | 30 | [−27.6, −16.9] | <0.001 | −1.53 | *Brown* |
| Purple | 4.69 | 30 | [6.8, 17.4] | <0.001 | 0.84 | *Green* |
| Magenta | 4.00 | 30 | [3.1, 9.5] | <0.001 | 0.72 | *Green* |

Results of t-tests across N = 31 participants comparing afterimage to cone-opponent hues are provided in the centre columns. The significance level is a = 0.00625 after correcting for running 8 tests (0.05/8). The last column indicates the colour terms corresponding to the afterimages according to the colour naming tasks; for further details, see Tables S1–S2, and Fig. S3 in the Supplementary Material.

*t* t-value, *df* degrees of freedom, *CI* two-tailed 95%-confidence interval, *p* two-tailed p-value, *d* Cohen's d.

shifted away from the cone-opponent hue directions (Fig. 2c) towards three directions that roughly coincided with the directions of maximal cone excitations (Fig. 2d). The triangular shape of the cone-adaptation prediction (red line in Fig. 1b) translates into three local peaks in the differences from opponency (dashed red line in Fig. 2c). The deviations of the measured afterimage hues from cone-opponent hues featured very similar peaks (black curve in Fig. 2c) and were thus strongly correlated with the predictions of the cone-adaptation model (Pearson $r(22) = 0.81$, $CI_{95\%} = [0.61, 0.92]$, $p < 0.001$). A hue histogram was calculated to capture the three hue clusters resulting from those shifts away from opponency. For this, responses were counted in 24 bins of 15-degree intervals across all 24 inducers. These were represented in a polar plot, where distance from the origin corresponds to relative frequency of response choices (grey area in Fig. 2d). The effect of cone adaptation on the hue histogram was simulated. Normally distributed response noise (black dots in Fig. 1b) was added to obtain a continuous probability distribution (red curve in Fig. 2d). The cone-adaptation model predicted the frequencies of hue selections with a correlation of $r(22) = 0.77$, $CI_{95\%} = [0.53, 0.90]$, $p < 0.001$, across the 24 hue bins.

## Experiment 2

The model also makes predictions about the strength, i.e., their colourfulness and chroma, of the afterimages (Fig. 1b). A chaser-like task (Fig. 2e) has been conceived to maintain the afterimage over time, allowing for adjustments of both hue and chroma to match the induced colours, without decay of afterimage strength over time. The average adjustments across 72 inducers (black curve in Fig. 2f) closely followed the predictions by the cone-adaptation model (red outline in Fig. 2f), resulting in a correlation of $r(70) = 0.90$, $CI_{95\%} = [0.85, 0.94]$, $p < 0.001$, between predicted and measured chroma. As in the other task, deviations from cone-opponent hues were strongly correlated between simulated and measured afterimages (Pearson $r(70) = 0.88$, $CI_{95\%} = [0.81, 0.92]$, $p < 0.001$, Fig. 2g), and the hue histograms for measured and modelled afterimages were strongly correlated (Pearson $r(70) = 0.82$, $CI_{95\%} = [0.73, 0.87]$, $p < 0.001$; Fig. 2h) across the 72 colours. Estimates from the correlation with smoothed data (running mean across 9 adjacent bins) suggest that the cone-adaptation model might explain around 80% of the variance in afterimage hue matches (smooth $R^2$ in Fig. 2f–h) when discounting for measurement noise. These results could also be produced at the individual level (Table 1, Figs. S8–S10): For each of 10 observers, there was a positive correlation between predicted and measured chroma ($r(70) = 0.60$ to $0.85$, all $p < 0.001$); between predicted and measured deviations from cone-opponent colours ($r(70) = 0.41$ to $0.90$, all $p < 0.001$); and between measured and simulated hue histograms ($r(70) = 0.28$ to $0.56$, all $p < 0.05$). In a second part (*Experiment 2b*), all these results have been replicated when sampling inducers in DKL instead of CIELUV space to exclude biases from stimulus sampling (Table 2, Figs. S11–12).

According to all four independent measurements (*Experiments 1.a & b, 2.a & b*), afterimage hues were closer to the cone-adaptation model than to any colour-opponent or colour-appearance model (Fig. 2i and Table 3). The only predictions that came close to the simple cone-adaptation model were those by the CIECAM02 adaptation transform (light blue symbols in Fig 2i), which approximates cone-adaptation. The measurement of illusory afterimages involves considerable degrees of measurement noise (grey symbols in Fig. 2i) and individual variation (Figs. S8–S10). Noise and individual differences set a lower boundary for all model fits. The strength and chroma of afterimages are still more difficult to measure and predict than their hues, as they depend on individuals' compliance with task instructions. Thus, measurements and the model predictions of chroma have been z-scored before subtraction to represent them on the same scale (Fig. 2j and Table 3). This allows to exclude effects of absolute differences and to focus on hue-dependent variation of chroma. Cone-adaptation provides the best model of hue-specific afterimage strength. It is striking that cone-opponent adaptation at the second stage of processing provides a particularly poor prediction of afterimage strength (red symbols in Fig. 2j).

## Experiment 3

The cone-adaptation model also predicts nonlinear changes in hue when increasing inducer chroma, which were tested in *Experiment 3* (Fig. 3a–d). Afterimages were measured for inducers at different levels of chroma. Figure 3 illustrates the extensive measurements with observer *f7*, and these results are replicated with four other observers and different stimulus samples (Fig. S13). For all observers, resulting afterimage hues differed from cone-opponent hues, and those differences were strongly correlated with predictions by cone adaptation ($r = 0.72$ to $0.85$, all $p < 0.001$, cf. Table 4 and Fig. 3e), extending the above observations (Fig. 2c, g) to measurements across chroma. The measurement and prediction of absolute afterimage strength and chroma is a particular challenge when testing the change of hue across chroma. To deal with that, the average afterimage hue across all chroma levels was calculated for each inducer hue. Each of these average afterimage hues defines a line across chroma for each corresponding inducer hue. This line is used as a reference for the variation of hue across chroma (grey lines in Fig. 3d). I calculated the difference between that average reference line and the afterimage hue measured (grey dots and black lines) and predicted (red lines in Fig. 3d) at each chroma level. For all five observers, the chroma-specific hue differences were correlated between cone-adaptation predictions and measurements (cf. Table 4, Fig. 3e and Fig. S13; $r = 0.34$ to $0.73$, $p < 0.05$), showing that afterimage hue changes with chroma in the way predicted by the cone-adaptation model.

## Discussion

The precise colours of complementary afterimages were measured with two different afterimage-inducing paradigms. All results revealed that the colours of afterimages are not aligned with cone-opponency. Instead, they can be precisely predicted by a model of cone-adaptation at the first stage of colour processing. The cone-adaptation model explains variance in hue (>60% of variance) and chroma (>80%) of afterimages (Fig. 2). Despite variation across observers, all individual data provided unequivocal evidence for the signature of cone-adaptation when accounting for individual differences in induction strength in the model (Figs. S8–S10). Hue changes of afterimages across chroma also followed the cone-adaptation model (Fig. 3). Considering measurement noise, these results suggest that complementary colours in afterimages are fully determined by cone-adaptation at the first stage of colour processing.

## Cone adaptation

The cone-adaptation model used here implements divisive adaptation according to Weber's Law, which implies that cone sensitivities decrease proportionally to the magnitude of current cone excitation. According to the classical explanation of afterimages through "cone bleaching," this model reflects the proportional decrease of photon catches by the cone photoreceptors[28,30,54]. A given light has a certain probability with which it isomerises molecules of retinal bound to opsin. When retinal molecules have already undergone isomerisation due to ongoing stimulation by an inducer, that probability applies only to the remaining proportion of retinal molecules, implying that the amount of isomerising retinal molecules is proportional to the available, non-isomerised molecules[12,16,55]. As a result, the adapted cone response is proportional to the inverse of the molecules isomerised by the inducer. The formation of afterimages can be explained through a time lag between cone isomerisation and regeneration[28,54].

Classical criticisms of the cone-bleaching explanation have been the difference between afterimages from alternating and static inducers, the duration of afterimages (>10 s) being much longer than cone adaptation («100 ms), and the low level of isomerisation at usual, moderate brightness[20,32,33]. These criticisms seem questionable. First, similar chromatic afterimages for alternating and static inducers have been found more recently[56,57], and the perception of afterimages from dynamic inducers may be affected by factors that are unrelated to receptor adaptation[26]. Second, it has been recognised that cone regeneration may take minutes rather than milliseconds[58,59]. Third, isomerisation levels correspond to the cone responses of the adapting stimulus, here the inducer. It seems contradictory

## Table 2 | Correlations of Experiments 1–2

| PPs | n | Chroma | | | | Hue Deviations | | | | Hue Histogram | | | |
|---|---|---|---|---|---|---|---|---|---|---|---|---|---|
| | | r | CI$_{95\%}$ | p | sR² | r | CI$_{95\%}$ | p | sR² | r | CI$_{95\%}$ | p | sR² |
| **Experiment 1** | | | | | | | | | | | | | |
| Exp 1a | 8 | - | - | - | - | 0.98 | [0.88, 1.00] | <0.001 | - | - | - | - | - |
| Exp 1b | 24 | - | - | - | - | 0.81 | [0.61, 0.92] | <0.001 | - | 0.77 | [0.53, 0.90] | <0.001 | - |
| **Experiment 2a (Luv)** | | | | | | | | | | | | | |
| agg | 72 | 0.90 | [0.85, 0.94] | <0.001 | 84.0% | 0.88 | [0.81, 0.92] | <0.001 | 80.5% | 0.82 | [0.73 0.87] | <0.001 | 79.8% |
| CW | 72 | 0.85 | [0.77, 0.90] | <0.001 | 78.9% | 0.90 | [0.84, 0.94] | <0.001 | 82.2% | 0.56 | [0.38, 0.70] | <0.001 | 66.5% |
| f1 | 72 | 0.62 | [0.46, 0.75] | <0.001 | 53.9% | 0.81 | [0.71, 0.87] | <0.001 | 75.9% | 0.47 | [0.27, 0.63] | <0.001 | 65.5% |
| f2 | 72 | 0.73 | [0.59, 0.82] | <0.001 | 83.3% | 0.73 | [0.60, 0.82] | <0.001 | 67.0% | 0.50 | [0.31, 0.66] | <0.001 | 61.1% |
| f3 | 72 | 0.77 | [0.66, 0.85] | <0.001 | 78.3% | 0.80 | [0.69, 0.87] | <0.001 | 77.6% | 0.58 | [0.40, 0.72] | <0.001 | 70.7% |
| f4 | 72 | 0.73 | [0.60, 0.82] | <0.001 | 74.2% | 0.73 | [0.60, 0.82] | <0.001 | 66.5% | 0.49 | [0.30, 0.65] | <0.001 | 46.5% |
| f5 | 72 | 0.64 | [0.48, 0.76] | <0.001 | 50.9% | 0.55 | [0.37, 0.70] | <0.001 | 48.9% | 0.28 | [0.06, 0.48] | 0.02 | 24.7% |
| f6 | 72 | 0.60 | [0.43, 0.73] | <0.001 | 60.8% | 0.41 | [0.20, 0.59] | <0.001 | 33.2% | 0.45 | [0.24, 0.62] | <0.001 | 49.2% |
| m2 | 72 | 0.64 | [0.48, 0.76] | <0.001 | 74.3% | 0.78 | [0.66, 0.85] | <0.001 | 71.1% | 0.44 | [0.23, 0.61] | <0.001 | 53.4% |
| m3 | 72 | 0.65 | [0.49, 0.77] | <0.001 | 57.9% | 0.69 | [0.54, 0.79] | <0.001 | 56.6% | 0.48 | [0.28, 0.64] | <0.001 | 39.5% |
| f7 | 72 | 0.63 | [0.46, 0.75] | <0.001 | 66.3% | 0.77 | [0.65, 0.85] | <0.001 | 61.3% | 0.36 | [0.14, 0.54] | 0.002 | 36.8% |
| **Experiment 2b (DKL)** | | | | | | | | | | | | | |
| agg | 72 | 0.92 | [0.87, 0.95] | <0.001 | 86.9% | 0.80 | [0.70, 0.87] | <0.001 | 65.1% | 0.62 | [0.46, 0.75] | <0.001 | 65.7% |
| CW | 72 | 0.97 | [0.94, 0.98] | <0.001 | 97.1% | 0.90 | [0.85, 0.94] | <0.001 | 81.0% | 0.55 | [0.37, 0.69] | <0.001 | 76.1% |
| f1 | 72 | 0.76 | [0.65, 0.85] | <0.001 | 66.6% | 0.69 | [0.55, 0.80] | <0.001 | 52.7% | 0.52 | [0.32, 0.67] | <0.001 | 54.1% |

PPs participants, *agg* aggregated, *CW* author, *n* number of colours, *r* Pearson's correlation coefficient, *CI$_{95\%}$* = 95% confidence interval based on Fisher's transformation; *p* p-value, *sR²* percentage variance explained by the correlation with the smoothed data (average across 9 neighbours), which is considered a rough indicator of model fit with minimised measurement noise.

**Table 3 | Model comparisons**

| Model | Experiment 1 – Hue across Participants | | | | | | Exp2 – Hue | | | Exp2 – Chroma | | |
|---|---|---|---|---|---|---|---|---|---|---|---|---|
| | Mdiff | t | df | CI$_{95\%}$ | p | d | n | S | P | n | S | p |
| | *Exp1a* | | | | | | *Exp 2a* | | | | | |
| DKL | 4.91 | 7.17 | 30 | [3.5, 6.3] | <0.001 | 1.29 | 72 | 57 | <0.001 | 72 | 64 | <0.001 |
| CIELUV | | | | | | | | | | 72 | 55 | <0.001 |
| CIELAB | 2.52 | 5.63 | 30 | [1.6, 3.4] | <0.001 | 1.01 | 72 | 51 | <0.001 | 72 | 44 | 0.08 |
| CAM02 | 1.22 | 4.62 | 30 | [0.7, 1.8] | <0.001 | 0.91 | 72 | 51 | <0.001 | 72 | 42 | 0.19 |
| Munsell | 18.01 | 23.60 | 30 | [16.5, 19.6] | <0.001 | 4.24 | 72 | 72 | <0.001 | 72 | 66 | <0.001 |
| Hering | 34.26 | 38.67 | 30 | [32.4, 36.1] | <0.001 | 6.94 | 72 | 70 | <0.001 | 72 | 64 | <0.001 |
| | *Exp 1b* | | | | | | *Exp 2b* | | | | | |
| DKL | 0.52 | 2.45 | 51 | [0.09, 0.96] | 0.02 | 0.34 | 72 | 49 | 0.001 | 72 | 67 | <0.001 |
| CIELUV | | | | | | | | | | 72 | 38 | 0.72 |
| CIELAB | 1.23 | 7.86 | 51 | [0.92, 1.54] | <0.001 | 1.09 | 72 | 52 | <0.001 | 72 | 48 | 0.006 |
| CAM02 | 0.17 | 2.44 | 51 | [0.03, 0.31] | 0.01 | 0.34 | 72 | 44 | 0.08 | 72 | 45 | 0.04 |
| Munsell | 5.29 | 10.52 | 51 | [4.28, 6.30] | <0.001 | 1.46 | 72 | 67 | <0.001 | 72 | 63 | <0.001 |
| Hering | 26.09 | 30.03 | 51 | [24.35, 27.84] | <0.001 | 4.16 | 72 | 71 | <0.001 | 72 | 55 | <0.001 |

Results of tests comparing prediction errors between the cone adaptation and other models of complementarity (cf. Fig. 2i, j). For *Experiment 1*, these comparisons were paired t-tests across the N = 31 and N = 52 participants; for *Experiment 2*, these were nonparametric sign tests across the 72 inducer colours. The first columns report results for hue predictions; here, results are the same for DKL and CIELUV. The last columns compare chroma predictions using z-scored chroma deviations across inducers.

*Mdiff* average difference, *t* t-value, *df* degrees of freedom, *CI* two-tailed confidence interval, *p* two-tailed p-value, *d* Cohen's d, *n* sample size (number of inducers), *S* S-value of sign test.

**Table 4 | Correlations of Experiment 3**

| Pps | n | Deviations from Opponency | | | Deviations from Average | | |
|---|---|---|---|---|---|---|---|
| | | r | CI$_{95\%}$ | p | r | CI$_{95\%}$ | p |
| CW | 96 | 0.85 | [0.79, 0.90] | <0.001 | 0.73 | [0.62, 0.81] | <0.001 |
| f7 | 216 | 0.74 | [0.68, 0.80] | <0.001 | 0.55 | [0.45, 0.63] | <0.001 |
| f8 | 56 | 0.79 | [0.67, 0.87] | <0.001 | 0.53 | [0.31, 0.70] | <0.001 |
| f9 | 48 | 0.72 | [0.55, 0.84] | <0.001 | 0.39 | [0.12, 0.61] | =0.006 |
| f10 | 72 | 0.75 | [0.63, 0.84] | <0.001 | 0.31 | [0.09, 0.51] | =0.008 |

*Pps* participants, *CW* author, *n* number of colours, *r* Pearson's correlation coefficient, *CI$_{95\%}$* two-tailed 95% confidence interval based on Fisher's transformation; *p* p-value.

that the level of isomerisation under usual brightness levels (1–10%) is sufficient to produce the saturated non-illusory colours of the inducers while being considered insufficient to explain the less saturated afterimages.

At the same time, it has become clear that cone adaptation at moderate light levels involves processes other than "cone bleaching."[60,61] In addition, the time course of adaptation is complicated[50,62–64] and differs across cone types[65,66]. The tasks in the present study were designed to measure afterimage colours close to maximum adaptation and to avoid temporal effects during afterimage formation and fading. The complexity of the underlying physiological processes notwithstanding, the resulting divisive cone adaptation to moderately bright, photopic colours can be approximated by cone contrasts according to Weber's law[11–14,16,55].

After cone adaptation, the adapted cone signal is propagated through the subsequent stages of colour processing[3,31,44]. The rebound signal observed in the retinal ganglion cells indicates that the cone-adapted signal is temporally maintained at the second, cone-opponent stage[20]. This temporal maintenance likely contributes to the sustained duration of afterimages[3,20,32,33]. According to the present results, processing of the colour signal at the second-stage seems not to affect the perceived colours of afterimages in any specific way.

Instead, divisive, Weber-like cone-adaptation explains why the hues of afterimages are not cone-opponent, not reciprocal, and produce three clusters of hue[25,45]. The multiplicative inverse function of inducer cone excitations produces non-linear effects of cone adaption that result in the three hue clusters and the three peaks of afterimage chroma in comparison to cone-opponency (Fig. 2d, f, h). Each one of the three afterimage peaks is determined by the least adapted and not by the most adapted cone (cf. Fig. S14): If L- and M-cones are most adapted by inducers, S cones dominate the afterimage; if L- and S-cones are adapted, M-cones dominate the afterimages; and if S- and M-cones are most adapted, L-cones dominate the afterimages. L- and M-cone sensitivities overlap. So, maximal adaptation of one always implies some adaptation of the other, and the three local maxima (peaks) of the cone-adaptation model (*red curve in* Fig. 1b) do not coincide with the directions of isolated cone excitations (*grey lines in* Fig. 1b). The overlap between L- and M-cones is stronger for broadband spectra of desaturated colours, producing a shift of hue away from isolated cone excitations with decreasing chroma (*bent black lines in* Fig. 3b). Since S-cone sensitivity overlaps comparatively little with M- and L-cone sensitivity, the effect of S-cones on L-M-adapted afterimages is much stronger than the effects of the correlated L- or M-cones, resulting in a pronounced asymmetry along the cone-opponent $S–(L + M)$ axis (close to the blue-yellow axis v*, cf. Figs. 1b, 3c, and S11.b).

**Non-opponency**

In contrast, subtractive adaptation of the cone-opponent mechanism fails to predict the three hue clusters and the asymmetry along the $S–(L + M)$ mechanism (Fig. 1c and S11.c). The failure to predict the asymmetry explains why subtractive adaptation in DKL space is comparatively

unsuccessful in predicting afterimage chroma (Fig. 2j). A cursory look at afterimages may be misleading because afterimages resemble cone-opponency at a very coarse level[34,38]. This is not due to adaptation of the cone-opponent mechanisms at the second stage of colour processing, but to the cone-adapted colour signal from the first stage being propagated to the second and subsequent stages. Afterimage nulling procedures (adjustments that cancel the afterimage) in cone-opponent space[20,39] do not allow for showing the non-opponency of afterimages, either. Successful nulling works independently of whether the appearance of afterimages follows a straight line due to subtractive cone-opponent adaptation, or a curve due to non-linear, divisive adaptation, such as cone adaptation. Although the deviations of afterimage hues from cone-opponency are subtle and require tailormade methods to be measured empirically, they are strong enough to be visible in first-person experience. This can be demonstrated using the chaser-like arrangement on a standard computer display as in the attached animated visualisations (Supplementary Movies 1–4; for instructions, see section A of the Supplementary Material).

CIELAB and CIECAM02 provide better predictions than cone-opponency because their von-Kries-like adaptation transforms produce 3 clusters close to those of cone-adaptation and capture part of the asymmetry along the S-(L + M) mechanism. Nevertheless, they differ from cone-adaptation in that CIELAB and CIECAM02 use a pseudo von-Kries transformation, applied to primaries that are similar, but not the same as the cones (see supplementary sections E.4-5 for details). This suggests that the incorporation of a proper cone-adaptation transform in colour appearance models would improve their ability to predict complementary afterimages.

Although they have been long-time favourites to explain phenomena of colour appearance[7,8], the Munsell system and Hering opponency are the worst predictors of afterimage colours (Fig. 2i, j). They predict afterimages to shift away from cone-opponency (Fig. 1d) in directions opposite to the predictions by cone-adaptation (Fig. 1b). The differences between after-images and Hering opponency are so large that the corresponding colour terms, red, yellow, green, and blue, are not even rough approximations of afterimage complementarity (Fig. S3, Table 1, S2–S3). Together, these findings refute the most widespread assumption that complementary afterimages are colour-opponent in any of the widely believed ways[2,4–8,20,34,36,38].

Others claimed that the colours of afterimages are shaped through adaptation of hypothesised cortical mechanisms that involve shape and object recognition[35,38–40,67], binocular integration[41–43] and/or colour constancy[36], contradicting classical evidence for a retinal origin[27,31]. Top-down effects on colour perception are well-known[10,68]. Such effects may be particularly strong for illusory percepts like afterimages where a physical stimulus is absent[40]. However, evidence for mid-level (e.g., contours and shape[38,40]) or high-level (e.g., knowledge[39,67]) effects on afterimages do not necessarily reflect the origin of afterimages. Afterimages occur in the absence of top-down cues or knowledge, as in the present experiments. Hence, top-down interference effects may modulate the subjective appearance of afterimages; but they are not the origin of afterimages. The cone-adaptation model leaves little variance of perceived afterimages unexplained (estimated 20% with smoothed data) that would require additional adaptation at later, subsequent levels of processing. A large part of that variance is most likely the result of noise and individual differences (see Limitations below). Speculating about yet unknown cortical mechanisms to explain the complementary colours of afterimages is thus unnecessary.

## Limitations

Despite individual differences in cone sensitivities[11,13], the present approach assumed the same cone sensitivities of a CIE standard observer for all individuals. A cone-adaptation model based on individual cone sensitivities might produce still better predictions and explain part of the individual differences observed in the afterimage measurements (Figs. S8–S10).

The validity of the cone-adaptation model notwithstanding, it does not inform us about the precise physiological processes that implement those Weber-like computations. It should also be noted that the cone adaptation

evidenced by afterimages is an important, but not the only contributor to colour adaptation and constancy under natural viewing conditions[10,69,70]. The stimulus displays in this study have been designed to specifically measure the colours of afterimages while controlling other sources of colour induction, such as local contrast induction[71], shape recognition[38], illusory contours/areas[35], or meaningful objects[39,72]. The colours perceived under adaptation to more complex stimulus configurations, including natural scenes and other real-world environments, are likely determined by a combination of different mechanisms of adaptation[3] and cognitive top-down effects[10].

## Conclusions

In sum, the present findings show that the quantitative model of cone adaptation is sufficient to explain the perceptual nature of complementary colours in afterimages. This insight resolves contradictions between the perceptual nature of afterimages and theories about their physiological origin that have been subject of a decades-old debate. Unlike what current textbooks suggest[4–6], colour afterimages are neither the result of cone- nor Hering-like colour opponency. Instead, they can be fully and coherently explained by a well-founded, straight-forward model of cone-adaptation at the first stage of colour processing.

Furthermore, the chromatic adaptation revealed by afterimages plays a fundamental role in visual perception, as it also modulates sensitivity to colour, enables colour constancy, and determines the way colours sub-jectively appear to human observers[3,6,10]. For example, divisive cone-adaptation enables us to perceive colours constantly under changes of lighting by compensating the divisive colour shifts (captured by stable cone-excitation ratios) that are typical for lighting changes in natural environments[69,70]. As illustrated by those widely used colour appearance models (Fig. 1b, c), a better understanding of the effects of adaptation on colour appearance is also key for the measurement, specification, and control of colours (colorimetry) in scientific and industrial applications[7].

## Data availability
The data used for the analyses reported in this paper are available in a Zenodo repository at https://doi.org/10.5281/zenodo.13328099[53].

## Code availability
The code for computing models and analyses is available in the Zenodo repository[53] and a linked Github repository at https://github.com/christophWit/The-Non-Opponent-Nature-of-Colour-Afterimages.

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

## Acknowledgements

Thanks to Bevil Conway, Stephen Engel, Karl Gegenfurtner, Shin'ya Nishida, Daniel Osorio, Kevin O'Regan, Joshua Solomon, Matteo Toscani, Mike Webster, Rio Coleman, and Wendy Adams for helpful comments, and Alexander Nowak and Ying Chen for help with data collection. Research was partially funded by the Deutsche Forschungsgemeinschaft (DFG, German Research Foundation)—project number 222641018 – SFB/TRR 135 TP C2, and research funds from the School of Psychology at the University of Southampton. The funders had no role in study design, data collection and analysis, decision to publish or preparation of the manuscript.

## Author contributions

C.W. conceptualised, developed and conducted all experiments, analysed data, wrote and revised manuscript.

## Competing Interests

The author declares no competing interests.
