## [Transparent Peer Review file · Communications Psychology]

The Non-Opponent Nature of Colour Afterimages

Corresponding Author: Dr Christoph Witzel

Version 0:

Decision Letter:

Dear Christoph,

Thank you for your patience during the peer-review process. Your manuscript titled "The Mechanisms Underlying Colour Afterimages" has now been seen by 3 reviewers, and I include their comments at the end of this message. They find your work of interest but raised some important points. We are interested in the possibility of publishing your study in Communications Psychology, but would like to consider your responses to these concerns and assess a revised manuscript before we make a final decision on publication.

We therefore invite you to revise and resubmit your manuscript, along with a point-by-point response to the reviewers. Please highlight all changes in the manuscript text file.

Editorially, we consider it important that you address concerns regarding the model comparison as well as presentational issues. This should lead to a manuscript that better articulates how the proposed mechanism extends, confirms, or contradicts existing models.

I am attaching an Editorial Requests Table that details critical reporting requirements for the revised manuscript. Please attend to each item and ensure your manuscript is fully compliant. If your revised manuscript is not aligned with these requests on major issues, such as those concerning statistics, it may be returned to you for further revisions without re-review.

Please submit the following items:

- Revised manuscript
- Point-by-point response to the referees' comments
- Cover letter (as a separate document)
- <https://www.nature.com/documents/nr-reporting-summary.zip>>Nature Research Reporting Summary
- <https://www.nature.com/documents/nr-editorial-policy-checklist.pdf>>Editorial Policy Checklist
- Completed Editorial Request Table (attached).

via this link: Link Redacted .

Additional guidance is available in our style and formatting guide Communications Psychology formatting guide.

Best regards,

Marike

Marike Schiffer, PhD
Chief Editor
Communications Psychology

REVIEWER EXPERTISE:

The reviewers have expertise in colour vision and psychophysics.

REVIEWER REPORTS:

Reviewer #1 (Remarks to the Author):

This study characterizes the colorimetric relationships between adapting colors and the afterimage colors they generate. The measurements are rigorous and the work would provide a valuable service in correcting persistent errors found in many textbooks about the basis for color afterimages. However, the paper itself has a number of errors and misattributions that should be addressed in a revision.

1. In considering these issues it is important to better clarify the distinction and relationship between adaptation, afterimages, and aftereffects. Specifically, if a visual mechanism adapts, there will almost always be some perceptual aftereffect corresponding to the changes in the neural response, and in the case of a steady light, these aftereffects are termed afterimages. However, not all color aftereffects are afterimages of the kind considered in the paper. Lumping them together misses the point that adaptation is happening at many levels and is manifest perceptually in different ways. Also, if receptors adapt, then they must necessarily contribute to afterimages. The paper instead seems to take the converse approach of using afterimages as a test of where adaptation is occurring, but overlooks the fact that we already know a great deal about the sites of adaptation in color processing.

2. That adaptation happens in the receptors is well established and widely accepted. The paper instead implies that this is still an open question. The specific model the author uses for cone adaptation is multiplicative scaling that occurs within the cones. This model was proposed by Von Kries more than a century ago and is widely incorporated in most color models and applications as the Von Kries transform. It has also been documented in countless studies. There is no mention in the manuscript of any of this work or the origins of this widely recognized principle. Moreover, the specific model described for adaptation is pigment bleaching, but chromatic adaptation occurs at light levels where bleaching is negligible. The paper should include a calculation of the percent of bleaching for the stimuli used to better support this claim.

3. As noted above, adaptation happens at multiple stages. This is considered in the discussion but should be addressed in the introduction when different color aftereffects are discussed. Some of these were specifically designed to test for adaptation after the receptors, but they were not intended as an alternative to receptor adaptation, as the text implies. Indeed, in early studies these effects were actually called "second-site" adaptation because they recognized that adaptation also occurs at a "first-site," in the receptors. The paper also muddles the distinction between color afterimages and aftereffects that arise from adaptation at later sites. An example is the discussion of the aftereffects in refs 19 and 30. These have been localized to cortical sites for a variety of reasons, including that they are orientation selective and show interocular transfer. These contrast or pattern-selective aftereffects are very different from chromatic adaptation in the cones, and in fact the stimuli for probing them are specifically designed to hold the state of cone adaptation constant. Arguing that these were presented as alternative to, rather than additional to, cone adaptation and conventional color afterimages is like saying that light adaptation and the tilt or motion aftereffect are just different examples of the same sensitivity changes.

4. The models used to describe the opponent predictions are not models of mechanisms but of color spaces. Consider DKL space in the current study. In the present paper, this space represents chromaticities by the linear differences in cone excitation along the L-M and L+M-S axes. For lights of equal luminance (constant L+M) only S signals vary along the L+M-S axis. Suppose the adapting stimuli form a circle of fixed radius around the nominal white chromaticity, which is the prediction evaluated in the study. Along the S axis this compares the aftereffects for a fixed linear decrement (-Sa) or increment (+Sa) in the S cones relative to the S value at the white point (Sw). However, these two stimuli are different proportional changes in the S cones – decrements are a larger percentage change. Since it is the proportional change that is relevant for multiplicative scaling and Weber's law, these two adaptors will produce afterimages of different (linear) magnitude within the space. It will also result in biases away from the complementary angle for adapting stimuli that are intermediate to the two cardinal axes, as reported. The point is that this result is because of how DKL space - and not necessarily retinal opponent mechanisms - represent color. This scaling issue is also well known and has been widely documented in empirical studies, as well as built in to standard equations for DKL space that include transforming the space for different white points. For the DKL predictions, I'm guessing that the deviations from opponency plotted in the figures could simply be replaced by showing how a multiplicative scaling would be represented in the linear space. One could also (and studies have) instead use a log DKL space. Would there then be any difference in the cone and DKL predictions?

5. Using CIELUV is even more problematic because it is a nonlinear space designed to predict color differences. I don't know of any cases where this has been used as an actual model of color mechanisms. I wonder if the 3 peaks shown in the afterimage results are again a peculiarity of the LUV space. In DKL space the changes along the L+M-S and L-M dimensions should be separable in terms of the cone signals, so I don't see how these peaks would be predicted by cone-specific adaptation represented within DKL space.

6. The final spaces comparing the Hering and Munsell predictions are a bit of a "red herring," because it is already very widely known that complementary colors are not opposite in terms of the Hering primaries. For example, many studies have noted that red and green are not complementary colors, and I believe it's already been well established that the afterimage of pure red looks like the complementary color in terms of cone excitation (blue-green) and not the pure green of the Hering opponency.

7. There is a further conceptual issue with trying to model color afterimages as sensitivity changes within an opponent mechanism. These mechanisms receive inputs of opposite sign and signal deviations from their neutral point. Adaptation "within" the opponent mechanism could make it more or less responsive, but would not shift the neutral point. The aftereffect should instead show up as a change in saturation. Altering the neutral point would instead require changing the balance of the inputs, or in other words adaptation at a pre-opponent level. An illustration of this idea is given in Webster JOV 2011 (see Fig 4). I can't see how in the present paper the effects of adaptation within – as opposed to prior to - an opponent mechanism have actually been modeled.

8. The discussion includes reference 3 for the statement that most research assumes afterimages occur at an opponent site, but (since I wrote that review) I went back and looked at what I actually said:

"Consider the fate of color afterimages, which begin with the first steps of seeing but then percolate through the system to reach our awareness. After viewing a red patch, a gray patch appears greenish. The sensitivity changes generating the aftereffect originate in the cone receptors. However, the afterimage lasts several seconds. This is too long to depend on the cones, and instead parallels the sluggish after-discharge in ganglion cells (Zaidi et al., 2012)." Webster Ann Rev 2015 This relates to points 1 and 3 above. It's important to distinguish where the sensitivity changes occur and how different stages are impacted and influence the resulting signals. In this regard the title of the present paper is misleading because there are many different mechanisms contributing to the properties of color afterimages, even if the sensitivity changes initially generating them occur at a particular site. A more appropriate title might be to replace "underlying" with "initiating." The present paper does discuss the idea that the perception of afterimages can be shaped by downstream processes, but presents it without acknowledging the many previous analyses of how adaptation effects are inherited and modified by subsequent processes.

9. Finally, it's worth noting that there is a logical flaw in the textbook accounts of color afterimages, which as the paper notes, typically argue that color afterimages are evidence of opponency, because adapting to one color produces an afterimage of the opponent color. The flaw is that a negative aftereffect is not itself evidence of opponency. For example, in the tilt aftereffect, the test pattern is also tilted in the opposite direction, but no one suggests that this is because orientation is encoded by opponent mechanisms. For color, if we assume the cones adapt, then this adaptation is just rebalancing the responses of the cones. However, the same cone responses could also be produced by varying the stimulus rather than changing the state of adaptation, so adaptation itself is irrelevant to the question of why complementary cone responses appear as complementary colors, and again does not provide any evidence for opponency.

additional points

1. To describe any stimulus as "actually colorless" is incorrect, because color is a product of the mind and what "looks" colorless depends directly on the state of adaptation.

2. The statement that S cone sensitivity barely overlaps with M and L cone sensitivity is also incorrect, since the L and M sensitivities span the entire visible spectrum.

3. I did not understand the logic of the control experiment reported in the supplement. It seems to simply be showing that observers can accurately match two colors presented side by side, which for the conditions of the study seems evident.

4. The statement that color appearance models include adaptation parameters but don't predict the afterimage colors seems surprising, since for some of these models the adaptation transform is Von Kries scaling and therefore the same as the cone-specific adaptation proposed here. It would be good to clarify why this doesn't work.

5. The paper ends by arguing that the present results demonstrating cone specific adaptation have important implications for science and industrial applications, but again ignores the fact that this model is already widely adopted in both science and industry.

Michael Webster

Reviewer #2 (Remarks to the Author):

This seems to be an excellent paper that is far outside my area of expertise.

That being said, I have played around with chromatic afterimages using an adjustment method similar to the one used here with "chaser" stimuli. NB: Using chaser stimuli was a nice idea. My only query is whether perfect (metameric) matches could be obtained without allowing for a change in brightness (i.e., in addition to the changes in hue and chroma that were allowed). Oh. Another query: Were inducers randomly interleaved and/or was the adjusted stimulus randomised at the beginning of each trial? I ask because when I was watching undergrads use the aforementioned adjustment method, they had a bias to make settings similar to previous settings.

Even more minor comments:

Please ensure the experiments are consistently numbered. The legend of Figure 2 says that Experiment 2 uses the chaser stimuli, but these are described under Experiment 1 in the Method section.

In the Method section, under Models: "Since adaptation is proportional to the intensity of the adapting stimulus, one needs proportionally more stimulation for detecting a non-adapted stimulus." "more" should be "less," right?

Given this paper's focus on quantitative modelling, I was disappointed that I couldn't find the quantitative model anywhere in this paper. Yes, there are some words, but I think the reader should be given enough information to compute the same quantitative predictions for afterimage hue and chroma given inducer hue, chroma, and brightness. I don't think all of that information appears in this paper. (It may appear in the Supplementary Material. Can't it fit in the paper itself?)

Reviewer #3 (Remarks to the Author):

This manuscript addresses a basic phenomenon- negative color afterimages- that has received surprisingly little attention from the color psychophysics community recently, and presents convincing evidence that afterimages are caused by adaptation to the cone photoreceptors. The experimental methods and basic results appear to me to be solid and important. I have a number of concerns about their presentation, however, which I detail below.

1. Introduction: Framing of the debate. I find the intensity of the debate regarding afterimages to be overstated as it mainly arises from researchers not based in the color community. This in turn leads the issue to be treated in isolation (see #2, below). But, is anyone from the color psychophysics community currently seriously arguing- based on data- that adaptation of cone signals is not at least one major cause of negative afterimages? As far as I can tell, the main empirical "hard core" evidence against the photoreceptor hypothesis are only the two Loomis papers from the 1970s, and the more recent Zaidi et al. paper. This hardly constitutes a raging debate. Sure, non-color researchers have been arguing that certain stimulus configurations may produce adaptation in post-receptoral, even cortical, mechanisms as well. And engineers designing color spaces may not be modeling adaptation correctly. But even in these two groups, I think very few would propose that receptoral adaptation plays no role at all in afterimages. While the present manuscript makes a reasonably important contribution, it could more accurately characterize the consensus in the core community.

2. Introduction: Missing background. Part of the reason some current papers have argued for exotic mechanisms being responsible for afterimages might be that they treat the issue in isolation from what is known about color vision more generally, and this manuscript repeats that mistake to some extent. The paper would greatly benefit from presenting the "standard" process model of color vision at the beginning. Many, and I would venture most, "back pocket" process models of color vision (vs models used to form color spaces) assume that photoreceptor signals adapt somewhere along the visual pathways, and that this adaptation has behavioral consequences; i.e. Von Kries or "generalized Von Kries" adaptation has received much psychophysical and physiological support. This should be noted up front in the paper. This manuscript, then, tests, and fails to support, the specific hypothesis that the appearance of color afterimages is dependent upon different mechanisms, that somehow bypass photoreceptor adaptation. For example, photoreceptor adaptation might fade too quickly to be relevant for relatively long-lasting color afterimages. Similarly, the discussion should emphasize more that the photoreceptor adaptation that is in fact the basis for color afterimages has lots of support from different behavioral and neural paradigms besides measures of afterimages.

3. Introduction and discussion: Later mechanisms. Similarly, the standard process model also proposes that "second site" adaptation occurs, depending upon induction procedures, for in example transient tritanopia and contrast adaptation. It would seem likely that these have effects upon afterimages, and in the discussion the authors acknowledge that later

adaptive processes that adapt can have effects on afterimage appearance. So, I would suggest clearly stating up front the hypothesis this paper is testing in more detail: That photoreceptor adaptation is necessary to produce color afterimages, and that adaptation of cone-opponent mechanisms is not necessary (but not that photoreceptor adaptation is sufficient to explain their appearance). This or similar phrasing might be helpful throughout the manuscript, including the discussion, which seems to suggest that second site adaptation does not affect afterimage formation. That may be true for the present data, but many other stimulus configurations, including more natural ones, could well produce both first- and second-stage adaptation.

4. More details of models needed. The results were unnecessarily difficult to follow, even for this somewhat educated reader, and could be unintelligible to the general audience. Most importantly, it would greatly help to formally spell out in the results section exactly what the cone-opponency model is- preferably in equations. It's weird to me how hard this is, so I must be missing something obvious, but I am guessing that the cone-opponent model includes a mechanism that compares L-M to M-L (with each half-wave rectified?) and that adaptation subtracts the activity produced by the adapter in the activated sub-mechanism? Similarly, the paper should spell out more formally what the Hering-opponency model and cone-adaptation models are. I also imagine a general audience would deserve definitions of the color spaces and color terms like hue and chroma up front, as well as some experimental details like the number of observers.

5. Discussion: More discussion of the mechanisms of cone adaptation needed. Zaidi et al. suggest that physiologically measured cone adaptation recovers too rapidly to account for afterimages. That paper also distinguishes between light levels where adaptation occurs due to photoreceptor "bleaching" and levels where adaptation must have some other mechanism- presumably some sort of early gain control. The discussion of the current paper seems to adopt the bleaching model- that simply the isomerization of retinal can explain the adaptation producing color afterimages. But based on the Zaidi paper I would ask, but does that model match the slow timecourses seen here and can it explain afterimage presence at relatively low light levels?

6. Minor: Correlation coefficients are not accurate (or at least their corresponding p-values are not) for non-independent samples- this is definitely the case for the smoothed data, and could be the case for the nonsmoothed data (e.g. if multiple points are from the same adapter). A more complicated statistic may be needed. Smoothing is not a standard English word.

Signed- Steve Engel

* TRANSPARENT PEER REVIEW: Communications Psychology uses a transparent peer review system. This means that we publish the editorial decision letters including Reviewers' comments to the authors and the author rebuttal letters online as a supplementary peer review file. However, on author request, confidential information and data can be removed from the published reviewer reports and rebuttal letters prior to publication. If your manuscript has been previously reviewed at another journal, those Reviewers' comments would not form part of the published peer review file.

Version 1:

Decision Letter:

Dear Christoph,

Your manuscript titled "The Non-Opponent Nature of Colour Afterimages" has now been seen by our reviewers, whose comments appear below. In light of their advice I am delighted to say that we are happy, in principle, to publish a suitably revised version in Communications Psychology.

We therefore invite you to revise your paper one last time to address the remaining concerns of our reviewers and a list of editorial requests. At the same time we ask that you edit your manuscript to comply with our format requirements and to maximise the accessibility and therefore the impact of your work.

EDITORIAL REQUESTS:

SUBMISSION INFORMATION:

OPEN ACCESS:

* DATA AVAILABILITY:

Link Redacted

Best wishes,

Marike

Marike Schiffer, PhD
Chief Editor
Communications Psychology

REVIEWERS' COMMENTS:

Reviewer #1 (Remarks to the Author):

The revision has addressed all of my concerns and I recommend publication. Mike Webster

Reviewer #2 (Remarks to the Author):

The authors have addressed all of my concerns.

Reviewer #3 (Remarks to the Author):

The authors have done an excellent job correcting the sources of my previous concerns. This remains interesting and important work.

I would just suggest two small changes to aid readability: In the second paragraph of the manuscript I would suggest altering "adaptation may occur...", which leaves room for doubt, to "evidence for adaptation has been found at many points along this visual hierarchy."

I would also suggest a small change to the second paragraph of the results section, bringing in one sentence or phrase from the discussion about >why< cone adaptation produces the triangular prediction, something like "...with appearance of the afterimage showing smallest contribution from the most-adapted and greatest from the least-adapted cones..."

Reply to Reviewers

GENERAL RESPONSE / RESPONSE TO EDITORIAL LETTER

Editorially, we consider it important that you address concerns regarding the model comparison as well as presentational issues. This should lead to a manuscript that better articulates how the proposed mechanism extends, confirms, or contradicts existing models.

RESPONSE

This study resolved fundamental contradictions about the neural origin and phenomenological nature of colour afterimages, which have prevented a coherent explanation of this important phenomenon. These findings have important implications for our understanding of colour perception, perceptual adaptation, and the link between neural mechanisms and the phenomenology of perception.

From reviews 1 and 3 from world-leading specialists on adaptation, I took that my original Introduction and Discussion were unsatisfactory to experts on adaptation. In particular, the manuscript miscommunicated the contribution of the study by shifting the reader's focus to the mechanisms of adaptation in general, which may lead the reader to expect a discussion about intricacies of the physiological details of adaptation and a contribution of the study to understand more details about them. This is indeed not what my study is about. Instead, this study is about the powerful link between well-established knowledge about the mechanisms of adaptation and the perceptual nature of afterimages. The important implications of this link have been ignored and not surprisingly, the state-of-the-art about this link is completely confused and contradictory. The important contribution of this study is the almost perfect fit of very substantial data from different paradigms with a simple, straight-forward model of adaptation that is widely accepted, at least as a good approximation under typical viewing conditions, like the ones in this study. This finding provides a coherent framework to understand colour afterimages in a way that is so accessible that non-specialists (textbook authors, practitioners in art and industry) will find it useful.

My strategy for this revision has thus been to acknowledge the intricacies of physiological details of adaptation, avoid the inherent conceptual pitfalls highlighted by the reviewers, and yet to focus on the key contribution of the study beyond those details. I hope the resulting very thoroughly revised version of the manuscript is an optimal compromise between satisfying specialist scrutiny, while highlighting the simple explanatory power of the study beyond physiological details.

Here are some highlights of my revision with respect to the editorial comments:

1. Presentational issues

(a) Clarification of the Research Question in the Introduction:

Even more importantly, the link between neural mechanisms and the perceived colours of afterimages has been ignored or misconstrued. Irrespective of the hypothesised neural origin, most state-of-the-art research assumes that the perceived colours of afterimages are cone-opponent,^{20,34,36,38,44} but there is evidence against that assumption.^{25,45} As a result of this confusion, textbooks, reviews, and other broader communications either do not attempt any explanation of colour afterimages^{2,4} or disseminate misleading narratives, such as the idea that afterimages reflect Hering-opponency between red-green and blue-yellow.^{6,8} (p.1)

(b) Framing in the Introduction:

Human colour perception is the result of several stages of processing, starting with the excitation of the cone photoreceptors in the eyes, propagating through cone-opponent mechanisms in the retinal ganglion cells, the LGN of the thalamus, and the double-and single-opponent cells in the primary visual cortex, until the colour signal reaches higher cortical areas that produce the subjective experience of colour.¹⁰ Adaptation may occur at any point along this visual hierarchy of colour processing. (p.1)

(c) Title

To avoid misguided expectations towards the physiological details about mechanisms of adaptation, I suggest changing the title from “The mechanisms underlying colour afterimages” to “The non-opponent nature of colour afterimages.” The new title has the benefit that “nature” encompasses both, the perceptual nature and the neural origin, and it conveys information about the key finding of the study.

(d) Discussion: The discussion has been very thoroughly revised, too. Oversimplifications have been rectified, reviewer comments integrated, and the reasoning has also been made much clearer.

2. Model Comparisons

(a) Details on Models: I thoroughly revised the *Models* subsection of the Method, adding key information requested by reviewers. I also added more detailed explanations with mathematical proofs and derivations and Matlab Codes to section *D. Models* of the *Supplementary Material*.

(b) Explanation of Predictions. I improved the explanation of predictions derived from the models at the beginning of the Results section.

(c) Visualisation of Predictions. I improved the visualisations of the model predictions in Figure 1.

3. Extension, Confirmation, or Contradiction of Existing Models

I hope it became clear that this study is not about extending, confirming, or contradicting existing models, but about applying existing models that have been established and used in a broader context, to make specific predictions about the phenomenological, perceptual nature of afterimages and test those predictions with substantial data. The result identifies a specific model as pertinent, in fact, almost completely explanatory, and contradicts widespread assumptions about the phenomenological nature of afterimages (namely that they are colour-opponent). Apart from the clarifications of the research question in the Introduction, the first point is clearly articulated in the Discussion:

Considering measurement noise, our results suggest that **complementary colours in afterimages** are fully determined by cone-adaptation at the first stage of colour processing. (p.7)

The latter point is clearly said here in the Discussion:

Together, these findings debunk the most widespread misconception that complementary afterimages are colour-opponent in any of the widely believed ways.^{2,4,8,20,34,36,38} (p.8)

REVIEWER #1 (REMARKS TO THE AUTHOR):

This study characterizes the colorimetric relationships between adapting colors and the afterimage colors they generate. The measurements are rigorous and the work would provide a valuable service in correcting persistent errors found in many textbooks about the basis for

color afterimages. However, the paper itself has a number of errors and misattributions that should be addressed in a revision.

1. Adaptation, Afterimages, and Aftereffects

In considering these issues it is important to better clarify the distinction and relationship between adaptation, afterimages, and aftereffects. Specifically, if a visual mechanism adapts, there will almost always be some perceptual aftereffect corresponding to the changes in the neural response, and in the case of a steady light, these aftereffects are termed afterimages. However, not all color aftereffects are afterimages of the kind considered in the paper. Lumping them together misses the point that adaptation is happening at many levels and is manifest perceptually in different ways. Also, if receptors adapt, then they must necessarily contribute to afterimages. The paper instead seems to take the converse approach of using afterimages as a test of where adaptation is occurring, but overlooks the fact that we already know a great deal about the sites of adaptation in color processing.

RESPONSE

I understand this comment so that I should clarify that the focus is on explaining afterimages, starting from the knowledge we have about different types and sites of adaptation. I fundamentally revised the Introduction in response.

(1.a) Candidate sites of adaptation independent of afterimages.

The revised Introduction now starts with a brief overview of the candidate sites of adaptation before discussing how those have been associated with afterimages:

Although, candidate mechanisms of adaptation have been known for more than a century,⁹ state-of-the-art explanations of this important phenomenon remain contradictory and confused. [...]

Adaptation of photoreceptors at the first stage of colour processing produces divisive adaptation, similar to von Kries' original proposal,⁹ and can be approximated as contrast coding following Weber's law.^{11,12} A slower type of "second-site adaptation" occurs in the retinal ganglion cells,¹⁷ and is assumed to be subtractive.^{11,18-20} Other observations suggested "higher-order" adaptation of yet unknown, cortical mechanisms.^{3,21-24} [p.1]

This study is not about aftereffects in general and the manuscript had never mentioned the term aftereffects to avoid losing focus, except – confusingly – for the keywords, which I changed, and two unfortunate citations, which I address below (comment 3).

(1.b) The link between neural origin and perceptual nature of chromatic afterimages

I also better focus the literature review of the Introduction on the research question about afterimages, and most notably the missing link between the neural origin and their phenomenological characteristics. The below citations follows the one cited above (a):

Researchers have disagreed about the neural mechanisms of adaptation that cause complementary afterimages.^{6,20,25,26} Classical studies suggested that such afterimages are caused by photoreceptor desensitisation ("cone bleaching") at the first stage of processing.²⁷⁻³¹ Later studies firmly contradicted this idea,^{20,32,33} attributed afterimages to adaptation in the cone-opponent channels at the second stage,^{13,24,26} or emphasised the effects of cortical adaptation.^{33,35,43}

Even more importantly, the link between neural mechanisms and the perceived colours of afterimages has been ignored or misconstrued. Irrespective of the hypothesised neural origin, most state-of-the-art research assumes that the perceived colours of afterimages are cone-opponent,^{20,34,36,38,44} but there is evidence against that assumption.^{25,45} As a result of this confusion, textbooks, reviews, and other broader communications either do not attempt any

explanation of colour afterimages^{2,4} or disseminate misleading narratives, such as the idea that afterimages reflect Hering-opponency between red-green and blue-yellow.^{6,8} [p.1]

(1.c) Confusion about the origin and nature of chromatic afterimages

I am not quite sure whether the comment is meant to question the literature review on afterimages per se. So, I also dug out a couple of examples to illustrate the confusing situation of the state-of-the-art. Zaidi et al. (2012) write:

“Proposed physiological mechanisms for color afterimages range from bleaching of cone photopigments to cortical adaptation [4-9], but direct neural measurements have not been reported.”

Koenderink et al. (2020) conclude:

Indeed, it cannot be said that the mechanism of negative afterimages is fully understood (Anstis, 2017). (p. 12)

Second, recent arguments against cone adaptation have been very strong and had impact on specialist research. As acknowledged in the comment, Zaidi et al. (2012) firmly argued against cone adaptation:

“This result should correct the notion, found on the web and in many textbooks, that photoreceptor desensitization is responsible for color after-images generated by normal light levels.” (p.222)

Kingdom et al. (2020) took this idea from Zaidi et al. (2012), although it does not easily align with their nonlinear results:

“Negative, or complementary afterimages are experienced following brief adaptation to chromatic or achromatic stimuli, and are believed to be formed in the post-receptoral layers of the retinae.” (from the abstract)

“In relation to the present study, in which TvC functions were measured, the RGC explanation for afterimages provided by Zaidi et al. needs to be squared with the psychophysical evidence for separability of the poles of the cardinal axes in the context of stimulus detection [...]” (p.32)

Others claim that afterimages are neither the result of receptor adaptation nor cone-opponency but of cortical adaptation, for example Zeki et al. (2017):

Hence, traditional accounts of after-images as being the result of retinal adaptation or the perceptual result of physiological opponency, are inadequate. We propose instead that the color of after-images is generated after colors themselves are generated in the visual brain. (from the abstract)

Shimojo et al. (2001) claim in their Conclusion:

“Our findings are consistent with previous studies suggesting that afterimages produced by long exposures to moderate-intensity patterns are not due to photopigment bleaching but rather to neural adaptation (26-28), and that the appearance of the afterimage depends on the perceptual, not physical, attribute of the adapting stimulus (26, 29, 30).” (p.4)

Similarly, Phuangsuwan et al. (2018):

The relation relationship of $\Delta\theta$ to the angle of the adapting color θ_{ing} was quite similar to the results obtained by the two-room technique, implying that the chromatic adaptation shown by the afterimage also occurs in the brain rather than in the retina. (from the abstract)

Others claim that everything is somehow involved in the formation of afterimages, at least under some conditions. For example, Dong et al. (2017) say in their abstract:

“Our results thus contradict the retinal generation notion, and suggest that in addition to the retina, cortex is directly involved in the generation of AI signals.” (from the abstract, but see also Introduction & Conclusion)

Or Van Lier et al. (2009):

“Our results with afterimages indicate that cortical color filling-in processes are also involved when incoming signals are caused by adaptation of retinal receptors.” (p.323)

There is a lot left open about what exactly those claims might mean about the origins of the observed effects, and I have had a little discussion about that in the Discussion session of the manuscript.

As you know, I also polled attendees of the colour symposium in Gießen, and results showed that one quarter believed afterimages are cone-opponent, about one half that they are not, and another quarter responded they don't know:

So, even experts and PhD students in the field do not agree on the nature of afterimages, and their explanation thus remains guesswork, open to speculations and selective reading of the literature.

The contradictory and confusing state of the art has had important effects on textbooks and reviews. Explanations in our most important textbooks are fragmented or missing, leaving students in vision science confused. After claiming that afterimages are a signature of Hering colours in earlier editions (Goldstein & Brockmole, 2017), the newest edition of Goldstein's Sensation and Perception (Goldstein & Cacciamani, 2021) has completely given up trying to explain afterimages. Although, Conway et al's (2023) TICS review "against" Hering colour-opponency would have greatly benefited from a firm argument about the nature and origin of afterimages, they refrain from any explanation and limit themselves to a loose statement about the colour of afterimages:

„The afterimage (and complement) of red is bluish-green (not green).” (p.8)

Wolfe's Sensation and perception correctly (according to the present perspective) explains afterimages through receptor adaptation but claims at the same time that afterimages are cone-opponent. This does not fit together, as I show in this study. Reviews and textbooks outside the field of colour science are all over the place. Many suggest that afterimages follow

Hering-opponency, as the philosopher Manzotti (2017) correctly observes (see also his table 5):

Remarkably, scientists and philosophers have provided biased and imprecise reports based on color opponency (Byrne & Hilbert, 2003; Churchland, 2005; Goldstein, 2010; Hurvich, 1981; Jones, 1972; Lycan, 2002; Mach, 1897; Macpherson & Platchias, 2013; Palmer, 1999; Schwitzgebel, 2011; Werner & Bieber, 1997), and only a minority of accounts have considered the complementary nature of afterimages (Bidwell, 1897; Geisler, 1978; Livingstone, 2002; Livitz, Yazdanbakhsh, Eskew, & Mingolla, 2011; Pridmore, 2008; Wilson & Brocklebank, 1955; Zaidi et al., 2012). Finally, the notion of complementary color varies from author to author (Anstis, Vergeer, & van Lier, 2012; Goldstein, 2010; Hurvich, 1981; Livingstone & Hubel, 1987; Zaidi et al., 2012), and it is occasionally confused with that of opponent colors (Livingstone, 2002; Tsuchiya & Koch, 2005).

2. Foundational Literature on cone adaptation

That adaptation happens in the receptors is well established and widely accepted. The paper instead implies that this is still an open question. The specific model the author uses for cone adaptation is multiplicative scaling that occurs within the cones. This model was proposed by Von Kries more than a century ago and is widely incorporated in most color models and applications as the Von Kries transform. It has also been documented in countless studies. There is no mention in the manuscript of any of this work or the origins of this widely recognized principle. Moreover, the specific model described for adaptation is pigment bleaching, but chromatic adaptation occurs at light levels where bleaching is negligible. The paper should include a calculation of the percent of bleaching for the stimuli used to better support this claim.

RESPONSE

I am sorry for the confusion. I did not mean to say that adaptation in the photoreceptors per se is an open question, but whether colour afterimages are the result of that type of adaptation. As explicit in this comment, many have doubted cone adaptation is relevant at usual, moderate brightness levels. Thus, evidence for cone adaptation in afterimages at this brightness level also make a broader contribution to that question, but this is not the primary concern of the study. I hope I clarified the focus with the above revision of the Introduction (1.a-b), and I hope that the above quotes (1.c) provide convincing evidence that many experts in the field have indeed doubted that cone adaptation is the source of chromatic afterimages under usual levels of brightness, as also implied by the end of this comment. I thus focus here on the claim that (a) I did not cite any references about von Kries and/or multiplicative scaling (often also called divisive scaling), and (b) the calculation of the percent of bleaching for the stimuli.

(2.a) Additional References:

I agree that there are “countless studies” that are tangentially related to the topic of this study in that they concern adaptation of cone photoreceptors in general. Especially a large part of the classical literature is about rod or cone adaptation to achromatic rather than chromatic stimuli, which seems still further away from the focus of this study. In contrast to those very general references, the focus of the present study is the nature and origin of chromatic afterimages. Previous research directly relevant to this focus is much more “countable,” and I think I rather exhaustively represented the contradictions in that literature in the manuscript, especially considering that this is not a scoping review.

I agree with the reviewer that I also need to refer to the literature on cone adaptation more generally. However, I feel I cannot cite “countless studies” that are not about afterimages and must instead select a few particularly representative and pertinent references on chromatic adaptation. I had done so. I had already cited five foundational studies that link afterimages to the retinal origin and cone adaptation (Brindley, 1962; Craik, 1940; Hurvich & Jameson, 1957;

Pöppel, 1986; Williams & MacLeod, 1979) and nine references about the Von Kries / Weber-like cone-contrast model (Brainard, 1996; Brainard & Stockman, 2010; Burkhardt, 1994; Burkhardt & Gottesman, 1987; Pugh & Lamb, 2000; Pugh et al., 1999; Shapley & Enroth-Cugell, 1984; Stockman & Brainard, 2010; Whittle, 1986). The reason why I had chosen these references as representative is because they explicitly refer to Weber's law and Weber contrast, which seems more complete than pure von Kries multiplicative scaling and is the exact model used in this study. I am thus confused by the claim that there was "no mention" of the "countless studies" in the original version of the manuscript. So, I thought might have missed classical references that are dear to the reviewer. I checked the reviewer's publications on visual adaptation (Webster, 2011, 2016) to identify those references to von-Kries adaptation but was unable find them. If the reviewer would like me to cite the original reference to von Kries, I have done so now with *reference 9*:

Although, candidate mechanisms of adaptation have been known for more than a century,⁹ state-of-the-art explanations of this important phenomenon remain contradictory and confused. [p.1]

And here:

Adaptation of photoreceptors at the first stage of colour processing produces divisive adaptation, similar to von Kries original proposal,⁹ and can be approximated as contrast coding following Weber's law.¹¹⁻¹⁶ [p.1]

Maybe the criticism came up because the original manuscript features the above references came too late, namely in the Method rather than the Introduction. They now correspond to references 11-16 in the above quote and are thus at the heart of the Introduction of the revised manuscript.

In a further attempt to better represent classical studies, I have also added Pugh and Mollon (1979) in ref 17 and Stromeyer (1985) in ref 18, which introduce the idea of second-site adaptation, and Krauskopf (1986) in ref 22 about higher-order adaptation:

A slower type of "second-site adaptation" occurs in the retinal ganglion cells,¹⁷ and is assumed to be subtractive.^{11,18,20} Other observations suggested "higher-order" adaptation of yet unknown, cortical mechanisms.^{3,21-24} [p.1]

I feel this is a much better way to review the literature and hope the reviewer agrees with this approach.

(2.b) Percent bleaching of the stimuli

I am happy to compute percentages of pigment isomerisation. I obtain 1.7% for Experiment 1 and 3.5% for Experiment 2 (depending on the set-up) using the Rushton and Henry (Rushton & Henry, 1968) half-bleach constant of 4.3 log₁₀ trolands (20,000 td), and estimating irradiance based on stimulus luminance (41.2 cd/m² and 101.8cd/m²). I have added this information to the Method section of the manuscript, for Experiment 1:

This brightness corresponds to 1.7% cone bleaching according to the Rushton and Henry half-bleach constant.⁵⁹

And for Experiment 2:

The brightness corresponded to about 3-3.5% cone bleaching according to the Rushton and Henry half-bleach constant.⁵⁹

However, I am not sure this helps understanding the phenomenon of afterimages. Following the logic of photoreceptor adaptation and the Weber model, the order of magnitude of bleaching is equivalent to the bleaching produced by the inducer colour. This means it is about

the same amount of isomerisation that makes the inducer having a very saturated real (non-illusory) colour. If this is so, it seems plausible that this level of bleaching is sufficient to produce the slightly weaker, illusory colour of an afterimage. I also note that the most important evidence for two types of adaptation depending on brightness levels comes from achromatic contrast adaptation (Baccus & Meister, 2002; Dunn et al., 2007; Hayhoe et al., 1987; Stockman et al., 2006), which is not the same as chromatic afterimages. Just consider how the small difference between M- and L-cone activation produces a whole L-M colour dimension. If one compares the size of that difference with the absolute size of the cone excitations, one will likely miss the importance of the L-M dimension for colour vision. Comparisons of absolute intensities may neither be an adequate approach to understand colour perception in general, nor afterimages in particular. I added a short discussion of the classical criticism about speed and intensity to the Discussion:

Classical criticisms of the cone-bleaching explanation have been the difference between afterimages from alternating and static inducers, the duration of afterimages (>10s) being much longer than cone adaptation (<<100ms), and the low level of isomerisation at usual brightness levels.^{20,32,33} These criticisms seem questionable. First, similar chromatic afterimages for alternating and static inducers have been found more recently,^{52,53} and the perception of afterimages from dynamic inducers may be affected by factors that are unrelated to receptor adaptation.²⁶ Second, it has been recognised that cone regeneration may take minutes rather than milliseconds.^{54,55} Third, isomerisation levels correspond to the cone responses to the adapting stimulus, here the inducer. It seems contradictory that the level of isomerisation under usual brightness levels (1-10%) is sufficient to produce the saturated non-illusory colours of the inducers while being considered insufficient to explain the less saturated afterimages. (p.7)

I nevertheless agree with the reviewer that it is important to consider that “cone-bleaching” is not the only source of Weber-like, divisive adaptation and that even the evidence against cone-bleaching does not undermine the results in support of divisive adaptation found in the present study. I added this to the Discussion:

At the same time, it has become clear that cone adaptation at moderate light levels involves processes other than “cone bleaching.”^{56,57} In addition, the time course of adaptation is complicated⁵⁸⁻⁶¹ and differs across cone types.^{62,63} The tasks in the present study were designed to measure afterimage colours close to maximum adaptation and to avoid temporal effects during afterimage formation and fading. The complexity of the underlying physiological processes notwithstanding, the resulting divisive cone adaptation to moderately bright, photopic colours can be approximated by cone contrasts according to Weber’s law.^{11-14,16,51} (p.7)

3. Different Levels of Adaptation

As noted above, adaptation happens at multiple stages. This is considered in the discussion but should be addressed in the introduction when different color aftereffects are discussed. Some of these were specifically designed to test for adaptation after the receptors, but they were not intended as an alternative to receptor adaptation, as the text implies. Indeed, in early studies these effects were actually called “second-site” adaptation because they recognized that adaptation also occurs at a “first-site,” in the receptors. The paper also muddles the distinction between color afterimages and aftereffects that arise from adaptation at later sites. An example is the discussion of the aftereffects in refs 19 and 30. These have been localized to cortical sites for a variety of reasons, including that they are orientation selective and show interocular transfer. These contrast or pattern-selective aftereffects are very different from chromatic adaptation in the cones, and in fact the stimuli for probing them are specifically designed to hold the state of cone adaptation constant. Arguing that these were presented as alternative to, rather than additional to, cone adaptation and conventional color afterimages is like saying that light adaptation and the tilt or motion aftereffect are just different examples of the same sensitivity changes.

RESPONSE**(3.a) Adaptation at multiple stages:**

I believe my thorough revision of the Introduction now better represents our knowledge of different mechanisms of adaptation in general (see my response to point 1.a on candidate sites of adaptation). I also hope the revised Introduction now makes it clearer that this study was not about whether some kind of adaptation happens at some point during the processing of afterimages, but about the nature and origin of afterimages, i.e., the mechanisms that generate afterimages and determine how they look (see response to point 1.b). More specifically, the Introduction now also introduces the term “second-site adaptation”:

A slower type of “second-site adaptation” occurs in the retinal ganglion cells,¹⁷ and is assumed to be subtractive.^{11,18-20} [p.1]

(3.b) Rectification of References 19 and 30

References 19 and 30 were Webster and Mollon (1991) and He and MacLeod (2001), respectively. In the original manuscript, they were cited in the Introduction:

Many studies assume that afterimages reflect cone-opponent mechanisms,^{2,6,13,18} but many others provided conflicting evidence.^{3,19-21} The alternative idea that complementary afterimages follow Hering-opponency²² is hotly debated.^{6,10} Most recent studies question the retinal origin of afterimages and instead attribute afterimages to yet unknown cortical mechanisms.^{19,23-30}

And Discussion, here:

Most state-of-the-art research assumes that afterimages correspond to the colour predicted by cone-opponent mechanisms at the second stage of colour processing,^{2,3,6,13,22} but again others found contradictory results.¹⁹⁻²¹

And here:

Speculating about yet unknown cortical mechanisms to explain deviations from cone-opponency is thus unnecessary.^{19,30}

Webster and Mollon (1991) is now *reference 21*. Its citation in the revised manuscript does not claim anymore that the study involved afterimages. Here is the citation in the Introduction:

Other observations suggested “higher-order” adaptation of yet unknown, cortical mechanisms.^{3,21-24} [p.1]

I removed the citation of He and MacLeod (2001) to avoid confusion between colour (here) and tilt aftereffect (in He & MacLeod, 2001) as in comment 9.

4. DKL Space and Cone-Opponent Mechanisms

The models used to describe the opponent predictions are not models of mechanisms but of color spaces. Consider DKL space in the current study. In the present paper, this space represents chromaticities by the linear differences in cone excitation along the L-M and L+M-S axes. For lights of equal luminance (constant L+M) only S signals vary along the L+M-S axis. Suppose the adapting stimuli form a circle of fixed radius around the nominal white chromaticity, which is the prediction evaluated in the study. Along the S axis this compares the aftereffects for a fixed linear decrement (-Sa) or increment (+Sa) in the S cones relative to the S value at the white point (Sw). However, these two stimuli are different proportional changes in the S cones – decrements are a larger percentage change. Since it is the proportional change that is relevant for multiplicative scaling and Weber’s law, these two adaptors will produce afterimages of different (linear) magnitude within the space. It will also result in biases away from the complementary angle for adapting stimuli that are intermediate to the two cardinal axes, as reported. The point is that this result is because of how DKL space - and not necessarily retinal opponent mechanisms - represent color. This scaling issue is also well known and has been widely documented

in empirical studies, as well as built in to standard equations for DKL space that include transforming the space for different white points. For the DKL predictions, I'm guessing that the deviations from opponency plotted in the figures could simply be replaced by showing how a multiplicative scaling would be represented in the linear space. One could also (and studies have) instead use a log DKL space. Would there then be any difference in the cone and DKL predictions?

RESPONSE

(4.a) The effect of first-site on second-site adaptation

I agree with the phrasings that cone-adaptation is “inherited by” or “percolates through” the second-stage and later mechanisms. This is exactly what happens when modelling cone-adaptation through Weber-like cone-contrasts (or von Kries adaptation, if you prefer) and represent them in cone-opponent or colour appearance models in this study. I am confused that the comment seems to claim that this is second-stage adaptation and asks whether there is a difference between “cone and DKL predictions.” Of course there is a difference, that is one of the key points of this study. To understand this, we should not mix up the input to the mechanisms and the adaptation of the mechanisms themselves. We need to consider what happens in the mechanisms when the cone input to the cone-opponent mechanisms (and the axes of DKL space) are NOT adapted to the inducer but are only adapted to the overall background grey. In that case, there is no multiplicative inverse because the background grey is in the denominator of the input, and the processes explained in the comment above do not happen. In addition, while cone adaptation involves a multiplicative inverse (the adaptation transform), second-stage adaptation has been assumed to be subtractive. I hope this is now better explained in the revised Introduction (see responses to comment 1). I added an explanation and the formula to the Method, which now comes before the Results in line with journal guidelines:

Second-stage adaptation. Cone-opponency has been modelled with DKL colour space.(Derrington et al., 1984) Cone-opponent adaptation is assumed to be subtractive.^{11,19} So, it was calculated by subtracting the cone-opponent signal of the adapting colour from the cone-opponent signal of the achromatic probe.²⁰

(2) Cone-opponent adaptation to inducer:

$$DKL_{L-M} = CC_M - CC_L - k * iDKL_{L-M}$$

$$DKL_S = CC_S - k * iDKL_S$$

Where CC is the cone contrast calculated with equation (1), and the index indicates the respective cone (details on the cone-opponent axes are provided in section D-2 of the supplementary material). $iDKL_{LM}$ and $iDKL_S$ correspond to the cone-opponent signal of the inducer. Cone-opponent adaptation produces proportional shifts along the axes, implying cone-opponent afterimage hues, 180 degrees rotated away from the inducer and afterimage chroma that is k times the inducer chroma. Here, k was set to match the estimated strength of the afterimages, but the value of k was irrelevant to the results.

In addition, I revised the explanation of the predictions in the Result section accordingly to make this clearer:

Predicted hue and chroma of afterimages differ across models. Predictions for cone adaptation are computed with the mathematical model of Weber's law.¹¹⁻¹⁶ Using the inducer as the adapting colours produces the triangular shape of the red curve in Figure 1.b. This means that cone adaptation does agree with the wide-spread assumption that afterimages are cone-opponent. In contrast, cone-opponent afterimages result from subtractive, second-site adaptation, as illustrated by the red circle in Figure 1.c. Cone-opponency has been modelled with DKL colour space.⁴⁶

(4.b) Models and Spaces

Comment 4 also criticises that my models are not models of mechanisms, but colour spaces. That is certainly true for the colour appearance models, but not for the models of cone-adaptation and cone-opponency. That computations can be represented in a colour space does not contradict the idea that these computations mimic physiological mechanisms. Both, cone adaptation and DKL space have been formulated based on physiological data (Derrington et al., 1984). In addition, I had checked that it does not make a difference when the six opponent mechanisms are modelled separately (often denoted as “half-wave rectification”) instead of the three axis that combine the pairs of mechanisms. I added a note on this to the model descriptions:

Although, a theoretically complete model would differ between the two separate mechanisms that constitute each opponent channel (“half-wave rectification”), such a separation did not change the predictions in this context.

(4.c) Scaling of DKL space

Some part of the above comment seems to suggest that the non-opponent afterimages might be a property of scaling the axes of DKL-space. However, the 3 hue clusters are not a characteristic of DKL-space or its scaling. They appear no matter how DKL is scaled and when spaces other than DKL are used, such as CIELUV or XYZ or xyY. Linear axis scaling cannot transform a circle into a triangle. It can, however, stretch a triangle along one compared to the other axis, and if adapted cone input affects the different directions of the axes differently, this asymmetry will be emphasized if the respective axis is at comparatively high resolution. This is what happens along the S-(L+M) axis in DKL-space. It is one of the reasons why sampling colour stimuli in DKL space implies that results are dominated by adaptation along the S-cone axes, which makes it more likely to obtain a correlation between predictions and data. That’s the reason why sampling in CIELUV is a stricter, more conservative test of the models. I added a little discussion of this issue for the specialist reader to section B.3 about Experiment 2.b in the supplementary material. The asymmetry along the S-(L+M) axis is also the reason why the subtractive cone-opponent model fails: It predicts exactly the contrary of what actually happens. I added that observation to the Discussion as it contributes to better understand the findings:

In contrast, subtractive adaptation of the cone-opponent mechanism fails to predict the three hue clusters and the asymmetry along the S-(L+M) mechanism (*Figure 1.c and S6.c*). The failure to predict the asymmetry explains why subtractive adaptation in DKL-space is comparatively unsuccessful in predicting afterimage chroma (*Figure 2.j*). (p.7)

5. The Equality of Results for DKL and CIELUV space

Using CIELUV is even more problematic because it is a nonlinear space designed to predict color differences. I don’t know of any cases where this has been used as an actual model of color mechanisms. I wonder if the 3 peaks shown in the afterimage results are again a peculiarity of the LUV space. In DKL space the changes along the L+M-S and L-M dimensions should be separable in terms of the cone signals, so I don’t see how these peaks would be predicted by cone-specific adaptation represented within DKL space.

RESPONSE

5.a Nonlinear effects of projective transformations

Although the comment does not sound like it, I think it might aim more at the flow of the manuscript rather than the actual content. Otherwise, I am struggling to understand what is missing in the manuscript to make clear that it is impossible that “the 3 peaks shown in the afterimage results are again a peculiarity of the LUV space.” Clearly, the first worry of any psychophysicist must be how a bias in stimuli or analyses could create spurious patterns of

results. This is the reason why I have spent so much time and effort to exclude this possibility – these conclusive efforts has been reported throughout the manuscript:

First, it has been shown in Figure 1 that neither stimulus representation nor determination of opponency in CIELUV produces 3 peaks (originally panel a, now panel c) as neither the red (subtractive adaptation in DKL) nor the black (CIELUV adaptation) curve show three peaks, and the predicted opponent angles are the same for both:

Panel c illustrates adaptation of the cone-opponent channels, where the afterimage is shifted to the hue opposite to the inducer hue (fat red line). For comparison, predictions based on adaptation in CIELUV space (thin black line) are also shown. Predictions of hue are the same as for cone-opponency, but predicted chroma differs from cone-opponent predictions-space because chroma remains constant across hue in CIELUV adaptation. (From the caption of Figure 1)

Second, the relationship between CIELUV and DKL-space is known from the literature, including my own publications, cited in the relevant Method section:

At isoluminance, opponent hues in CIELUV are the same as cone-opponent hues in Derrington-Krauskopf-Lennie (DKL) space⁴⁶ (Figure 1.a) and in (gamma-corrected) HSV- and RGB-space,^{6,25} Tristimulus Values (XYZ) and chromaticity coordinates (xyY); see section D.2.3 for mathematical details.⁴⁵

The cited references (Witzel & Gegenfurtner, 2018; Witzel et al., 2019) had shown that (1) lines in DKL/XYZ remain straight in isoluminant CIELUV, (2) discrimination follows Weber's law in CIELUV, and (3) CIELUV compensates for the strong variation of discriminability in DKL-space, which may produce stimulus and response biases. Point 1 has been shown by Figure S1 of (Witzel et al., 2019):

Figure S1. Hue circle in CIELUV-space. X- and y-axis correspond to u^* (red-green) and v^* (blue-yellow). The colored circle shows the CIELUV hue circle at $L^* = 76$ with a radius of 50 as used for stimulus sampling when investigating yellow. The thick grey-and-white lines correspond to the L-M and the (L+M)-S axis of DKL-space. Black and white segments correspond to equal steps along these axes in DKL-space. Note that the grey and white segments are almost constant along the two axes, indicating a simple, almost linear relationship between the isoluminant plane in CIELUV and in cone-opponent DKL-space (see Discussion of main experiments).

Point 2 can be seen from Figure 5 of Witzel et al. (2019); for other examples, see Figures S4 and S5 of that study:

Figure 5. Threshold intensity plot (TVI) for observer f1 in Experiment 1. The x-axis represents CIELUV chroma of the test colors as radius in u^*v^* . JNDs (y-axis) are differences in u^*v^* radius between test and just noticeable comparisons. Black circles correspond to measured JNDs; blue lines are fits to these JNDs with linear functions, the green curves are fits with power functions. The red line is the extension of the blue line up to the visible gamut along which JNDs were extrapolated for the main analyses. The panels in the first (a–c), second (d–f), third (g–i), and fourth (j–l) rows show data for the red, yellow, green, and blue stimulus sets, respectively. The panels in the center column (b, e, h, k) depict results for typical hues; those on the left (a, d, g, j) and right (c, f, i, l) side show the JNDs for lower- and upper azimuth boundaries. Corresponding results for other observers may be found in Supplementary Figures S4 and S5. Note that JNDs increase linearly as a function of test color radius (cf. results), in line with the Weber–Fechner law (cf. discussion).

Point 3 is shown by Figure 1.a in Witzel & Gegenfurtner (2018):

Fig. 1. Stimulus sampling. Panel a: Measurement of JNDs by Witzel and Gegenfurtner (2013). The colored circle illustrates the hue circle from which stimuli were sampled for the measurements in DKL-space. The distance of the grey curve from the center illustrates the size of average JNDs (in degree azimuth) for different hue directions. For illustration purposes JNDs are rescaled (divided by 13) to fit into the hue circle. The black ellipse is fitted to the JNDs (cf. Witzel & Gegenfurtner, 2013). For comparison, the dashed curves show CIELUV- (blue) and CIELAB- (red) chroma, rescaled (i.e. divided by 90 and 80, respectively) to fit into the hue circle. Panel b: Adjustments of unique (colored lines and symbols) and binary hues (black lines and symbols) in DKL-space. Disks correspond to the hue adjustments of individual observers. The size of the disks corresponds the frequency with which that hue has been adjusted. The line indicates the hue direction of the average across observers and the bars at the end of the lines correspond to standard errors of mean. The length of the lines is slightly lower than the radius of the test colors to avoid covering single dots. Fig. S1 provides corresponding figures for the measurements of unique and binary hues in CIELUV-space. Panel c: The fat grey ellipse corresponds to a circle of radius 50 at $L^* = 76$ in CIELUV-space. The thin dotted ellipses inside the grey ellipse correspond to a CIELUV-circle with radius 55 at $L^* = 60$, with radius 50 at $L^* = 50$, and with radius 34 at $L^* = 38$. The grey disk in the center is the adapting white-point and the origin of the DKL-space. Unique hues measured in DKL-space and at the four lightness levels in CIELUV-space are

Third, in response to this comment, I now added a mathematical proof of the projective, homographic transformation from XYZ to isoluminant CIELUV to the supplementary material (see subsection *D.3 CIELUV*). In short, when Y and L^* are constant, the relationship between XZ and u^*v^* depends on the transformation of the uniform chromaticity diagram, which is known from the literature to be a projective transformation of XYZ (Schanda, 2016). I also mention the relationship between isoluminant CIELUV and DKL at the beginning of the Results section and refer for details to the Method and from there to the aforementioned supplementary section:

At isoluminance, CIELUV predicts the same afterimage hues as cone-opponency, but different afterimage chroma (*Figure 1.c, cf. Method*).

Fourth, I had replicated the results of *Experiment 2.a* by sampling colours in DKL instead of CIELUV in *Experiment 2.b*. I now added a complement of *Figure 1* in form of *Figure S6* to the supplementary material, that illustrates the models in DKL space and illustrates the stimulus sampling of *Experiment 2.b*:

In terms of experimental practice, stimulus sampling in CIELUV space provides much better stimulus control and measurements than sampling in DKL because of the scaling along the cone-opponent (L+M)-S axis. I have added a discussion of this problem to the supplementary material:

Figure S6 shows that sampling in DKL space (*Figure 6.a*) is expected to produce even stronger afterimages in the purplish S-direction (*Figure S6.b*) than sampling in CIELUV space (*Figure 1.b*). In DKL, the afterimages in the S-direction are so strong that the grand average chroma is almost the same as the inducer chroma (0.47 vs 0.5) for participant CW and even higher than inducer chroma for participant f1 (0.71 vs 0.7). This complicates stimulus sampling in DKL-space in two ways: (1) The stronger shift of afterimages in the purplish direction dominates the predictions through cone adaptation. As a result, the S-pole dominated pattern of cone-adaptation in DKL space is less characteristic than the more distinctively tripolar pattern obtained in CIELUV space (cf. *Figure 1.b*). (2) Chroma had to be lowered in *Experiment 2.b* to avoid measurements reaching the display gamut. However, the difference between the afterimage strength along the S-cone direction and the comparative weakness of afterimages in the other hue directions make such a sampling almost impossible in DKL-space. For this reason, the chosen chroma setting in *Experiment 2.b* is a compromise between avoiding the gamut and keeping saturation along the circle sufficiently high to obtain clear afterimages in hue directions other than the purplish S-cone direction. At close inspection, some limitations of this approach are visible in *Figure S7*. A few individual measurements (grey dots) reach the gamut in the S-cone direction, especially for observer f1, indicating that the gamut limited the saturation of the adjustment. This clipping effect of the display gamut along the S-cone direction undermines the match of measurements with the predictions. The detrimental effect did obviously suffice to totally undermine those correlations as the measurements for the DKL-sampled stimuli were still in line with the cone-adaptation model. Nevertheless, the sampling in CIELUV allowed for a better stimulus control and cleaner measurements of afterimage colours and has therefore been the main approach to stimulus sampling adopted in this study.

Fifth, Experiment 3 shows that, even though lines remain straight, afterimage hue changes as a function of chroma in the way cone adaptation predicts.

Finally, I conducted a control experiment to show that three peaks are not inherent to the colour transformation, to the variation of hue (Witzel & Gegenfurtner, 2013; Witzel & Gegenfurtner, 2018) and chroma (Witzel et al., 2019) discrimination across hue, or to failures of device calibration. I don't see how the 3 peaks can be attributed to CIELUV space or any other spurious effect or bias.

(5.b) The CIELUV adaptation transform as a model of adaptation

Comment 5 also criticises CIELUV as a model of colour mechanisms. However, CIELUV has not been used as a model of colour mechanisms in this study. Instead, it has been included as a colour appearance model. The only overlap with models of mechanisms is the fact that predictions of afterimage hues through CIELUV adaptation are the same as predictions through cone-opponent adaptation. I revised the introduction of predictions based on colour appearance models in the Results section to clarify their role in this study.

Alternative models have been proposed to better predict several aspects of colour appearance than those physiological models and might better match the colour appearance of afterimages, either. CIELUV, CIELAB, and CIECAM02 implement complex, nonlinear models to approximate various empirical data on colour appearance.⁷

6. The purpose of Hering and Munsell predictions

The final spaces comparing the Hering and Munsell predictions are a bit of a “red herring,” because it is already very widely known that complementary colors are not opposite in terms of the Hering primaries. For example, many studies have noted that red and green are not complementary colors, and I believe it's already been well established that the afterimage of pure red looks like the complementary color in terms of cone excitation (blue-green) and not the pure green of the Hering opponency.

RESPONSE

The idea of a red herring suggests that the analyses of Hering and Munsell predictions are misleading or inappropriate. The present study measures the appearance of complementary colours. Whether we like it or not, the Munsell colour system and Hering colours have been considered as models of complementarity and colour appearance. This is clear from the literature (Conway et al., 2023; Fairchild, 2013), and I will further elaborate on this below (6.b and 6.c, respectively). The presented quantitative comparisons clearly show that these two approaches (Munsell and Hering) are the worst possible to understand the colours of afterimages (see Figure 2.i-j). They are so far off compared to other solutions that they cannot even be considered as very rough approximations. This point is not the focus of this study and is made in passing while other more important findings are highlighted. Given the relevance of the findings and the clarity of the results, I don't think these observations are inappropriate or misleading at all. Although it does not sound like it, I wonder again whether the comment is partially due to the framing of the idea in the original version of the manuscript and I revised those parts to make the motivation clear (see 6.a).

(6.a) Clarification of motivation for including Munsell and Hering opponency

I reworded the motivation in the Introduction as part of the revisions:

As a result of this confusion, textbooks, reviews, and other broader communications either do not attempt any explanation of colour afterimages^{2,4} or disseminate misleading narratives, such as the idea that afterimages reflect Hering-opponency between red-green and blue-yellow.^{6,8}

I now explain the role of these two approaches at the beginning of the Results section:

The Munsell system and Hering colours (Figure 1.e) provide yet another approach to identify complementary colours, which have been considered as points of reference for colour appearance.^{7,8}

And I expanded Figure 1 to include predictions of all models side-by-side for comparison. I hope that this makes clearer how each of these approaches provides a different take about the nature of complementary colours and are thus candidates for complementary afterimages:

I reworded the key result that undermines the idea that Hering colours approximately describe afterimages colours:

In naming tasks, afterimage hues of red, yellow, green and blue were called blue, purple, purple/pink and orange/brown, in stark contrast to Hering's red-green and blue-yellow opponency (cf. naming in Table S1-S2, Figure S2).

In that quote, I refer to a new Table S2 and Figure S2 that convey those (and other) results in detail. I then pick up that result in the Discussion to make this point more clearly before drawing the overall conclusion:

Although they have been long-time favourites to explain phenomena of colour appearance,^{7,8} the Munsell system and Hering opponency are in the worst predictors of afterimage colours (Figure 2.i-j). They predict afterimages to shift away from cone-opponency (Figure 1.d) in directions opposite to the predictions by cone-adaptation (Figure 1.a). The differences between afterimages and Hering opponency are so large that the corresponding colour terms, red, yellow, green, and blue, are not even rough approximations of afterimage complementarity (Figure S2, Table S1-S2). Together, these findings debunk the most widespread misconception that complementary afterimages are colour-opponent in any of the widely believed ways.^{2,4,8,20,34,36,38}

And I also mention it when discussing the implications at the very end of the Discussion:

Unlike what current textbooks suggest,^{4,6} colour afterimages are neither determined by cone- nor by Hering-like colour opponency. Instead, they can be fully and coherently explained by a well-founded straight-forward model of cone-adaptation at the first stage of colour processing. It's time to update the textbooks.

I hope the revised manuscript better justifies the inclusion of Munsell and Hering colours. In case, that there is doubt in content that these are relevant approaches to colour complementarity, I provide further details below (6.b, 6.c).

(6.b) Relevance of the Munsell System

The Munsell system has been a reference to evaluate opponency in colour appearance models until today (Fairchild, 2013). In addition, Munsell (Munsell, 1912, 1921) used a specific colour mixture procedure to determine Munsell complementarity through perceptual opponency. Here is how Munsell (1912) describes that test:

“A circuit of pigment hues around the equator is formed of five complementary pairs, -all of equal chroma to accord with equal departure from neutrality, and of equal value to accord with the central level. The sphere is mounted so as to permit rapid rotation, which melts this

equator of ten balanced hues in a band of neutral gray, proving by this visual balance that all degrees of sensation are compensated and constitute a true color equator.”

See also Munsell (1921) for more details. The complementary colours produced by this surface-colour based approach are fundamentally different in hue and saturation from any other model of colour appearance, as can be seen from the newly added red curve in Figure 1.e.

(6.c) Relevance of Hering Colours.

Afterimages were a key argument in the development of an opponent theory of colour perception following Hering's colour-opponency (Hurvich & Jameson, 1957). Still today, many textbooks refer to Hering colours when explaining afterimages. Here are a few examples. Goldstein and Brockmole (2016) write in the 10th edition of the world-leading textbook “Sensation and Perception”:

In the afterimage demonstration, red and green switch places and blue and yellow switch places. Thus, looking at red causes a green afterimage, looking at blue causes a yellow afterimage, and so on. These afterimages, which are called complementary afterimages because the afterimage is the color on the opposite side of the color circle, were taken by Hurvich and Jameson as supporting opponent-process theory.

Interestingly, Goldstein and Cacciamani (2021) have removed any explanation of colour afterimages from the new, 11th edition of the textbook. The compendium of colour psychology features the following in the chapter on colour appearance (Johnson, 2015, p. 687):

The fact that the afterimage is complementary hints at the opponent theory of color vision, as Hering postulated in the late 1800s (Hering, 1920).

Textbooks closer to colour applications than to colour research are even more likely to equate afterimages with Hering opponency, e.g., Rhyne (2017, pp. 20-21):

Hering's premise was that our psychological experience produces four distinct color hues from which all other colors are mixed. Hering based some of his opponent processing theory on color afterimages. For example, Red and Green are opposite colors. If we focus our vision on a Red dot and then gaze at a White wall, we will see a Green dot as an afterimage. Reversely, if we focus on a Green dot and gaze at a White wall, we will see a Red dot afterimage. The same results occur for Blue and Yellow.

For more textbook examples, see Manzotti (2017). Beyond textbooks, Bevil Conway et al's (2023) recent review in the renowned *Trends in Cognitive Sciences* highlights the importance and topicality of establishing the difference between Hering colours and complementary afterimages. They write:

Hering's hypothesis was compelling, for it not only invoked Johannes Müller's idea of 'specific nerve energies' but also gave scientific muscle to Goethe's poetic descriptions of color afterimages, which imply that opponency of some form must underwrite color appearance. (p. 3)

[...] Proof of color opponency was established decades before Hering, with the discovery of complementary-color pairs and color afterimages [29,30]; color opponency is implemented by retinal cone-opponent neurons [22]. (p.3)

However, their argument about the relationship between Hering colours and afterimages is unspecific and open to subjective considerations:

The unique hues [...] are not predicted by afterimages [84], and they are not complementary (although Hering thought they were) [9]. The afterimage (and complement) of red is bluish-green (not green). (p.8)

Conway's review observes that the afterimage of red is bluish-green, the reviewer comment describes it as blue-green, others have described it as cyan. Without any quantification of the difference, the emphasis that bluish green differs from (simple?) green might appear rather nitpicking, especially when considering that these statements are often made without reference to empirical measurements of afterimage colours. In stark contrast, the present study provides very precise measurements of perceived afterimages, colour naming, and Hering colours, and it quantifies how large the difference between Hering colours and afterimages is compared to alternative models. These observations debunks any even rough comparison of afterimages with Hering colours as misleading. I hope this contribution is clearer in the revised version of the manuscript.

7. Adaptation of Cone-Opponent Mechanisms

There is a further conceptual issue with trying to model color afterimages as sensitivity changes within an opponent mechanism. These mechanisms receive inputs of opposite sign and signal deviations from their neutral point. Adaptation "within" the opponent mechanism could make it more or less responsive, but would not shift the neutral point. The aftereffect should instead show up as a change in saturation. Altering the neutral point would instead require changing the balance of the inputs, or in other words adaptation at a pre-opponent level. An illustration of this idea is given in Webster JOV 2011 (see Fig 4). I can't see how in the present paper the effects of adaptation within – as opposed to prior to - an opponent mechanism have actually been modeled.

RESPONSE

Explanations of how adaptation within opponent mechanisms have been modelled were given in response (4.a) to the reviewer's comment 4, in particular in the revised Method:

Second-stage adaptation. Cone-opponency has been modelled with DKL colour space.(Derrington et al., 1984) Cone-opponent adaptation is assumed to be subtractive.^{11,19} So, it was calculated by subtracting the cone-opponent signal of the adapting colour from the cone-opponent signal of the achromatic probe.²⁰ [...]

(2) Cone-opponent adaptation to inducer:

$$DKL_{L-M} = CC_M - CC_L - k * iDKL_{L-M}$$

$$DKL_S = CC_S - k * iDKL_S$$

Where CC is the cone contrast calculated with equation (1), and the index indicates the respective cone (details on the cone-opponent axes are provided in section D-2 of the supplementary material). $iDKL_{LM}$ and $iDKL_S$ correspond to the cone-opponent signal of the inducer. Cone-opponent adaptation produces proportional shifts along the axes, implying cone-opponent afterimage hues, 180 degrees rotated away from the inducer and afterimage chroma that is k times the inducer chroma. Here, k was set to match the estimated strength of the afterimages, but the value of k was irrelevant to the results.

And in the Results:

Predictions for cone adaptation are computed with the mathematical model of Weber's law.¹¹⁻¹⁶ Using the inducer as the adapting colours produces the triangular shape of the red curve in *Figure 1.b*. This means that cone adaptation contradicts the wide-spread assumption that afterimages are cone-opponent. In contrast, cone-opponent afterimages result from subtractive, second-site adaptation, as illustrated by the red circle in *Figure 1.c*.

The subtractive model changes the neutral point. This is in line with the cited literature (Lee et al., 2008; Stockman & Brainard, 2010; Zaidi et al., 2012).

8. Role of other Mechanisms and Change of Title

The discussion includes reference 3 for the statement that most research assumes afterimages occur at an opponent site, but (since I wrote that review) I went back and looked at what I actually said: “Consider the fate of color afterimages, which begin with the first steps of seeing but then percolate through the system to reach our awareness. After viewing a red patch, a gray patch appears greenish. The sensitivity changes generating the aftereffect originate in the cone receptors. However, the afterimage lasts several seconds. This is too long to depend on the cones, and instead parallels the sluggish after-discharge in ganglion cells (Zaidi et al., 2012).” Webster Ann Rev 2015

This relates to points 1 and 3 above. It's important to distinguish where the sensitivity changes occur and how different stages are impacted and influence the resulting signals. In this regard the title of the present paper is misleading because there are many different mechanisms contributing to the properties of color afterimages, even if the sensitivity changes initially generating them occur at a particular site. A more appropriate title might be to replace “underlying” with “initiating.” The present paper does discuss the idea that the perception of afterimages can be shaped by downstream processes, but presents it without acknowledging the many previous analyses of how adaptation effects are inherited and modified by subsequent processes.

Response.

(8.a) Role of other Mechanisms

Reference 3 in the original version of the manuscript referred to the review of Webster (2016). I think *comment 8* is referring to this citation in the original manuscript:

Most state-of-the-art research assumes that afterimages correspond to the colour predicted by cone-opponent mechanisms at the second stage of colour processing^{2,3,6,13,22}

The quote mentioned above is now integrated in the Discussion and Webster (2016) is cited as *reference 3* here:

After cone adaptation, the adapted cone signal is propagated through the subsequent stages of colour processing.^{3,31,44} The rebound signal observed in the retinal ganglion cells indicates that the cone-adapted signal is temporally maintained at the second, cone-opponent stage.²⁰ This temporal maintenance likely contributes to the sustained duration of afterimages.^{3,20,32,33} According to the present results, processing of the colour signal at the second-stage seems not to affect the perceived colours of afterimages in any specific way. (p.7)

The classical criticisms about timing are presented just before:

Classical criticisms of the cone-bleaching explanation have been the difference between afterimages from alternating and static inducers, the duration of afterimages (>10s) being much longer than cone adaptation (<<100ms), and the low level of isomerisation at usual brightness levels.^{20,32,33} These criticisms seem questionable. First, similar chromatic afterimages for alternating and static inducers have been found more recently,^{52,53} and the perception of afterimages from dynamic inducers may be affected by factors that are unrelated to receptor adaptation.²⁶ Second, it has been recognised that cone regeneration may take minutes rather than milliseconds.^{54,55} Third, isomerisation levels correspond to the cone responses to the adapting stimulus, here the inducer. It seems contradictory that the level of isomerisation under usual brightness levels (1-10%) is sufficient to produce the saturated non-illusory colours of the inducers while being considered insufficient to explain the less saturated afterimages. (p.7)

(8.b) Title

The comment asks me to change the title from “The mechanisms underlying colour afterimages” to “The mechanisms initiating colour afterimages.” I don't think that “initiating” properly captures the point of the study. I am not a native speaker, but I would have thought

that “initiating” says little about the causal role but emphasises the temporal sequence. In that sense, afterimages are initiated by the presentation of the inducer/adaptor stimulus rather than cone or other adaptation. I also thought “underlying” means something like “at the origin,” which is exactly what this study is about. I feel the problem is really that the title puts too much weight on “mechanisms” and neglects the perceptual nature of complementary colours in afterimages, which is key to understand their origin. I suggest a new title that emphasises the nature of afterimages and conveys one of the most striking observations about that nature, the non-opponency: “The Non-Opponent Nature of Colour Afterimages.”

9. Link between Afterimages and Mechanisms

Finally, it's worth noting that there is a logical flaw in the textbook accounts of color afterimages, which as the paper notes, typically argue that color afterimages are evidence of opponency, because adapting to one color produces an afterimage of the opponent color. The flaw is that a negative aftereffect is not itself evidence of opponency. For example, in the tilt aftereffect, the test pattern is also tilted in the opposite direction, but no one suggests that this is because orientation is encoded by opponent mechanisms. For color, if we assume the cones adapt, then this adaptation is just rebalancing the responses of the cones. However, the same cone responses could also be produced by varying the stimulus rather than changing the state of adaptation, so adaptation itself is irrelevant to the question of why complementary cone responses appear as complementary colors, and again does not provide any evidence for opponency.

RESPONSE:

Yes, textbooks are wrong about opponency and afterimages. According to the present results, it is wrong to claim that afterimages precisely follow cone-opponency, but the comment aims at something else. Although the comment explicitly criticises textbooks rather than my study, I am worried that I might miss a criticism of my approach. While it is important to understand that aftereffects may arise from normalisation within mechanisms, I do not agree that colour opponency would occur in afterimages if there were no cone-opponent or other mechanisms that compare the responses of the cones.

First of all, I think the tilt aftereffect does not lend itself for a direct analogy with colour vision because orientation is physically opponent (a physical thing cannot be simultaneously oriented in one, say clockwise, and the opposite direction, say counterclockwise) while wavelengths are not. Second, I feel there is a need to clarify (a) what phenomena are considered to be opponent behaviour or phenomenology, and (b) under what conditions mechanisms are called “opponent”. The idea of colour-opponency is that one wavelength can affect, not responses to the same wavelength, but responses to a different wavelength. This is only possible if signals are compared by some sort of opponent mechanism. In comparative biology, phenomenological opponency is routinely tested in animals to understand whether receptor channels are compared or whether they are processed independently (Kelber & Osorio, 2010). Third, “nulling” or “balancing” within one receptor, i.e., adaptation, alone does not by itself produce colour opponency and complementary colours. If responses to wavelengths remain unrelated throughout the processing stages, we can null one mechanism, but there will not be an effect on the response to the stimulation of another mechanism. In the realm of colour, such unrelated processing may be found, for example, for the “labelled coding” of wavelengths in mantis shrimps (Thoen et al., 2014; Zaidi et al., 2014). Adapting one such labelled-coding receptor does not have an effect on responses to light that stimulates other receptors, but not that one. The same reasoning applies to human colour vision. According to univariance, activity of a single cone type confuses wavelength and intensity information. It is thanks to the comparison happening in the cone-opponent channels that colours are complementary. This is why I believe that afterimages would not produce opponent colours if there was no opponent processing – in fact, there would not even be different colours if we want to distinguish colour from ‘wavelength-specific behaviours’ (Kelber & Osorio, 2010).

This all said, I wonder whether these arguments are necessary to understand the rationale of my study. Let me reiterate this rationale as it seems from previous comments that it did not come across: The point of this study is not that cone-opponency is absent in afterimages, but that the adaptation that causes afterimages does not happen in the cone-opponent channels, but in the photoreceptors. Evidence for that can be seen from the observation that complementary colours in afterimages are not cone-opponent, but systematically bent towards cone sensitivities, exactly as predicted by cone adaptation.

10. Additional points

10.1 Physical sameness

1. To describe any stimulus as “actually colorless” is incorrect, because color is a product of the mind and what “looks” colorless depends directly on the state of adaptation.

RESPONSE

Sorry, this was an attempt to phrase this in a simple fashion. I reworded the sentence as follows:

For example, if you fixate on a yellow circle, when taken away you will see a blue-purple circle even though it is **physically the same as the colourless surround** (see animated illustration 1).

10.2 Overlapping Cone Sensitivities

2. The statement that S cone sensitivity barely overlaps with M and L cone sensitivity is also incorrect, since the L and M sensitivities span the entire visible spectrum.

RESPONSE

Thank you, I changed it to “comparatively little”:

Since S-cone sensitivity overlaps **comparatively little** with M- and L-cone sensitivity, the effect of S-cones on L-M-adapted afterimages is much stronger than the effects of the correlated L- or M-cones, resulting in a pronounced asymmetry along the cone-opponent S-(L+M) axis.

10.3 Control Experiment

3. I did not understand the logic of the control experiment reported in the supplement. It seems to simply be showing that observers can accurately match two colors presented side by side, which for the conditions of the study seems evident.

RESPONSE:

As explained above (5.a), the control experiment was meant to assess biases of colour adjustment that are not effects specific to afterimages. Such biases could be the results of stimulus sampling and colour transformations, variation of hue (Witzel & Gegenfurtner, 2013; Witzel & Gegenfurtner, 2018) and chroma (Witzel et al., 2019) discrimination across hues, or to imperfections of device calibration. I reworded the statement in the Results section of the main text:

These shifts in hue selection did not occur in a control task, in which observers matched real colours in an otherwise same test display, thus excluding spurious technical artifacts or response biases (Figure S1.b).

And I added a little summary of results section to section A.3 of the supplementary material:

The results of the control task show that the biases in colour selection in the afterimage task can neither be attributed to technical artefacts, such as imperfections of device calibration, nor to nonlinearities in stimulus sampling and representation, nor to unspecific response biases, for example because of the variation of discriminability between the comparison colours (cf. Witzel & Gegenfurtner, 2018; Witzel et al. (2019)).

10.4 Von Kries in Colour Appearance Models

4. The statement that color appearance models include adaptation parameters but don't predict the afterimage colors seems surprising, since for some of these models the adaptation transform is Von Kries scaling and therefore the same as the cone-specific adaptation proposed here. It would be good to clarify why this doesn't work.

RESPONSE

Colour appearance models aim at fitting a large range of different types of colour measurements, including for example colour discrimination, opponency, and adaptation. To my knowledge, none of them uses pure von Kries scaling (Fairchild, 2013). CIELUV uses a Judd transform (simple shift). CIELAB involves a pseudo-von Kries applied to the Tristimulus Values instead of LMS. CIECAM02 implements a von-Kries transform of Judd primaries, RGB, which differ from LMS. I added detailed explanations of the relevant adaptation transforms in those models to the supplementary material (Section D. Models). I also added the resulting models to Figure 1 to give a better impression of how they differ from cone adaptation:

One can see, indeed, that CIELAB and CIECAM02 adaptation transforms predict three clusters, but the clusters are tilted compared to cone adaptation. In line with the computational inclusion of von-Kries like adaptation transforms, CIELAB and CIECAM02 tend to provide better

predictions of afterimages than cone-opponency, but lag behind proper von-Kries cone adaptation. I added a brief explanation of this to the main text of the Results (see below).

CIELAB and CIECAM02 (Figure 1.d) incorporate adaptation transforms inspired by von-Kries adaptation and produce a similar triangular shape as cone-adaptation but tilted.

I then discuss these results in the Discussion:

CIELAB and CIECAM02 provide better predictions than cone-opponency because their von-Kries-like adaptation transforms produce 3 clusters close to those of cone-adaptation. Nevertheless, there is a difference from cone-adaptation because CIELAB and CIECAM02 use a pseudo von-Kries transformation, applied to primaries that are similar, but not the same as the cones (see supplementary sections D.4-5 for details). This suggests that the incorporation of a proper cone-adaptation transform in colour appearance models would improve their ability to predict complementary afterimages.

This observation is relevant for the implication raised at the end of the Discussion:

As illustrated by those widely used colour appearance models (Figure 1.b-c), a better understanding of the effects of adaptation on colour appearance is also key for the measurement, specification, and control of colours (colorimetry) in scientific and industrial applications.⁷

10.5 Importance in science and industry

5. The paper ends by arguing that the present results demonstrating cone specific adaptation have important implications for science and industrial applications, but again ignores the fact that this model is already widely adopted in both science and industry.

RESPONSE

This comment refers to the following statement from the original version of the manuscript:

As a result, this quantitative model is key for the measurement, specification, and control of colours (colorimetry) in scientific and industrial applications^{4,6,8,9}.

Thank you! I did not think about this connotation of the statement. What I meant is that understanding what factors influence colour appearance also helps improving colour appearance models and measurements used in art and industry. From my perspective, the fact that science and industry already use this type of knowledge underscores the importance and relevance of such knowledge in science and industry. I tried to capture this idea by rewording the statement as follows:

As illustrated by those widely used colour appearance models (Figure 1.b-c), a better understanding of the effects of adaptation on colour appearance is also key for the measurement, specification, and control of colours (colorimetry) in scientific and industrial applications.⁷

REVIEWER #2 (REMARKS TO THE AUTHOR):

This seems to be an excellent paper that is far outside my area of expertise. That being said, I have played around with chromatic afterimages using an adjustment method similar to the one used here with “chaser” stimuli. NB: Using chaser stimuli was a nice idea.

1. The Role of Brightness

My only query is whether perfect (metameric) matches could be obtained without allowing for a change in brightness (i.e., in addition to the changes in hue and chroma that were allowed).

RESPONSE

Yes, I had double-checked this. It is known from the literature (e.g., Kelly & Martinez-Uriegas, 1993) that isoluminant inducers produce isoluminant afterimages. However, unlike them I did not control luminance individually using heterochromatic flicker photometry because I wanted to test general adaptation models. So, I double-checked the perceived brightness in a few pilot-measurements with adjustable lightness L^* and myself as the participant. The results, i.e., the adjusted afterimage lightness (L^*) as a function of inducer hues, are pasted below (left panel). For reference, the grey horizontal line visualises inducer luminance ($L^*=70$). The grey curve with error bars illustrates averages and standard deviations, and the black dots are single datapoints. Note the y-axis scaling. Average deviations from inducer luminance were smaller or equal to 1 L^* unit, which is meant, according to the CIELUV rationale, to represent a JND when test areas are adjacent. In other words, variation of adjustments was tiny even though luminance was not controlled individually and despite noise in the adjustment method. The right panel shows the adjusted chromaticities, showing the triangular pattern known from the data in the article, where luminance was fixed. It seems safe to assume that lightness is sufficiently close to inducer lightness to focus the main measurements on hue and chroma.

2. Randomisation

Oh. Another query: Were inducers randomly interleaved and/or was the adjusted stimulus randomised at the beginning of each trial? I ask because when I was watching undergrads use the aforementioned adjustment method, they had a bias to make settings similar to previous settings.

RESPONSE

(2.a) Randomisation of trials

Randomisation of trials in all experiments is specified in the Method section (page 3):

For Experiment 1: “The order of trials was randomized.”

For Experiment 2/3: “One block of measurements featured one trial for each of eight inducer colours in a series, presented in random order.”

(2.b) Random Initial Colour

I added an explanation about the randomisation of the initial colour of the test circle in Experiments 2 and 3:

The initial colour of that circle was set to a random hue at inducer chroma.

3. Even more minor comments:

3.1 Numbering of Experiments

Please ensure the experiments are consistently numbered. The legend of Figure 2 says that Experiment 2 uses the chaser stimuli, but these are described under Experiment 1 in the Method section.

RESPONSE

Yes, thank you! I corrected those issues throughout the Method of Experiment 1 (highlighted blue); this is the example mentioned in the comment:

The eight inducer colours in Experiment 1.a were chosen to correspond to the typical lightness and hue of red, orange, yellow, green, turquoise, blue, purple, and magenta at the maximum chroma possible within monitor gamut. [...] In Experiment 1.b, twenty-four inducers were sampled along a hue circle in CIELUV at chroma 71 and equal steps of 15 deg starting at 0 deg (cf. hue circles in Figure 1). (p.2)

3.2 Confusing sentence

In the Method section, under Models: "Since adaptation is proportional to the intensity of the adapting stimulus, one needs proportionally more stimulation for detecting a non-adapted stimulus." "more" should be "less," right?

RESPONSE

That text bit was confusing and unnecessary. I deleted it for the sake of brevity.

3.3 Equations for Models

Given this paper's focus on quantitative modelling, I was disappointed that I couldn't find the quantitative model anywhere in this paper. Yes, there are some words, but I think the reader should be given enough information to compute the same quantitative predictions for afterimage hue and chroma given inducer hue, chroma, and brightness. I don't think all of that information appears in this paper. (It may appear in the Supplementary Material. Can't it fit in the paper itself?)

RESPONSE

I revised the introductory part of the Model subsection of the *Method*:

With all models, the colour signal of the grey probe circle (same as background colour) was calculated under local adaptation to the inducer. Then, the colour of the resulting afterimage was determined as the locally adapted colour signal under global adaptation to the grey background. The known fact that afterimages are not as saturated as inducers (Figure 2.f) implies that adaptation to inducers is not complete. As an estimate of the strength of adaptation, we set the adapting chroma for modelling afterimages to the chroma of the comparison colours in *Experiment 1*, and to the grand average chroma (27.0 in CIELUV and 0.47 in DKL) of adjustments in *Experiment 2*. Even if we assumed complete adaptation to the chroma of inducers, results would be largely the same. However, not surprisingly, correlation coefficients involving cone adaptation would be slightly lower because non-linearities in the model are then higher than in the measurements. There are no free parameters in any of the models.

I added the equations of the two main models on cone adaptation and cone-opponency to the main text and details on these and the other models to the section *D.Models* of the Supplementary materials. First, the model for cone adaptation is reported as:

(1) Cone adaptation to inducer:

$$CC_{LMS} = \frac{bgLMS - iLMS}{iLMS}$$

Where *bgLMS* and *iLMS* refer to Stockman-Sharpe cone excitations of the grey background and the adapting inducer. Cone excitations were scaled to match the luminous efficiency function.⁵¹ Cone contrast changes with increasing cone adaptation *iLMS* according to a multiplicative inverse function (because *iLMS* increases in the denominator). This adaptation produces shifts towards the peak cone sensitivities (Figure S8), as illustrated by the three peaks in the hue histograms (Figure 1.c, Figure 2.d, h).

And then model for cone-opponent adaptation is provided here:

Second-stage adaptation. Cone-opponency has been modelled with DKL colour space.(Derrington et al., 1984) Cone-opponent adaptation is assumed to be subtractive.^{11,19} So, it was calculated by subtracting the cone-opponent signal of the adapting colour from the cone-opponent signal of the achromatic probe.²⁰

(2) Cone-opponent adaptation to inducer:

$$DKL_{L-M} = CC_M - CC_L - k * iDKL_{L-M}$$

$$DKL_S = CC_S - k * iDKL_S$$

Where *CC* is the cone contrast calculated with equation (1), and the index indicates the respective cone (details on the cone-opponent axes are provided in section D.2 of the supplementary material). *iDKL_{LM}* and *iDKL_S* correspond to the cone-opponent signal of the inducer. Cone-opponent adaptation produces proportional shifts along the axes, implying cone-opponent afterimage hues, 180 degrees rotated away from the inducer and afterimage chroma that is *k* times the inducer chroma. Here, *k* was set to match the estimated strength of the afterimages, but the value of *k* was irrelevant to the results. Although a theoretically complete model would differentiate between the two separate mechanisms that constitute each opponent channel ("half-wave rectification"), such a separation did not change the predictions in this context.

Section D.2 *Cone-Opponent Model* of the *Supplementary Material* provides details on the relationship between this formulation of cone-opponency and the comparatively complicated, but widely used transformation matrix, e.g., in Psychtoolbox. I also improved the description of colour appearance models:

Adaptation of colour appearance. Predictions by colour appearance models (CIELUV, CIELAB, CIECAM02) were computed as the colour appearance of the grey background when the adapting white-point was the inducer, using the respective adaptation transform. At isoluminance, CIELUV is a projective transformation of DKL-space and involves a subtractive adaptation, which explains the identity of predicted hues. Sections D3-D5 in the supplementary material provide mathematical details and explanations. For predictions by Munsell and Hering colours, opponent colours were determined in the respective coordinate systems, which is equivalent to the effects of subtractive adaptation. Opponent colours in the Munsell system were interpolated using the CIELAB coordinates of the Munsell renotation table⁵² as provided by the Munsell Lab of the Rochester Institute of Technology. Details about the interpolations are provided in section D-6 *Munsell-Opponent Model* of the *Supplementary Material*. Hering-opponent colours were calculated by linearly interpolating the hue direction between the respective two empirically measured prototypes for red, yellow, green, and blue in CIELUV space. There was no prediction for Hering chroma. For details, see section D-7 *Hering Opponent Model* of the *Supplementary Material*. (p.4)

I added extensive details about these models to subsections D.3-7 of the *Supplementary Material*. These include: The proof and homogenous coordinates for the projective transformation of Tristimulus Values to isoluminant CIELUV (subsection D.3), the adaptation

transforms for CIELAB and CIELUV (subsections *D.4-5*), details about the identification and interpolation of complementary colours in the Munsell system and Hering opponency (subsections *D.6-7*).

REVIEWER #3 (STEVE ENGELS):

This manuscript addresses a basic phenomenon- negative color afterimages- that has received surprisingly little attention from the color psychophysics community recently, and presents convincing evidence that afterimages are caused by adaptation to the cone photoreceptors. The experimental methods and basic results appear to me to be solid and important. I have a number of concerns about their presentation, however, which I detail below.

1. Introduction: Framing of the debate.

I find the intensity of the debate regarding afterimages to be overstated as it mainly arises from researchers not based in the color community. This in turn leads the issue to be treated in isolation (see #2, below). But, is anyone from the color psychophysics community currently seriously arguing- based on data- that adaptation of cone signals is not at least one major cause of negative afterimages? As far as I can tell, the main empirical "hard core" evidence against the photoreceptor hypothesis are only the two Loomis papers from the 1970s, and the more recent Zaidi et al. paper. This hardly constitutes a raging debate. Sure, non-color researchers have been arguing that certain stimulus configurations may produce adaptation in post-receptoral, even cortical, mechanisms as well. And engineers designing color spaces may not be modeling adaptation correctly. But even in these two groups, I think very few would propose that receptoral adaptation plays no role at all in afterimages. While the present manuscript makes a reasonably important contribution, it could more accurately characterize the consensus in the core community.

RESPONSE

(1.a) The nature of disagreements

Before I started this comparatively effortful line of research, I had assumed that the nature and origin of complementary afterimages had been established since a long time. I was indeed surprised to learn how much experts in the field disagree about the explanation of this fundamental phenomenon. The contradictions and confusions of the state-of-the-art are not simply about the possible involvements of different mechanisms of adaptation but concern the relationship of those mechanisms and the perceptual nature of afterimages. For example, it turned out that experts in the field were not aware that simple cone-adaptation has nonlinear effects. Others believed that afterimages are cone-opponent because colours defined in cone-opponent DKL-space cancel (or "null") afterimages. This is not true either, as I explain in the Discussion. However, it seems that the original *Introduction* failed to properly convey the contradictions between the assumed neural mechanisms at the origin and the perceptual nature of afterimages. I have made this clearer in the revised version of the Introduction, in particular through the part highlighted green:

Researchers have disagreed about the neural mechanisms of adaptation that cause complementary afterimages.^{6,20,25,26} Classical studies suggested that such afterimages are caused by photoreceptor desensitisation ("cone bleaching") at the first stage of processing.²⁷⁻³¹ Later studies firmly contradicted this idea,^{20,32,33} attributed afterimages to adaptation in the cone-opponent channels at the second stage,^{13,24,26} or emphasised the effects of cortical adaptation.^{33,35,43}

Even more importantly, the link between neural mechanisms and the perceived colours of afterimages has been ignored or misconstrued. Irrespective of the hypothesised neural origin, most state-of-the-art research assumes that the perceived colours of afterimages are cone-opponent,^{20,34,36,38,44} but there is evidence against that assumption.^{25,45} As a result of this confusion, textbooks, reviews, and other broader communications either do not attempt any

explanation of colour afterimages^{2,4} or disseminate misleading narratives, such as the idea that afterimages reflect Hering-opponency between red-green and blue-yellow.^{6,8} (p.1)

I also revised the rationale to better communicate that the contribution is the link between perceptual nature and underlying mechanisms:

This study leveraged the fact that adaptation of mechanisms at different stages along the hierarchy of colour processing make markedly different quantitative predictions about the colours of complementary afterimages (*Figure 1*). Using tailormade experimental paradigms, the exact colours perceived in afterimages were measured for a large range of inducers to test those predictions. Three comprehensive experiments unambiguously showed that the colours of afterimages tightly follow the non-opponent predictions that are specific to cone adaptation. (p.1)

(1.b) Confused State of the Art

However, I don't think state-of-the-art knowledge about the mechanisms that form afterimages are as settled and homogeneous as this comment suggests. A thorough literature review, a poll, and many discussions with colleagues in the field, revealed that there is indeed complete confusion about the explanation of afterimages. Let me provide some examples. First of all, the idea that the mechanisms underlying afterimages are not established is shared by other colleagues. For example, Zaidi et al. (2012) write:

“Proposed physiological mechanisms for color afterimages range from bleaching of cone photopigments to cortical adaptation [4-9], but direct neural measurements have not been reported.”

Koenderink et al. (2020) conclude:

Indeed, it cannot be said that the mechanism of negative afterimages is fully understood (Anstis, 2017). (p. 12)

Second, recent arguments against cone adaptation have been very strong and had impact on specialist research. As acknowledged in the comment, Zaidi et al. (2012) firmly argued against cone adaptation:

“This result should correct the notion, found on the web and in many textbooks, that photoreceptor desensitization is responsible for color after-images generated by normal light levels.” (p.222)

Kingdom et al. (2020) took this idea from Zaidi et al., although it does not easily align with their nonlinear results:

“Negative, or complementary afterimages are experienced following brief adaptation to chromatic or achromatic stimuli, and are believed to be formed in the post-receptoral layers of the retinae.” (from the abstract)

“In relation to the present study, in which TvC functions were measured, the RGC explanation for afterimages provided by Zaidi et al. needs to be squared with the psychophysical evidence for separability of the poles of the cardinal axes in the context of stimulus detection [...]” (p.32)

Others claim that afterimages are neither the result of receptor adaptation nor cone-opponency but of cortical adaptation, for example Zeki et al. (2017):

Hence, traditional accounts of after-images as being the result of retinal adaptation or the perceptual result of physiological opponency, are inadequate. We propose instead that the color of after-images is generated after colors themselves are generated in the visual brain. (from the abstract)

Shimojo et al. (2001) claim in their Conclusion:

“Our findings are consistent with previous studies suggesting that afterimages produced by long exposures to moderate-intensity patterns are **not due to photopigment bleaching** but rather to neural adaptation (26–28), and that the appearance of the afterimage depends on the perceptual, not physical, attribute of the adapting stimulus (26, 29, 30).” (p.4)

Similarly, Phuangsuan et al. (2018):

The relation relationship of $\Delta\theta$ to the angle of the adapting color θ_{ing} was quite similar to the results obtained by the two-room technique, implying that the chromatic adaptation shown by the afterimage also **occurs in the brain rather than in the retina**. (from the abstract)

Others claim that everything is somehow involved in the formation of afterimages, at least under some conditions. For example, Dong et al. (2017) say in their abstract:

“**Our results thus contradict the retinal generation notion**, and suggest that in addition to the retina, cortex is directly involved in the generation of AI signals.” (from the abstract, but see also Introduction & Conclusion)

Or Van Lier et al. (2009):

“Our results with afterimages indicate that **cortical** color filling-in processes are also involved when incoming signals are caused by adaptation of **retinal receptors**.” (p.323)

I also polled attendees of an international colour symposium, and results showed that one quarter believed afterimages are cone-opponent, about one half that they are not, and another quarter responded they don't know:

So, even experts cannot agree on the nature of afterimages, and their explanation thus remains guesswork, open to speculations and selective reading of the literature.

The contradictory and confusing state of the art has had important effects on textbooks and reviews. Explanations in our most important textbooks are fragmented or missing, leaving students in vision science confused. After claiming that afterimages are a signature of Hering colours in earlier editions, the newest edition of Goldstein's Sensation and Perception (Goldstein & Cacciamani, 2021) has completely given up trying to explain afterimages. Wolfe's Sensation and perception correctly explains afterimages through receptor adaptation but

claims at the same time that afterimages are cone-opponent. This does not fit together, as I show in this study.

Reviews and textbooks outside the fields are all over the place. Many suggest that afterimages follow Hering-opponency, as summarized by Manzotti's review (Manzotti, 2017)

Remarkably, scientists and philosophers have provided biased and imprecise reports based on color opponency (Byrne & Hilbert, 2003; Churchland, 2005; Goldstein, 2010; Hurvich, 1981; Jones, 1972; Lycan, 2002; Mach, 1897; Macpherson & Platchias, 2013; Palmer, 1999; Schwitzgebel, 2011; Werner & Bieber, 1997), and only a minority of accounts have considered the complementary nature of afterimages (Bidwell, 1897; Geisler, 1978; Livingstone, 2002; Livitz, Yazdanbakhsh, Eskew, & Mingolla, 2011; Pridmore, 2008; Wilson & Brocklebank, 1955; Zaidi et al., 2012). Finally, the notion of complementary color varies from author to author (Anstis, Vergeer, & van Lier, 2012; Goldstein, 2010; Hurvich, 1981; Livingstone & Hubel, 1987; Zaidi et al., 2012), and it is occasionally confused with that of opponent colors (Livingstone, 2002; Tsuchiya & Koch, 2005).

No publication makes the point that cone adaptation to different inducers implies systematic deviations from cone-opponency. However, exactly that resolves the confusing and contradictory literature by closing the gap between candidate neural mechanisms and the phenomenological nature of colour afterimages.

2. Introduction: Missing background.

Part of the reason some current papers have argued for exotic mechanisms being responsible for afterimages might be that they treat the issue in isolation from what is known about color vision more generally, and this manuscript repeats that mistake to some extent. The paper would greatly benefit from presenting the "standard" process model of color vision at the beginning. Many, and I would venture most, "back pocket" process models of color vision (vs models used to form color spaces) assume that photoreceptor signals adapt somewhere along the visual pathways, and that this adaptation has behavioral consequences; i.e. Von Kries or "generalized Von Kries" adaptation has received much psychophysical and physiological support. This should be noted up front in the paper. This manuscript, then, tests, and fails to support, the specific hypothesis that the appearance of color afterimages is dependent upon different mechanisms, that somehow bypass photoreceptor adaptation. For example, photoreceptor adaptation might fade too quickly to be relevant for relatively long-lasting color afterimages. Similarly, the discussion should emphasize more that the photoreceptor adaptation that is in fact the basis for color afterimages has lots of support from different behavioral and neural paradigms besides measures of afterimages.

RESPONSE

(2.a) The standard process model of colour vision

Yes, the compartmentalisation of existing knowledge might have been the reason why the important link between cone adaptation and afterimage phenomenology has not yet been established. I have added a brief summary of the hierarchy of colour processing:

Human colour perception is the result of several stages of processing, starting with the excitation of the cone photoreceptors in the eyes, propagating through cone-opponent mechanisms in the retinal ganglion cells, the LGN of the thalamus, and the double- and single-opponent cells in the primary visual cortex, until the colour signal reaches higher visual areas that produce the subjective experience of colour.¹⁰ Adaptation may occur at any point along this visual hierarchy of colour processing.

(2.b) Mentioning Von Kries Adaptation

I now mention von Kries up front in the Introduction, first here:

Although, candidate mechanisms of adaptation have been known for more than a century,⁹ state-of-the-art explanations of this important phenomenon remain contradictory and confused. [p.1]

And then here:

Adaptation of photoreceptors at the first stage of colour processing produces divisive adaptation, similar to von Kries original proposal,⁹ and can be approximated as contrast coding following Weber's law.¹¹⁻¹⁶ [p.1]

(2.c) The bypass hypothesis

I hope my revision in response to *comment 1* made clearer that this study is not merely about identifying neural mechanisms but about linking the mechanisms at the origin of afterimages and the perceptual nature of afterimages to provide a coherent explanation that resolves the confusion in the field. On this background, I don't think the by-passing hypothesis does justice to the contribution of this study. The null-hypothesis that cone adaptation is "bypassed" would assume that the cone-contrast model has been known as an explanation of the non-linear distribution of afterimage colours, and that this study merely shows that it does so to a high degree. But that's not the case! Although, the cone-adaptation model has been known and used in other contexts, and some studies (Koenderink et al., 2020; Wilson & Brocklebank, 1955) support the idea that colours are not cone-opponent contrary to what everybody assumes, the link between both has not been recognised and measurements of afterimage colours were too imprecise to properly test the idea. Neither a quantitative model nor the perceptual nature of afterimages is established. That a simple cone-adaptation model explains precisely the colours of afterimages is the key contribution of this study. Without knowing this (in my opinion) very important result, alternative hypotheses are divers and have been diverse because all kinds of assumptions about mechanisms and about the perceptual nature of afterimages were combined without realising that they are mutually exclusive – that is the confusion in the state-of-the-art that I am referring to in my response to *Comment 1*. A first contribution of the study is that the testing logic, as illustrated in *Figure 1*, debunks these contradictory assumptions:

If we believe that afterimages are shaped by cone adaptation, then we should expect, according to my modelling, that perceived afterimages are clustered, as shown in *Figure 1.b*, instead of being cone-opponent. If we believe instead that afterimages are cone-opponent, then we need to assume, instead, that the underlying adaptation is subtractive as illustrated in *Figure 1.c* and suggested, for example, by Zaidi et al. (2012). If something else happens, be it bypass, complex combinations, or additional adaptation of yet unknown mechanisms, then none of the models in *Figure 1.b-c* should be successful. Colour appearance models (*Figure 1.d-e*) might provide alternative approaches that capture the complexities of subjective colour appearance and phenomenology. Still other models that we do not yet know, might be possible. This is the full range of alternative hypotheses ones we acknowledge that there is a link between mechanisms and perceived afterimage colours. I have revised the explanations of alternative predictions at the beginning of the Results section, and hope this better shows the range of possible alternative explanations:

Predicted hue and chroma of afterimages differ across models. Predictions for cone adaptation are computed with the mathematical model of Weber's law.¹¹⁻¹⁶ Using the inducer as the adapting colours produces the triangular shape of the red curve in Figure 1.b. This means that cone adaptation does agree with the wide-spread assumption that afterimages are cone-opponent. In contrast, cone-opponent afterimages result from subtractive, second-site adaptation, as illustrated by the red circle in Figure 1.c. Cone-opponency has been modelled with DKL colour space.⁴⁶ Alternative models have been proposed to better predict several aspects of colour appearance than those physiological models and might better match the colour appearance of afterimages, either. CIELUV, CIELAB, and CIECAM02 implement complex, nonlinear models to approximate various empirical data on colour appearance.⁷ At isoluminance, CIELUV is a projective transformation of DKL-space^{36,37} and the subtractive adaptation transform in CIELUV predicts the same afterimage hues as cone-opponency, but different afterimage chroma (Figure 1.c, see also section D-3 in the Supplementary Material). CIELAB and CIECAM02 (Figure 1.d) incorporate adaptation transforms inspired by von-Kries adaptation and produce a similar triangular shape as cone-adaptation but tilted. The Munsell system and Hering colours (Figure 1.e) provide yet another approach to identify complementary colours, which have been considered as points of reference for colour appearance.^{7,8} (p.4)

3. Introduction and discussion: Later mechanisms.

Similarly, the standard process model also proposes that "second site" adaptation occurs, depending upon induction procedures, for in example transient tritanopia and contrast adaptation. It would seem likely that these have effects upon afterimages, and in the discussion the authors acknowledge that later adaptive processes that adapt can have effects on afterimage appearance. So, I would suggest clearly stating up front the hypothesis this paper is testing in more detail: That photoreceptor adaptation is necessary to produce color afterimages, and that adaptation of cone-opponent mechanisms is not necessary (but not that photoreceptor adaptation is sufficient to explain their appearance). This or similar phrasing might be helpful throughout the manuscript, including the discussion, which seems to suggest that second site adaptation does not affect afterimage formation. That may be true for the present data, but many other stimulus configurations, including more natural ones, could well produce both first- and second-stage adaptation.

RESPONSE

(3.a) The possible role of later sites of adaptation in this study

I added a clearer explanation of other mechanisms (or "sites") of adaptation to the Introduction:

A slower type of "second-site adaptation" occurs in the retinal ganglion cells,¹⁷ and is assumed to be subtractive.^{11,18-20} Other observations suggested "higher-order" adaptation of yet unknown, cortical mechanisms.^{3,21-24} (p.1)

I would not like to start discussing transient tritanopia and contrast adaptation because they are different phenomena than afterimages. Transient tritanopia (Mollon & Polden, 1976; Mollon et al., 1987; Valeton & Norren, 1979) is about complexities in the time course of adaptation, which is not the topic of the present study. On the contrary, the experimental tasks here have been designed to keep afterimages at maximum strength and minimise afterimage fading to avoid complex effects of the time course of afterimage formation and fading. With respect to contrast adaptation, it is clear that complex aftereffects other than simple afterimages may be found that involve more complicated mechanisms of adaptation, but this is not the topic of this study. For example, faces have also colour, but this does not mean that I am claiming aftereffects in face perception are the result of cone adaptation. The manuscript has not mentioned aftereffects other than afterimages at any point. The only exception were some unfortunate citations, which I have rectified. I have also revised the discussion of the cone bleaching hypothesis in a way that makes the focus of this study clearer:

Classical criticisms of the cone-bleaching explanation have been the difference between afterimages from alternating and static inducers, the duration of afterimages (>10s) being much longer than cone adaptation (<<100ms), and the low level of isomerisation at usual brightness levels.^{20,32,33} These criticisms seem questionable. First, similar chromatic afterimages for alternating and static inducers have been found more recently,^{52,53} and the perception of afterimages from dynamic inducers may be affected by factors that are unrelated to receptor adaptation.²⁶ Second, it has been recognised that cone regeneration may take minutes rather than milliseconds.^{54,55} Third, isomerisation levels correspond to the cone responses to the adapting stimulus, here the inducer. It seems contradictory that the level of isomerisation under usual brightness levels (1-10%) is sufficient to produce the saturated non-illusory colours of the inducers while being considered insufficient to explain the less saturated afterimages.

At the same time, it has become clear that cone adaptation at moderate light levels involves processes other than “cone bleaching.”^{56,57} In addition, the time course of adaptation is complicated⁵⁸⁻⁶¹ and differs across cone types.^{62,63} The tasks in the present study were designed to measure afterimage colours close to maximum adaptation and to avoid temporal effects during afterimage formation and fading. The complexity of the underlying physiological processes notwithstanding, the resulting divisive cone adaptation to moderately bright, photopic colours can be approximated by cone contrasts according to Weber’s law.^{11-14,16,51}

After cone adaptation, the adapted cone signal is propagated through the subsequent stages of colour processing.^{3,31,44} The rebound signal observed in the retinal ganglion cells indicates that the cone-adapted signal is temporally maintained at the second, cone-opponent stage.²⁰ This temporal maintenance likely contributes to the sustained duration of afterimages.^{3,20,32,33} According to the present results, processing of the colour signal at the second-stage seems not to affect the perceived colours of afterimages in any specific way. (p.7)

(3.b) Chromatic adaptation with other stimuli

I am happy to note that the reviewer acknowledges the great fit between predictions and data in this study. When it comes to stimuli other than the ones used in this study, in particular natural ones, one must wonder: **What do they have that my stimuli don’t!** We can certainly conceive stimuli that involve other phenomena, such as transient tritanopia, or contrast adaptation. We may also combine other phenomena with afterimages, such as local contrast induction (Anstis et al., 1978), shape recognition (van Lier et al., 2009), and illusory contours/areas (Shimojo et al., 2001). Natural stimuli also contain meaningful objects and may produce additional top-down effects on colour appearance (Lee & Mather, 2019; Lupyan, 2015). Each of those effects (contrast induction, shape recognition, illusory contours, memory colour effects) exist independently of complementary afterimages, and the interaction of these effects with afterimages is a matter of its own. Confusion has been produced by research that concluded about the origin of afterimages from top-down effects on the perceptual evaluation of afterimages. I have addressed this in the Discussion:

Others claimed that the colours of afterimages are shaped through adaptation of cortical mechanisms that involve shape and object recognition,^{27,30-32,46} binocular integration^{33,47} and/or colour constancy²⁸, contradicting classical evidence for a retinal origin.^{19,23} Top-down effects on colour perception are well known.^{10,48} Such effects may be particularly strong for illusory percepts like afterimages where a physical stimulus is absent.³² However, evidence for mid-level (e.g., contours and shape^{30,32}) or high-level (e.g., knowledge^{31,46}) effects on afterimages do not necessarily reflect the origin of afterimages. Afterimages occur in the absence of top-down cues or knowledge, as in the present experiments. Hence, top-down interference effects may modulate the subjective appearance of afterimages; but they are not the origin of afterimage formation. The cone-adaptation model leaves little variance of perceived afterimages unexplained (estimated 21% with smoothed data) that would require additional adaptation at later, subsequent levels of processing. Speculating about yet

unknown cortical mechanisms to explain the complementary colours of afterimages is thus unnecessary. (p.8)

If we aim at experimentally investigating complementary colours in afterimages, then we need to choose a paradigm that controls other factors and effects that could contaminate or confuse our results on afterimages. The fact that this study focused on uniformly coloured areas has the purpose of investigating specifically the factors that determine the formation of complementary afterimages without getting confused by other factors. So, it seems to me that the only valid limitation of this choice could be that the sample of colours featured in this study are not representative of afterimages. Following the principle of univariance, spectral properties of the stimuli are lost. So, the question of representativity is about how well the stimulus samples in this study represent isoluminant chromaticities in general. Considering how widely colours have been sampled across the three experiments, including the sub-experiments, the sampling might be close to exhaustive. Sensitivity to colour differences is limited. According to an older study of mine (Witzel et al., 2013; Witzel & Gegenfurtner, 2018) 72%-visibility JNDs for hue discrimination in CIELUV varies between 7 and 17 degrees azimuth (see Figure below). In Experiment 2, we sampled inducers in steps of 5 deg azimuth, which is below those JNDs. The coverage of chroma levels in Experiment 3 has been maximised within technical limits. Due to Weber's law, chroma discrimination is high for desaturated colours (see second figure pasted below, taken from Figure 5 of Witzel et al. (2019)). We can't reliably measure afterimages for very desaturated colours because they are too faint. Another limitation is that we can't measure afterimages for maximally saturated colours because such measurements risk being contaminated by the display gamut (red line in below figure). Given those limitations, measurements for chromas of 20, 30, 50, and at maximum within gamut seem to provide quite a good impression of the variation of afterimages across chroma. I don't think any previous study has provided measurements of afterimages that came even close to this large sample of colour stimuli. I believe these stimuli are as representative as possible for complementary afterimages. Apart from the representativity of the stimulus sample, I don't think it is key for this study to discuss more broadly that experimental confounders in natural images or scenes might tap into other mechanisms of adaptation or more broadly other effects on colour appearance that are not the topic of this study.

4. More details about models needed.

The results were unnecessarily difficult to follow, even for this somewhat educated reader, and could be unintelligible to the general audience. Most importantly, it would greatly help to formally spell out in the results section exactly what the cone-opponency model is- preferably in equations. It's weird to me how hard this is, so I must be missing something obvious, but I am guessing that the cone-opponent model includes a mechanism that compares L-M to M-L (with each half-wave rectified?) and that adaptation subtracts the activity produced produced by the adapter in the activated sub-mechanism? Similarly, the paper should spell out more formally what the Hering-opponency model and cone-adaptation models are. I also imagine a general audience would deserve definitions of the color spaces and color terms like hue and chroma up front, as well as some experimental details like the number of observers.

RESPONSE

I thoroughly revised the presentation of models and predictions, including *Figure 1*, as already mentioned in response to *Comment 2*. I also revised the introductory part of the Model subsection of the *Method*:

With all models, the colour signal of the grey probe circle (same as background colour) was calculated under local adaptation to the inducer. Then, the colour of the resulting afterimage was determined as the locally adapted colour signal under global adaptation to the grey background. The known fact that afterimages are not as saturated as inducers (*Figure 2.f*) implies that adaptation to inducers is not complete. As an estimate of the strength of adaptation, we set the adapting chroma for modelling afterimages to the chroma of the comparison colours in *Experiment 1*, and to the grand average chroma (27.0 in CIELUV and 0.47 in DKL) of adjustments in *Experiment 2*. Even if we assumed complete adaptation to the chroma of inducers, results would be largely the same. However, not surprisingly, correlation coefficients involving cone adaptation would be slightly lower because non-linearities in the model are then higher than in the measurements. There are no free parameters in any of the models. (p.3)

I believe the revised version makes the logic behind the analyses much clearer. Below follow responses about detailed requests.

(4.a) Explanation of the cone-opponent model

Yes, the comment is correct with the description of the cone-opponent model. A subtractive model was used. More details on the model, including equations are now provided in the Method section:

Second-stage adaptation. Cone-opponency has been modelled with DKL colour space. (Derrington et al., 1984) Cone-opponent adaptation is assumed to be subtractive.^{11,19} So, it was calculated by subtracting the cone-opponent signal of the adapting colour from the cone-opponent signal of the achromatic probe.²⁰ [...]

(2) Cone-opponent adaptation to inducer:

$$DKL_{L-M} = CC_M - CC_L - k * iDKL_{L-M}$$

$$DKL_S = CC_S - k * iDKL_S$$

Where CC is the cone contrast calculated with equation (1), and the index indicates the respective cone (details on the cone-opponent axes are provided in section D-2 of the supplementary material). $iDKL_{LM}$ and $iDKL_S$ correspond to the cone-opponent signal of the inducer. Cone-opponent adaptation produces proportional shifts along the axes, implying cone-opponent afterimage hues, 180 degrees rotated away from the inducer and afterimage chroma that is k times the inducer chroma. Here, k was set to match the estimated strength of the afterimages, but the value of k was irrelevant to the results.

A model that distinguishes between 6 mechanisms (half-wave rectification) did not change the results. I have added this information at the end of the above paragraph:

Although, a theoretically complete model would differentiate between the two separate mechanisms that constitute each opponent channel ("half-wave rectification"), such a separation did not change the predictions in this context.

The predictions following from this model have been revised at the beginning of the *Results* section:

In contrast, cone-opponent afterimages result from subtractive, second-site adaptation, as illustrated by the red circle in *Figure 1.c*.

And visualised in *Figure 1.c*:

In addition, details about the mathematical formula and its relationship to other formulations of DKL-space were added to subsection *D.2 Cone-opponent model* in the *Supplementary Material*.

(4.b) Hering-process model

I improved the short explanation of the Hering opponent model in the Method:

Hering-opponent colours were calculated by linearly interpolating the hue direction between the respective two empirically measured prototypes for red, yellow, green, and blue in CIELUV space. There was no prediction for Hering chroma. For this, the 51 observers of Experiment 1.a adjusted colours of circles to match what they considered to be the prototypes following previous procedures (See section “Colour Naming” in Supplementary Material).⁵⁹

And I added a detailed explanation to section *D.7 Hering Opponent Model* in the *Supplementary Material*:

7 Hering-Opponent Model

The prototype adjustments task (see section A) yielded average typical hues in CIELUV at 10.6 deg for red, 72.2 deg for yellow, 125.1 deg for green and at 227.9 deg for blue (see Table S2). These hue directions are similar to the unique hues measured by Witzel & Gegenfurtner (2018, Figure 1) and Witzel et al. (2019, see Table S1). They were thus taken as estimates of unique red, yellow, green, and blue and as the directions for the green-vs-red (10.6 vs 125.1 deg) and blue-vs-yellow (227.9 vs 72.2 deg) axis of the Hering-opponent model. CIELUV azimuths in between those Hering-axes were linearly interpolated to produce a proportion of the distance between the adjacent axes and carve out four new quadrants lying in between the red-yellow, yellow-green, green-blue, and blue-yellow directions. This implies that hues in CIELUV are compressed when Hering axes were close in CIELUV. This was the case between red and yellow, which differ by 61.6 deg, and between yellow and green, which differ by only 53 deg in CIELUV. In contrast, hues were stretched out between blue and red, which differed by 142.7 deg. These compressions and stretches can be seen from the density of the black dots in Figure 1.e of the main article, which represent colours opponent in Hering coordinates for inducer hues that were equally distant in CIELUV. To calculate opponent colours in Hering coordinates, the inducer was first expressed as a proportion of the difference between two adjacent unique hues. For examples, an inducer of 60 deg in CIELUV lies in the red-yellow quadrant. It therefore corresponds to $(60 \text{ deg} - \text{red}) / (\text{yellow} - \text{red}) = 60 \text{ deg} - 10.6 \text{ deg} / 61.6 \text{ deg} = 80.2\%$. This proportion of the inducer would be applied to the quadrant opposite to the inducer quadrant. In the example, that would 80.2% off green in the green-blue quadrant. This corresponds to $\text{green} + 80.2\% * (\text{blue} - \text{green}) = 125.1 \text{ deg} + 0.802 * 102.8 \text{ deg} = 207.5 \text{ deg}$. Pulling these 2 steps together in one equation results in proportional coordinates RYGB for Hering-opponent colours:

$$RYGB = UH3 + \frac{\text{inducer} - UH1}{UH2 - UH1} * (UH4 - UH3)$$

Where all variables are in degree azimuth in CIELUV; $UH1$ and $UH2$ are the unique hues with an azimuth below (clockwise) and above (counterclockwise) to the inducer hue; $UH3$ and $UH4$ are the lower and upper unique hues of the opposite quadrant. Differences are circular differences.

Predictions from Hering opponency have been revised in the Results section:

The Munsell system and Hering colours (Figure 1.e) provide yet another approach to identify complementary colours, which have been considered as points of reference for colour appearance.^{7,8}

And visualised in Figure 1.e

(4.c) Cone-Adaptation model

Details of the cone-adaptation model have been added to the respective section of the *Method*, including the equation:

First-stage adaptation. Cone adaptation at the first stage of colour processing was modelled by cone contrasts.¹¹⁻¹⁶ Cone contrasts (CC) are Weber fractions, calculated as the difference between cone excitations of the stimulus (LMS) and cone excitations of the adapting colour (LMS0) relative to the cone excitations of the adapting colour: $CC = (LMS - LMS0) / LMS0$. Cone contrast is calculated independently for the short- (S), medium- (M), and long-wavelength (L) sensitive cones, resulting in S-cone, M-cone, and L-cone contrasts. In psychophysical experiments, adaptation is typically controlled by the colour of the background, which is then *LMS0*. To compute the induced colour of the afterimage, the roles are swapped because we model perception of the achromatic background after local adaptation to the inducer. Local adaptation to the inducer is modelled by inserting the cone excitations of the inducer, *iLMS*, instead of the background into *LMS0*. As the afterimage is elicited on a grey probe, *LMS* now corresponds to the achromatic, isoluminant grey of the background:

(1) Cone adaptation to inducer:

$$CC_{LMS} = \frac{bgLMS - iLMS}{iLMS}$$

Where *bgLMS* and *iLMS* refer to Stockman-Sharpe cone excitations of the grey background and the adapting inducer. Cone excitations were scaled to match the luminous efficiency function.⁵¹ Cone contrast changes with increasing cone adaptation *iLMS* according to a multiplicative inverse function (because *iLMS* increases in the denominator). This adaptation produces shifts towards the peak cone sensitivities (Figure S8), as illustrated by the three peaks in the hue histograms (Figure 1.c, Figure 2.d, h).

Figure S8 in Section D.1 *Cone Adaptation* of the *Supplementary Material* explains how the nonlinearities / multiplicative inverse affects induced colours.

Predictions by the cone-adaptation model are presented at the beginning of the Results section like this:

Predictions for cone adaptation are computed with the mathematical model of Weber's law.¹¹⁻¹⁶ Using the inducer as the adapting colours produces the triangular shape of the red curve in Figure 1.b. This means that cone adaptation does agree with the wide-spread assumption that afterimages are cone-opponent. (p.4)

(4.d) Definitions of colour space

I revised the presentation of other models in the *Method*:

Adaptation of colour appearance. Predictions by colour appearance models (CIELUV, CIELAB, CIECAM02) were computed as the colour appearance of the grey background when the adapting white-point was the inducer, using the respective adaptation transform. At isoluminance, CIELUV is a projective transformation of DKL-space and involves a subtractive adaptation, which explains the identity of predicted hues. Sections D3-D5 in the supplementary material provide mathematical details and explanations. For predictions by Munsell and Hering colours, opponent colours were determined in the respective coordinate systems, which is equivalent to the effects of subtractive adaptation. Opponent colours in the Munsell system were interpolated using the CIELAB coordinates of the Munsell renotation table⁵² as provided by the Munsell Lab of the Rochester Institute of Technology. Details about the interpolations are provided in section *D-6 Munsell-Opponent Model of the Supplementary Material*. Hering-opponent colours were calculated by linearly interpolating the hue direction between the respective two empirically measured prototypes for red, yellow, green, and blue in CIELUV space. There was no prediction for Hering chroma. For details, see section *D-7 Hering Opponent Model of the Supplementary Material*. (p.4)

Predictions are presented at the beginning of the Results section:

Alternative models have been proposed to better predict several aspects of colour appearance than those physiological models and might better match the colour appearance of afterimages, either. CIELUV, CIELAB, and CIECAM02 implement complex, nonlinear models to approximate various empirical data on colour appearance.⁷ At isoluminance, CIELUV is a projective transformation of DKL-space^{36,37} and the subtractive adaptation transform in CIELUV predicts the same afterimage hues as cone-opponency, but different afterimage chroma (*Figure 1.c*, see also section *D-3* in the *Supplementary Material*). CIELAB and CIECAM02 (*Figure 1.d*) incorporate adaptation transforms inspired by von-Kries adaptation and produce a similar triangular shape as cone-adaptation but tilted. The Munsell system and Hering colours (*Figure 1.e*) provide yet another approach to identify complementary colours, which have been considered as points of reference for colour appearance.^{7,8} (p.4)

I also added thorough explanations of the equations and the effects of adaptation transforms in sections D.3-6 of the supplementary material.

(4.d) Definitions of hue and saturation

I added a definition of hue and saturation at the beginning of the Results section:

Hue describes how reddish, yellowish, greenish, and bluish a colour is. Chroma refers to the colourfulness of a hue. At equal brightness (isoluminance), it is equivalent to saturation and corresponds to the difference of a colour from grey.

(4.e) The number of observers

The number of observers has been given in the respective Participants sections in the Method and is often mentioned in the Results if particularly important. For Experiment 1:

In **Experiment 1.a**, 32 observers participated (20 women, age: 25.2±3.94y). In **Experiment 1.b**, 52 observers took part (36 women, age: 25.1±4.3years).

For Experiment 2:

Ten voluntary participants (6 women), including the author (CW) took part (cf. Table S4, Figures S2-4) in **Experiment 2.a** and CW and f1 participated in **Experiment 2.b** (cf. Table S4, Figures S6).

The number of observers has also been mentioned in the Results section:

These results could also be produced at the individual level (*Figures S2-S4*): For each of 10 observers, [...]

For Experiment 3, there was some information missing, which I added (highlighted green):

In Experiment 3, measurements across chroma were done by the author (CW) and four of the nine naïve participants (f7, f8, f9, 10; cf. *Figure 3* and *Figure S7*).

The number of participants in Experiment 3 has also been mentioned in the Results section:

Figure 3 illustrates the extensive measurements with observer f7, but results are replicated with four other observers and different stimulus samples (*Figure S6*).

And in the caption of Figure 3:

Data for observer f7. Data for four other observers can be found in *Figure S6*, each replicating these results.

5. Discussion: More discussion of the mechanisms of cone adaptation needed.

Zaidi et al. suggest that physiologically measured cone adaptation recovers too rapidly to account for afterimages. That paper also distinguishes between light levels where adaptation occurs due to photoreceptor "bleaching" and levels where adaptation must have some other mechanism- presumably some sort of early gain control. The discussion of the current paper seems to adopt the bleaching model- that simply the isomerization of retinal can explain the adaptation producing color afterimages. But based on the Zaidi paper I would ask, but does that model match the slow timecourses seen here and can it explain afterimage presence at relatively low light levels?

RESPONSE

(5.a) Speed of adaptation

I believe the first part of the comment refers to the following statements from Zaidi et al. (2012):

We then use in vivo electrophysiological recordings to show that all three classes of primate retinal ganglion cells exhibit subtractive adaptation to prolonged stimuli, with much slower time constants than those expected of photoreceptors. (from the abstract, p.220)

And:

Because the estimated time constants were two to three orders of magnitude slower than photoreceptor adaptation, the adaptation must occur after the photoreceptors and at or before the RGCs. (p.221)

This is a classical argument that goes back to Loomis' experiments (Loomis, 1972; Loomis, 1978), who found different afterimage strengths for flickering (at 4Hz) and static inducers with the same average colour. He argued that photopigment bleaching is fast and should be achieved across the alternation of the flickering inducers, thus resulting in equivalent afterimages as with the static inducers. As this was not the case, he concluded that complementary afterimages are produced by a slow process of adaptation in contrast to fast photopigment bleaching:

Such after-images, which exhibit no interocular transfer, must be produced by a process of slow neural adaptation that is controlled chiefly by a correlate of color appearance. (Loomis, 1972, p.1593)

I agree with the general idea, as expressed by Zaidi et al. (2012), but doubt the validity of those classical experiments, especially because more recent experiments found contrary results. I added a discussion of the classical criticisms to the Discussion:

Classical criticisms of the cone-bleaching explanation have been the difference between afterimages from alternating and static inducers, the duration of afterimages (>10s) being much longer than cone adaptation (<<100ms), and the low level of isomerisation at usual brightness levels.^{20,32,33} These criticisms seem questionable. First, similar chromatic afterimages for alternating and static inducers have been found more recently,^{52,53} and the perception of afterimages from dynamic inducers may be affected by factors that are unrelated to receptor adaptation.²⁶ Second, it has been recognised that cone regeneration may take minutes rather than milliseconds.^{54,55} Third, isomerisation levels correspond to the cone responses to the adapting stimulus, here the inducer. It seems contradictory that the level of isomerisation under usual brightness levels (1-10%) is sufficient to produce the saturated non-illusory colours of the inducers while being considered insufficient to explain the less saturated afterimages. (p.7)

More importantly, I explain the seeming contradiction between the different time scales through the temporary maintenance of the adapted colour signal at the second stage, similar to a suggestion in Mike Webster's review (Webster, 2016):

After cone adaptation, the adapted cone signal is propagated through the subsequent stages of colour processing.^{3,31,44} The rebound signal observed in the retinal ganglion cells indicates that the cone-adapted signal is temporally maintained at the second, cone-opponent stage.²⁰ This temporal maintenance likely contributes to the sustained duration of afterimages.^{3,20,32,33} According to the present results, processing of the colour signal at the second-stage seems not to affect the perceived colours of afterimages in any specific way. (p.7)

This interpretation is in line with Zaidi's et al.'s (2012) observation of a rebound signal but contradicts their fierce rejection of photoreceptor adaptation. One reason why Zaidi et al. (2012) did not realise that afterimage colours could not be explained by subtractive post-receptor adaptation is their nulling method. I have pointed this out in the Discussion:

A cursory look at afterimages may be misleading because afterimages resemble cone-opponency at a very coarse level.^{34,38} This is not due to adaptation of the cone-opponent mechanisms at the second stage of colour processing, but to the cone-adapted colour signal from the first stage being propagated to the second and subsequent stages. The success of afterimage nulling procedures (adjustments that cancel the afterimage) in cone-opponent space^{20,39} do not support the cone-opponency of afterimages, either. Successful nulling works independently of whether the appearance of afterimages follows a straight line due to subtractive cone-opponent adaptation, or a curve due to nonlinear, divisive adaptation, such as cone adaptation. However, the deviations of afterimage hues from cone opponency are strong enough to be visible in first-person experience. This can be demonstrated using the chaser-like arrangement on a standard computer display as in the attached animated visualisations (for instructions, see section E of the *Supplementary Material*).

(5.b) Intensity of stimulation

I believe the second part of the comment refers to the following statements from Zaidi et al. (2012):

Evidence for afterimages resulting from adaptation in independent (L, M, S) cone classes [7] often used lights many orders of magnitude brighter than lights that generate afterimages in our method, thus causing pigment bleaching, but psychophysical evidence shows that afterimages of lights also occur at midphotopic levels [13]. (p.220)

And:

This result should correct the notion, found on the web and in many textbooks, that photoreceptor desensitization is responsible for color after-images generated by normal light levels. An elegant experiment has demonstrated afterimages generated by independent

photoreceptors [7], but this requires intense lights that are bright enough to bleach substantial amounts of photopigment, so it is a photochemical, rather than neural, effect. (p.222)

This is also a classical argument that goes back to Loomis (Loomis, 1972; Loomis, 1978), and is best summarised in this passage from Loomis (1972):

Given the low levels of bleaching here (at most 11%),¹⁸ loss of sensitivity resulting directly from the reduced concentration of available photopigment is not at issue.

Our displays had a luminance of 41 cd/m² in *Experiment 1* and 101/140cd/m² in *Experiments 2-3*. When compared to the luminance in Zaidi et al's (2012) study (63cd/m²), our displays do not involve "lights many orders of magnitude brighter than lights that generate afterimages" in their study. Furthermore, the brightness levels in this study correspond to bleaching of 1.7%-3.5% cone bleaching according to the Rushton and Henry half-bleach constant (Rushton & Henry, 1968), which is equal or less than those reported and discussed by Loomis (Loomis, 1972; Loomis, 1978). In other words, the present study contradicts this classical criticism.

At a closer look, that classical argument seems a bit peculiar. The amount of bleaching that results from adaptation to the inducers corresponds to the colour of the inducer, especially if we believe the cone-adaptation model. If this amount of isomerisation is sufficient to produce the vibrant colours of the inducers, then how can it be too little to explain the slightly less vibrant colours of complementary afterimages? More generally, arguments about absolute levels of isomerisation seem dubious to me when it comes to explaining the formation of the colour signal. In colour vision, the small difference between M- and L-cone activation produces a whole colour dimension, the *L-M* channel. If one compares the size of that difference with the absolute size of the cone excitations, one will likely miss the importance of the *L-M* dimension for colour vision. Comparisons of absolute intensities may neither be an adequate approach to understand colour perception in general, nor afterimages in particular. A short discussion of this issue may be found in the same paragraph I had cited above (5.a):

Classical criticisms of the cone-bleaching explanation have been the difference between afterimages from alternating and static inducers, the duration of afterimages (>10s) being much longer than cone adaptation (<<100ms), and the low level of isomerisation at usual brightness levels.^{20,32,33} These criticisms seem questionable. First, similar chromatic afterimages for alternating and static inducers have been found more recently,^{52,53} and the perception of afterimages from dynamic inducers may be affected by factors that are unrelated to receptor adaptation.²⁶ Second, it has been recognised that cone regeneration may take minutes rather than milliseconds.^{54,55} Third, isomerisation levels correspond to the cone responses to the adapting stimulus, here the inducer. It seems contradictory that the level of isomerisation under usual brightness levels (1-10%) is sufficient to produce the saturated non-illusory colours of the inducers while being considered insufficient to explain the less saturated afterimages. (p.7)

I believe that the patterns revealed and replicated throughout the experiments reported in this manuscript provide way more convincing evidence than Loomis' speculative argument, but see 5.c for alternatives to cone bleaching according to the state-of-the-art.

5.c Relevance of the cone bleaching hypothesis

I realise that the original manuscript sounded like cone bleaching is the only explanation for Weber-like adaptation of the cone-signal. It is a very detailed and plausible explanation, but I neither think it is necessarily the only explanation, nor that it is crucial for the findings in this study how exactly Weber-like, divisive adaptation of the cone signal is biochemically implemented. I reworded the presentation of the cone-bleaching explanation in the Discussion as a likely, rather than the only candidate:

The cone-adaptation model used here implements “divisive adaptation” according to Weber’s Law, which implies that cone sensitivities decrease proportionally to the magnitude of current cone excitation. According to the classical explanation of afterimages through “cone bleaching,” this model reflects the proportional decrease of photon catches by the cone photoreceptors.^{28,30,54} [...] (p.7)

I also acknowledge the possibility that other processes may produce Weber-like adaptation of the cone-signal:

At the same time, it has become clear that cone adaptation at moderate light levels involves processes other than “cone bleaching.”^{56,57} In addition, the time course of adaptation is complicated⁵⁸⁻⁶¹ and differs across cone types.^{62,63} The tasks in the present study were designed to measure afterimage colours close to maximum adaptation and to avoid temporal effects during afterimage formation and fading. The complexity of the underlying physiological processes notwithstanding, the resulting divisive cone adaptation to moderately bright, photopic colours can be approximated by cone contrasts according to Weber’s law.^{11-14,16,51} (p.7)

I hope it became clear through the revision of the Introduction that the key finding is the link between type of divisive, Weber-like adaptation of the cone signal and the perceived colours of the afterimages. This is the reason why this point is discussed in detail in the Discussion:

Instead, divisive, Weber-like cone-adaptation explains why the hues of afterimages are not cone-opponent, not reciprocal, and produce three clusters of hue.^{25,45} The multiplicative inverse effect of inducer cone excitations produces non-linear effects of cone adaptation that result in the three hue clusters and the three peaks of afterimage chroma in comparison to cone-opponency (Figure 2.d,f,h). [...] (p.7)

6. Minor: Smoothed Data

Correlation coefficients are not accurate (or at least their corresponding p-values are not) for non-independent samples- this is definitely the case for the smoothed data, and could be the case for the nonsmoothed data (e.g. if multiple points are from the same adapter). A more complicated statistic may be needed. Smoothing is not a standard English word.

RESPONSE

(6.a) Experiment 1: Multiple response options per adapter

To avoid having more data points than adapters, I binned response options into 24 bins corresponding to the 24 adapters instead of using the frequencies for the 72 response options. This has the additional benefit of avoiding the artifacts that I tried to compensate through smoothing, thus improving the diagram and making the Results section simpler, shorter, and more straight-forward. The new polar histogram in Figure 2.d looks like this now:

And these are the corresponding results (changes highlighted green):

A hue histogram was calculated to capture the three hue clusters resulting from those shifts away from opponency. For this, responses were counted in 24 bins of 15-degree across all 24 inducers. These were represented in a polar plot, where distance to origin corresponds to relative frequency of response choices (grey area in Figure 2.d). The effect of cone adaptation on the hue histogram was simulated. Normally distributed response noise (black dots in Figure 1.b) was added to obtain a continuous probability distribution (red curve in Figure 2.d). The cone-adaptation model predicted the frequencies of hue selections with a correlation of $r(22) = .71$, $p < .001$, across the 24 hue bins. (p.5)

(6.b) Smoothing in Experiment 2

I removed inference statistics for the smoothed data. The smoothing serves the purpose of estimating how much variance is explained when we discount noise. Here is the revised part of the Results section on Experiment 2:

As in the other task, deviations from cone-opponent hues were strongly correlated between simulated and measured afterimages ($r(70) = .84$, $p < .001$, Figure 2.g), and the hue histogram for measured and modelled afterimages were strongly correlated ($r(70) = .82$, $p < .001$; Figure 2.h). Estimates from the correlation with smoothed data (across 9 adjacent bins) suggest that the cone-adaptation model might explain around 79% of variance of afterimage hue matches when discounting for measurement noise. (p.6)

(6.c) Smoothing vs Smoothing

I changed those instances from smoothing to smoothing. Apart from the above citation (6.b), this was done in the capture of Figure 2:

In panel h, a smoothed version of the hue histogram is shown by the black line.

REFERENCES

- Anstis, S., Rogers, B., & Henry, J. (1978). Interactions between simultaneous contrast and coloured afterimages. *Vision Research*, 18(8), 899-911. [https://doi.org/10.1016/0042-6989\(78\)90016-0](https://doi.org/10.1016/0042-6989(78)90016-0)
- Baccus, S. A., & Meister, M. (2002). Fast and Slow Contrast Adaptation in Retinal Circuitry. *Neuron*, 36(5), 909-919. [https://doi.org/10.1016/S0896-6273\(02\)01050-4](https://doi.org/10.1016/S0896-6273(02)01050-4)
- Brainard, D. H. (1996). Cone contrast and opponent modulation color spaces. In P. K. Kaiser & R. M. Boynton (Eds.), *Human Color Vision* (2 ed., pp. 563-579). Optical Society of America.
- Brainard, D. H., & Stockman, A. (2010). Colorimetry. In M. Bass (Ed.), *OSA Handbook of Optics* (pp. 10.11-10.56). McGraw-Hill.

- Brindley, G. S. (1962). Two new properties of foveal after-images and a photochemical hypothesis to explain them. *Journal of Physiology*, *164*, 168-179. <https://doi.org/10.1113/jphysiol.1962.sp007011>
- Burkhardt, D. A. (1994). Light adaptation and photopigment bleaching in cone photoreceptors in situ in the retina of the turtle. *Journal of Neuroscience*, *14*(3), 1091-1105.
- Burkhardt, D. A., & Gottesman, J. (1987). Light adaptation and responses to contrast flashes in cones of the walleye retina. *Vision Research*, *27*(9), 1409-1420. [https://doi.org/10.1016/0042-6989\(87\)90151-9](https://doi.org/10.1016/0042-6989(87)90151-9)
- Conway, B. R., Malik-Moraleda, S., & Gibson, E. (2023). Color appearance and the end of Hering's Opponent-Colors Theory. *Trends in Cognitive Sciences*. <https://doi.org/10.1016/j.tics.2023.06.003>
- Craik, K. J. W. (1940). Origin of Visual After-images. *Nature*, *145*(3674), 512-512. <https://doi.org/10.1038/145512a0>
- Derrington, A. M., Krauskopf, J., & Lennie, P. (1984). Chromatic mechanisms in lateral geniculate nucleus of macaque. *The Journal of Physiology*, *357*(1), 241-265. <https://doi.org/10.1113/jphysiol.1984.sp015499>
- Dong, B., Holm, L., & Bao, M. (2017). Cortical mechanisms for afterimage formation: evidence from interocular grouping [Article]. *7*, 41101. <https://doi.org/10.1038/srep41101>
- Dunn, F. A., Lankheet, M. J., & Rieke, F. (2007). Light adaptation in cone vision involves switching between receptor and post-receptor sites. *Nature*, *449*(7162), 603-606. <https://doi.org/10.1038/nature06150>
- Fairchild, M. D. (2013). *Color appearance models*. Wiley.
- Goldstein, E. B., & Brockmole, J. (2016). *Sensation and perception*. Cengage Learning.
- Goldstein, E. B., & Brockmole, J. R. (2017). *Sensation and perception* (Tenth edition ed.). Cengage Learning. <https://ebookcentral.proquest.com/lib/soton-ebooks/detail.action?docID=5132700>
- Goldstein, E. B., & Cacciamani, L. (2021). *Sensation and perception* (Eleventh edition ed.). Wadsworth.
- Hayhoe, M. M., Benimoff, N. I., & Hood, D. C. (1987). The time-course of multiplicative and subtractive adaptation process. *Vision Research*, *27*(11), 1981-1996. [https://doi.org/https://doi.org/10.1016/0042-6989\(87\)90062-9](https://doi.org/https://doi.org/10.1016/0042-6989(87)90062-9)
- He, S., & MacLeod, D. I. A. (2001). Orientation-selective adaptation and tilt after-effect from invisible patterns. *Nature*, *411*(6836), 473-476. <https://doi.org/10.1038/35078072>
- Hurvich, L. M., & Jameson, D. (1957). An opponent-process theory of color vision. *Psychological Review*, *64*(6, Pt.1), 384-404. <https://doi.org/10.1037/h0041403>
- Johnson, G. M. (2015). Color appearance phenomena and visual illusions. In A. J. Elliot, A. Franklin, & M. D. Fairchild (Eds.), *Handbook of Color Psychology* (pp. 679-702). Cambridge University Press. <https://doi.org/10.1017/CBO9781107337930.034>
- Kelber, A., & Osorio, D. (2010). From spectral information to animal colour vision: experiments and concepts. *Proceedings of the Royal Society B: Biological Sciences*, *277*(1688), 1617-1625. <https://doi.org/10.1098/rspb.2009.2118>
- Kelly, D. H., & Martinez-Uriegas, E. (1993). Measurements of chromatic and achromatic afterimages. *Journal of the Optical Society of America. A, Optics and image science*, *10*(1), 29-37. <https://doi.org/10.1364/josaa.10.000029>
- Kingdom, F. A. A., Touma, S., & Jennings, B. J. (2020). Negative afterimages facilitate the detection of real images. *Vision Research*, *170*, 25-34. <https://doi.org/10.1016/j.visres.2020.03.005>
- Koenderink, J., van Doorn, A., Witzel, C., & Gegenfurtner, K. (2020). Hues of Color Afterimages. *i-Perception*, *11*(1), 2041669520903553. <https://doi.org/10.1177/2041669520903553>
- Krauskopf, J., Williams, D. R., Mandler, M. B., & Brown, A. M. (1986). Higher order color mechanisms. *Vision Research*, *26*(1), 23-32. [https://doi.org/0042-6989\(86\)90068-4](https://doi.org/0042-6989(86)90068-4) [pii]
- Lee, B. B., Smith, V. C., Pokorny, J., & Sun, H. (2008). Chromatic adaptation in red-green cone-opponent retinal ganglion cells of the macaque. *Vision Research*, *48*(26), 2625-2632. <https://doi.org/10.1016/j.visres.2008.01.007>

- Lee, R. J., & Mather, G. (2019). Chromatic adaptation from achromatic stimuli with implied color. *Attention, Perception, & Psychophysics*, 81(8), 2890-2901. <https://doi.org/10.3758/s13414-019-01716-5>
- Loomis, J. M. (1972). The photopigment bleaching hypothesis of complementary after-images: a psychophysical test. *Vision Research*, 12(10), 1587-1594. [https://doi.org/10.1016/0042-6989\(72\)90031-4](https://doi.org/10.1016/0042-6989(72)90031-4)
- Loomis, J. M. (1978). Complementary afterimages and the unequal adapting effects of steady and flickering light*. *Journal of the Optical Society of America*, 68(3), 411-416. <https://doi.org/10.1364/JOSA.68.000411>
- Lupyan, G. (2015). Object knowledge changes visual appearance: Semantic effects on color afterimages. *Acta Psychologica*, 161, 117-130. <https://doi.org/10.1016/j.actpsy.2015.08.006>
- Manzotti, R. (2017). A Perception-Based Model of Complementary Afterimages. *SAGE Open*, 7(1), 2158244016682478. <https://doi.org/10.1177/2158244016682478>
- Mollon, J. D., & Polden, P. G. (1976). Absence of transient tritanopia after adaptation to very intense yellow light. *Nature*, 259(5544), 570-572. <https://doi.org/10.1038/259570a0>
- Mollon, J. D., Stockman, A., & Polden, P. G. (1987). Transient tritanopia of a second kind. *Vision Research*, 27(4), 637-650. [https://doi.org/10.1016/0042-6989\(87\)90048-4](https://doi.org/10.1016/0042-6989(87)90048-4)
- Munsell, A. H. (1912). A Pigment Color System and Notation. *The American Journal of Psychology*, 23(2), 236-244. <https://doi.org/10.2307/1412843>
- Munsell, A. H. (1921). *A Grammar of Color*. Strathmore Paper Company. <https://munsell.com/color-blog/grammar-of-color-munsell-book-online/>
- Phuangsuwan, C., Ikeda, M., & Mepean, J. (2018). Color appearance of afterimages compared to the chromatic adaptation to illumination. *Color Research & Application*, 43(3), 349-357. <https://doi.org/doi:10.1002/col.22207>
- Pöppel, E. (1986). Long-range colour-generating interactions across the retina. *Nature*, 320(6062), 523-525. <https://doi.org/10.1038/320523a0>
- Pugh, E. N., & Lamb, T. D. (2000). Chapter 5 Phototransduction in vertebrate rods and cones: Molecular mechanisms of amplification, recovery and light adaptation. In D. G. Stavenga, W. J. DeGrip, & E. N. Pugh (Eds.), *Handbook of Biological Physics* (Vol. 3, pp. 183-255). North-Holland. [https://doi.org/10.1016/S1383-8121\(00\)80008-1](https://doi.org/10.1016/S1383-8121(00)80008-1)
- Pugh, E. N., & Mollon, J. D. (1979). A theory of the π_1 and π_3 color mechanisms of Stiles. *Vision Research*, 19(3), 293-312. [https://doi.org/10.1016/0042-6989\(79\)90175-5](https://doi.org/10.1016/0042-6989(79)90175-5)
- Pugh, E. N., Nikonov, S., & Lamb, T. D. (1999). Molecular mechanisms of vertebrate photoreceptor light adaptation. *Current Opinion in Neurobiology*, 9(4), 410-418. [https://doi.org/10.1016/S0959-4388\(99\)80062-2](https://doi.org/10.1016/S0959-4388(99)80062-2)
- Rhyne, T.-M. (2017). *Applying Color Theory to Digital Media and Visualization*. CRC Press, Taylor & Francis Group.
- Rushton, W. A., & Henry, G. H. (1968). Bleaching and regeneration of cone pigments in man. *Vision Research*, 8(6), 617-631. [https://doi.org/10.1016/0042-6989\(68\)90040-0](https://doi.org/10.1016/0042-6989(68)90040-0)
- Schanda, J. (2016). CIE u' , v' Uniform Chromaticity Scale Diagram and CIELUV Color Space. In M. R. Luo (Ed.), *Encyclopedia of Color Science and Technology* (pp. 185-188). Springer New York. https://doi.org/10.1007/978-1-4419-8071-7_12
- Shapley, R., & Enroth-Cugell, C. (1984). Visual adaptation and retinal gain controls. *Progress in retinal research*, 3, 263-346. [https://doi.org/10.1016/0278-4327\(84\)90011-7](https://doi.org/10.1016/0278-4327(84)90011-7)
- Shimojo, S., Kamitani, Y., & Nishida, S. (2001). Afterimage of perceptually filled-in surface. *Science*, 293(5535), 1677-1680. <https://doi.org/10.1126/science.1060161>
- Stockman, A., & Brainard, D. H. (2010). Color vision mechanisms. In M. Bass (Ed.), *OSA Handbook of Optics* (3 ed., pp. 11.11-11.104). McGraw-Hill.
- Stockman, A., Langendorfer, M., Smithson, H. E., & Sharpe, L. T. (2006). Human cone light adaptation: from behavioral measurements to molecular mechanisms. *Journal of Vision*, 6(11), 1194-1213. <https://doi.org/10.1167/6.11.5>
- Stromeyer, C. F., 3rd, Cole, G. R., & Kronauer, R. E. (1985). Second-site adaptation in the red-green chromatic pathways. *Vision Res*, 25(2), 219-237. <http://www.ncbi.nlm.nih.gov/pubmed/4013090>

- Thoen, H. H., How, M. J., Chiou, T. H., & Marshall, J. (2014). A different form of color vision in mantis shrimp. *Science*, 343(6169), 411-413. <https://doi.org/10.1126/science.1245824>
- Valeton, J. M., & Norren, D. V. (1979). Retinal site of transient tritanopia. *Nature*, 280(5722), 488-490. <https://doi.org/10.1038/280488a0>
- van Lier, R., Vergeer, M., & Anstis, S. (2009). Filling-in afterimage colors between the lines. *Current Biology*, 19(8), R323-324. <https://doi.org/10.1016/j.cub.2009.03.010>
- Webster, M. A. (2011). Adaptation and visual coding. *Journal of Vision*, 11(5). <https://doi.org/10.1167/11.5.3>
- Webster, M. A. (2016). Visual Adaptation. *Annual Review of Vision Science*, 1, 547-567. <https://doi.org/10.1146/annurev-vision-082114-035509>
- Webster, M. A., & Mollon, J. D. (1991). Changes in colour appearance following post-receptoral adaptation. *Nature*, 349(6306), 235-238. <https://doi.org/10.1038/349235a0>
- Whittle, P. (1986). Increments and decrements: luminance discrimination. *Vision Research*, 26(10), 1677-1691. [https://doi.org/0042-6989\(86\)90055-6](https://doi.org/0042-6989(86)90055-6) [pii]
- Williams, D. R., & MacLeod, D. I. (1979). Interchangeable backgrounds for cone afterimages. *Vision Research*, 19(8), 867-877. [https://doi.org/10.1016/0042-6989\(79\)90020-8](https://doi.org/10.1016/0042-6989(79)90020-8)
- Wilson, M. H., & Brocklebank, R. W. (1955). Complementary Hues of After-Images. *Journal of the Optical Society of America*, 45(4), 293-299. <https://doi.org/10.1364/JOSA.45.000293>
- Witzel, C., Flack, Z., & Franklin, A. (2013, 8-12. July 2013). Categorical colour constancy during colour term acquisition. AIC2013 - 12th international AIC congress, Newcastle upon Tyne, UK.
- Witzel, C., & Gegenfurtner, K. R. (2013). Categorical sensitivity to color differences. *Journal of Vision*, 13(7). <https://doi.org/10.1167/13.7.1>
- Witzel, C., & Gegenfurtner, K. R. (2018). Are red, yellow, green, and blue perceptual categories? *Vision Research*, 151, 152-163. <https://doi.org/10.1016/j.visres.2018.04.002>
- Witzel, C., Maule, J., & Franklin, A. (2019). Red, yellow, green, and blue are not particularly colorful. *Journal of Vision*, 19(14), 27-27. <https://doi.org/10.1167/19.14.27>
- Zaidi, Q., Ennis, R., Cao, D., & Lee, B. (2012). Neural locus of color afterimages. *Current Biology*, 22(3), 220-224. <https://doi.org/10.1016/j.cub.2011.12.021>
- Zaidi, Q., Marshall, J., Thoen, H., & Conway, B. R. (2014). Evolution of neural computations: Mantis shrimp and human color decoding. *Perception*, 5(6), 492-496. <https://doi.org/10.1068/i0662sas>
- Zeki, S., Cheadle, S., Pepper, J., & Mylonas, D. (2017). The Constancy of Colored After-Images [Original Research]. *Frontiers in Human Neuroscience*, 11(229). <https://doi.org/10.3389/fnhum.2017.00229>